# Testably Learning Polynomial Threshold Functions

**Lucas Slot**
Department of Computer Science
ETH Zurich
lucas.slot@inf.ethz.ch

**Stefan Tiegel**
Department of Computer Science
ETH Zurich
stefan.tiegel@inf.ethz.ch

**Manuel Wiedmer**
Department of Computer Science
ETH Zurich
manuel.wiedmer@inf.ethz.ch

## Abstract

Rubinfeld & Vasilyan recently introduced the framework of *testable learning* as an extension of the classical agnostic model. It relaxes distributional assumptions which are difficult to verify by conditions that can be checked efficiently by a *tester*. The tester has to accept whenever the data truly satisfies the original assumptions, and the learner has to succeed whenever the tester accepts. We focus on the setting where the tester has to accept standard Gaussian data. There, it is known that basic concept classes such as halfspaces can be learned testably with the same time complexity as in the (distribution-specific) agnostic model. In this work, we ask whether there is a price to pay for testably learning more complex concept classes. In particular, we consider polynomial threshold functions (PTFs), which naturally generalize halfspaces. We show that PTFs of arbitrary constant degree can be testably learned up to excess error $\varepsilon > 0$ in time $n^{\mathrm{poly}(1/\varepsilon)}$. This qualitatively matches the best known guarantees in the agnostic model. Our results build on a connection between testable learning and *fooling*. In particular, we show that distributions that approximately match at least $\mathrm{poly}(1/\varepsilon)$ moments of the standard Gaussian fool constant-degree PTFs (up to error $\varepsilon$). As a secondary result, we prove that a direct approach to show testable learning (without fooling), which was successfully used for halfspaces, cannot work for PTFs.

## 1 Introduction

The PAC learning model of Valiant [30] has long served as a test-bed to study which learning tasks can be performed efficiently and which might be computationally difficult. One drawback of this model is that it is inherently noiseless. In order to capture noisy learning tasks, the following extension, called the *agnostic model*, has been introduced [19, 24]: Let $\mathcal{F}$ be a class of boolean functions and let $\mathcal{D}_{\mathrm{joint}}$ be an (unknown) distribution over example-label-pairs in $\mathcal{X} \times \{\pm 1\}$. Typically, $\mathcal{X} = \{0,1\}^n$ or $\mathbb{R}^n$. As input, we receive iid samples from $\mathcal{D}_{\mathrm{joint}}$. For a small $\varepsilon > 0$, our task is to output a classifier $\hat{f}$ (not necessarily in $\mathcal{F}$) whose *loss* $L(\hat{f}, \mathcal{D}_{\mathrm{joint}}) := \mathbb{P}_{(x,z) \sim \mathcal{D}_{\mathrm{joint}}}(\hat{f}(x) \neq z)$ is at most $\mathrm{opt} + \varepsilon$, where $\mathrm{opt} := \inf_{f \in \mathcal{F}} L(f, \mathcal{D}_{\mathrm{joint}})$. The parameter $\mathrm{opt}$ thus indicates how "noisy" the instance is. We say that an algorithm *agnostically learns* $\mathcal{F}$ up to error $\varepsilon$ if it outputs such an $\hat{f}$. This model is appealing since it makes assumptions neither on the distribution of the input, nor on the type and amount of noise. After running an agnostic learning algorithm, we can therefore be certain that the output $\hat{f}$ achieves error close to that of the best function in $\mathcal{F}$ even without knowing what distribution the data came from.

**Efficient learning and distributional assumptions.** We are interested in understanding when agnostic learning can be performed efficiently. Unfortunately, efficient learning is likely impossible without making assumptions on the distribution $\mathcal{D}_{\text{joint}}$, even for very simple function classes $\mathcal{F}$. For instance, consider the class $\mathcal{F}_{\text{HS}}$ of *halfspaces*, i.e., boolean functions of the form $f(x) = \text{sign}(\langle v, x \rangle - \theta)$. Then, even if there exists a halfspace achieving arbitrarily small error, it is widely believed that outputting an $\hat{f}$ that performs better than a random guess in the agnostic model takes at least super-polynomial time if no assumptions are made on $\mathcal{D}_{\text{joint}}$ [8, 29]. To find efficient algorithms, one therefore has to make such assumptions. Typically, these take the form of assuming that the marginal $\mathcal{D}_{\mathcal{X}}$ of $\mathcal{D}_{\text{joint}}$ over the examples $\mathcal{X}$ belongs to a specific family of distributions.

**Definition 1** (Agnostic learning with distributional assumptions)**.** *Let $\varepsilon > 0$. A learner $\mathcal{A}$ agnostically learns $\mathcal{F}$ with respect to $\mathcal{D}$ up to error $\varepsilon$ if, for any distribution $\mathcal{D}_{\text{joint}}$ on $\mathcal{X} \times \{\pm 1\}$ whose marginal $\mathcal{D}_{\mathcal{X}}$ on $\mathcal{X}$ is equal to $\mathcal{D}$, given sufficient samples from $\mathcal{D}_{\text{joint}}$, it outputs with high probability a function $f : \mathcal{X} \to \{\pm 1\}$ satisfying $L(f, \mathcal{D}_{\text{joint}}) \le \text{opt}(\mathcal{F}, \mathcal{D}_{\text{joint}}) + \varepsilon$.*

For example, under the assumption that $\mathcal{D}_{\mathcal{X}}$ is standard Gaussian, we can find $\hat{f}$ such that $L(\hat{f}, \mathcal{D}_{\text{joint}}) \le \text{opt}(\mathcal{F}_{\text{HS}}, \mathcal{D}_{\text{joint}}) + \varepsilon$ in time $n^{O(1/\varepsilon^2)}$ [20, 11]. This runtime is likely best-possible [12, 29, 13].[1] Efficient learning is still possible under weaker assumptions on $\mathcal{D}_{\mathcal{X}}$, e.g., log-concavity [20]. Regardless, we cannot know whether a learning algorithm achieves its claimed error without a guarantee that the input actually satisfies our distributional assumptions. Such guarantees are inherently difficult to obtain from a finite (small) sample. Furthermore, approaches like cross-validation (i.e., computing the empirical error of $\hat{f}$ on a hold-out data set) fail in the noisy agnostic model, since we do not know the noise level $\text{opt}$. This represents a severe limitation of the agnostic learning model with distributional assumptions.

## 1.1 Testable learning.

To address this limitation, Rubinfeld & Vasilyan [28] recently introduced the following model, which they call *testable learning*: First, they run a *tester* on the input data, which attempts to verify a computationally tractable relaxation of the distributional assumptions. If the tester accepts, they then run a (standard) agnostic learning algorithm. The tester is required to accept whenever the data truly satisfies the distributional assumptions, and whenever the tester accepts, the output of the algorithm must achieve error close to $\text{opt}$. More formally, they define:

**Definition 2** (Testable learning [28])**.** *Let $\varepsilon > 0$. A tester-learner pair $(\mathcal{T}, \mathcal{A})$ testably learns $\mathcal{F}$ with respect to a distribution $\mathcal{D}$ on $\mathcal{X}$ up to error $\varepsilon$ if, for any distribution $\mathcal{D}_{\text{joint}}$ on $\mathcal{X} \times \{\pm 1\}$, the following hold*

1. *(Soundness). If samples drawn from $\mathcal{D}_{\text{joint}}$ are accepted by the tester $\mathcal{T}$ with high probability, then the learner $\mathcal{A}$ must agnostically learn $\mathcal{F}$ w.r.t. $\mathcal{D}_{\text{joint}}$ up to error $\varepsilon$.*

2. *(Completeness). If the marginal of $\mathcal{D}_{\text{joint}}$ on $\mathcal{X}$ is equal to $\mathcal{D}$, then the tester must accept samples drawn from $\mathcal{D}_{\text{joint}}$ with high probability.*

*Soundness* tells us that whenever a testable learning algorithm outputs a function $\hat{f}$, this function achieves low error (regardless of whether $\mathcal{D}_{\text{joint}}$ satisfies any distributional assumption). On the other hand, *completeness* tells us testable learners are no weaker than (distribution-specific) agnostic ones, in the sense that they achieve the same error whenever $\mathcal{D}_{\text{joint}}$ actually satisfies our assumptions (i.e., whenever this error can in fact be guaranteed for the agnostic learner). The testable model is thus substantially stronger than the agnostic model with distributional assumptions.

**Which function classes can be learned testably?** A natural question is whether testable learning comes at an additional computational cost compared to (distribution-specific) agnostic learning. We focus on the setting where $\mathcal{D}$ is the standard Gaussian on $\mathcal{X} = \mathbb{R}^n$. Following [28, 15], we consider the following simple tester: Accept if and only if the empirical moments up to degree $k$ of the input distribution (approximately) match those of $\mathcal{D}$. This tester satisfies completeness as the

---

[1]In particular, achieving a runtime in $\text{poly}(n, 1/\varepsilon)$ is likely not possible. We note that such runtimes can be achieved in the weaker model where one accepts a loss of $O(\text{opt}) + \varepsilon$ (vs. $\text{opt} + \varepsilon$ in the model we consider) [1].

empirical moments of a Gaussian concentrate well. Using this tester, Rubinfeld & Vasilyan [28] show that halfspaces can be testably learned in time $n^{\tilde{O}(1/\varepsilon^4)}$. Their runtime guarantee was improved to $n^{\tilde{O}(1/\varepsilon^2)}$ in [15], (nearly) matching the best known non-testable algorithm. This shows that there is no separation between the two models for halfspaces. On the other hand, a separation does exist for more complex function classes. Namely, for fixed accuracy $\varepsilon > 0$, testably learning the class of indicator functions of convex sets requires at least $2^{\Omega(n)}$ samples (and hence also time) [28], whereas agnostically learning them only takes subexponential time $2^{O(\sqrt{n})}$, see [25]. The relation between agnostic and testable learning is thus non-trivial, depending strongly on the concept class considered.

## 1.2 Our contributions

In this work, we continue to explore testable learning and its relation to the agnostic model. We consider the concept class of polynomial threshold functions (short PTFs). A degree-$d$ PTF is a function of the form $f(x) = \text{sign}(p(x))$, where $p$ is a polynomial of degree at most $d$. PTFs naturally generalize halfspaces, which correspond to the case $d = 1$. They form an expressive function class with applications throughout (theoretical) computer science, and have been studied in the context of circuit complexity [5, 26, 27, 3], and learning [18, 14]. Despite their expressiveness, PTFs can be agnostically learned in time $n^{O(d^2/\varepsilon^4)}$ [22], which is polynomial in $n$ for any fixed degree $d \in \mathbb{N}$ and error $\varepsilon > 0$. They are thus significantly easier to learn in the agnostic model than convex sets. Our main result is that PTFs can be learned efficiently in the testable model as well.

**Theorem 3** (Informal version of Theorem 19). *Fix $d \in \mathbb{N}$. Then, for any $\varepsilon > 0$, the concept class of degree-$d$ polynomial threshold functions can be testably learned up to error $\varepsilon$ w.r.t. the standard Gaussian in time and sample complexity $n^{\text{poly}(1/\varepsilon)}$.*

Theorem 3 is the first result achieving efficient testable learning for PTFs of any fixed degree $d$ (up to constant error $\varepsilon > 0$). Previously, such a result was not even available for learning degree-2 PTFs with respect to the Gaussian distribution. It also sheds new light on the relation between agnostic and testable learning: there is no *qualitative* computational gap between the two models for the concept class of PTFs, whose complexity lies between that of halfspaces and convex sets in the agnostic model.

In addition to Theorem 3, we also show an impossibility result ruling out a certain natural approach to prove testable learning guarantees for PTFs. In particular, we show in Section 2.4 that an approach which has been successful for testably learning halfspaces in [28] provably cannot work for PTFs.

**Limitations.** The dependence of the running time on the degree parameter $d$ and the error $\varepsilon$ is (much) worse than in the agnostic model (see Theorem 19). Moreover, we do not have access to *lower bounds* on the complexity of testably learning PTFs which might indicate whether these dependencies are inherent to the problem, or an artifact of our analysis. The only lower bounds available apply already in the agnostic model, and show that the time complexity of agnostically (and thus also testably) learning degree-$d$ PTFs is at least $n^{\Omega(d^2/\varepsilon^2)}$ in the SQ-model [12], and at least $n^{\tilde{\Omega}(d^{2-\beta}/\varepsilon^{2-\beta})}$ for any $\beta > 0$ under a cryptographic hardness assumption [29].

## 1.3 Previous work

The two works most closely related to this paper are [28, 15]. Both rely on the following high-level strategy. A standard result [20] shows that one can *agnostically* learn a concept class $\mathcal{F}$ w.r.t. a distribution $\mathcal{D}$ in time $n^{O(k)}$ if all elements of $\mathcal{F}$ are well-approximated w.r.t. $\mathcal{D}$ by degree-$k$ polynomials. That is, if for all $f \in \mathcal{F}$, there exists a degree-$k$ polynomial $h$ such that $\mathbb{E}_{X \sim \mathcal{D}} \left[ |h(X) - f(X)| \right] \leq \varepsilon$. This result can be extended to the testable setting, but now one needs a good low-degree $L_1$-approximation w.r.t. *any* distribution $\mathcal{D}'$ accepted by the tester. Using the moment-matching tester outlined above, one thus needs to exhibit low-degree approximations to all functions in $\mathcal{F}$ w.r.t. any distribution which approximately matches the first few moments of $\mathcal{D}$.

**A direct approach.** In [28], the authors use a direct approach to show that if $\mathcal{F} = \mathcal{F}_{\text{HS}}$ is the class of halfspaces and $\mathcal{D} = \mathcal{N}(0, I_n)$, these approximators exist for $k = O(1/\varepsilon^4)$, leading to an overall running time of $n^{O(1/\varepsilon^4)}$ for their testable learner. Their approach consists of two steps. First, they construct a low-degree approximation $q \approx \text{sign}$ of the sign function in one dimension using standard

techniques. Then, for any halfspace $f(x) = \text{sign}(\langle v, x \rangle - \theta)$, they set $h(x) = q(\langle v, x \rangle - \theta)$. By exploiting *concentration* and *anti-concentration* properties of the *push-forward* under linear functions of distributions that match the moments of a Gaussian, they show that $h$ is a good approximation of $f$. Unfortunately, this kind of approach cannot work for PTFs: We formally rule it out in Theorem 16. This is the aforementioned secondary contribution of our paper, which extends earlier impossibility results for (agnostic) learning of Bun & Steinke [6]. See Section 2.4 for details.

**An indirect approach using fooling.** In order to prove our main theorem we thus need a different approach. Gollakota, Klivans & Kothari [15] establish a connection between testable learning and the notion of *fooling*, which has played an important role in the study of pseudorandomness [4, 2, 9]. Its connection to learning theory had previously been observed in [23]. We say a distribution $\mathcal{D}'$ fools a concept class $\mathcal{F}$ up to error $\varepsilon > 0$ with respect to $\mathcal{D}$ if, for all $f \in \mathcal{F}$, it holds that $|\mathbb{E}_{X \sim \mathcal{D}}[f(X)] - \mathbb{E}_{X \sim \mathcal{D}'}[f(X)]| \leq \varepsilon$. Roughly speaking, the work [15] shows that, if any distribution $\mathcal{D}'$ which approximately matches the moments of $\mathcal{D}$ up to degree $k$ fools $\mathcal{F}$ with respect to $\mathcal{D}$, then $\mathcal{F}$ can be testably learned in time $n^{O(k)}$ (see Theorem 9 below). We remark that (approximately) moment-matching distributions have not been considered much in the existing literature on fooling. Rather, it has focused on distributions $\mathcal{D}'$ whose marginals on any subset of $k$ variables are equal to those of $\mathcal{D}$, which is a stronger condition a priori. While it coincides with moment-matching in special cases (e.g., when $\mathcal{D}$ is the uniform distribution over the hypercube), it does not when $\mathcal{D} = \mathcal{N}(0, I_n)$. Nevertheless, the authors of [15] show that (approximate) moment matching up to degree $k = \tilde{O}(1/\varepsilon^2)$ fools halfspaces with respect to $\mathcal{N}(0, I_n)$, allowing them to obtain the aforementioned result for testably learning $\mathcal{F}_{\text{HS}}$. In fact, they show that this continues to hold when $\mathcal{F}$ consists of arbitrary boolean functions applied to a constant number of halfspaces. They also use existing results in the fooling literature to show that degree-2 PTFs can be testably learned under the uniform distribution over $\{0, 1\}^n$ (but these do not extend to learning over $\mathbb{R}^n$ w.r.t. a standard Gaussian).

**Other previous work on testable learning.** In weaker error models than the agnostic model or under less stringent requirements on the error of the learner it is known how to construct tester-learner pairs with runtime $\text{poly}(n, 1/\varepsilon)$ [16, 10]. These results have been extended to allow for the following stronger completeness condition: The tester has to accept, whenever $\mathcal{D}_{\text{joint}}$ is an isotropic strongly log-concave distribution [17].

## 2 Technical overview

### 2.1 Preliminaries

From here, we restrict to the setting $\mathcal{X} = \mathbb{R}^n$. We let $\mathcal{D}$ be a well-behaved distribution on $\mathbb{R}^n$; usually $\mathcal{D} = \mathcal{N}(0, I_n)$ is the standard Gaussian. For $x \in \mathbb{R}^n$ and a multi-index $\alpha \in \mathbb{N}^n$, we write $x^\alpha := \prod_{i=1}^n x_i^{\alpha_i}$. For $k \in \mathbb{N}$, we write $\mathbb{N}_k^n := \{\alpha \in \mathbb{N}^n, \sum_{i=1}^n \alpha_i \leq k\}$. We say a statement holds 'with high probability' if it holds with probability $\geq 0.99$. The notation $O_d$ (resp. $\Omega_d$, $\Theta_d$) hides factors that only depend on $d$. We now define the moment-matching tester introduced above.

**Definition 4** (Moment matching). *Let $k \in \mathbb{N}$ and $\eta \geq 0$. We say a distribution $\mathcal{D}'$ on $\mathbb{R}^n$ approximately moment-matches $\mathcal{D}$ up to degree $k$ and with slack $\eta$ if*

$$|\mathbb{E}_{X \sim \mathcal{D}}[X^\alpha] - \mathbb{E}_{X \sim \mathcal{D}'}[X^\alpha]| \leq \eta \quad \forall \alpha \in \mathbb{N}_k^n.$$

**Definition 5.** *Let $k \in \mathbb{N}$ and $\eta \geq 0$. The approximate moment-matching tester $\mathcal{T}_{\text{AMM}} = \mathcal{T}_{\text{AMM}}(k, \eta)$ for a distribution $\mathcal{D}$ accepts the samples $(x^{(1)}, z^{(1)}), \ldots, (x^{(m)}, z^{(m)}) \in \mathbb{R}^n \times \{\pm 1\}$ if, and only if,*

$$\left| \mathbb{E}_{X \sim \mathcal{D}}[X^\alpha] - \frac{1}{m} \sum_{i=1}^m (x^{(i)})^\alpha \right| \leq \eta \quad \forall \alpha \in \mathbb{N}_k^n.$$

*That is, $\mathcal{T}_{\text{AMM}}(k, \eta)$ accepts if and only if the moments of the empirical distribution belonging to the samples $\{x^{(i)}\}$ match the moments of $\mathcal{D}$ up to degree $k$ and slack $\eta$. Note that $\mathcal{T}_{\text{AMM}}(k, \eta)$ requires time at most $O(m \cdot n^k)$ to decide whether to accept a set of $m$ samples.*

The tester $\mathcal{T}_{\text{AMM}}$ does not take the labels of the samples into account. In general, for testers $\mathcal{T}$ which depend only on the marginal $\mathcal{D}$ of $\mathcal{D}_{\text{joint}}$ on $\mathbb{R}^n$, we say that $\mathcal{T}$ accepts a distribution $\mathcal{D}'$ on $\mathbb{R}^n$ if it accepts samples drawn from $\mathcal{D}'$ with high probability (regardless of the labels).

## 2.2 Review of existing techniques for testable learning

In this section, we review in more detail the existing techniques to establish guarantees for agnostic and testable learning discussed in Section 1.3. Our goals are twofold. First, we wish to highlight the technical difficulties that arise from proving error guarantees in the testable model versus the agnostic model. Second, we want to introduce the necessary prerequisites for our proof of Theorem 3 in Section 2.3, namely testable learning via *fooling* (see Theorem 9).

**Learning and polynomial approximation.** A standard result [20] shows that one can agnostically learn any concept class that is well-approximated by low-degree polynomials in the following sense.

**Theorem 6** ([20]). *Let $k \in \mathbb{N}$ and $\varepsilon > 0$. Suppose that, for any $f \in \mathcal{F}$, there exists a polynomial $h$ of degree $k$ such that*

$$\mathbb{E}_{X \sim \mathcal{D}} [|h(X) - f(X)|] \leq \varepsilon.$$

*Then, $\mathcal{F}$ can be agnostically learned up to error $\varepsilon$ in time and sample complexity $n^{O(k)}/\mathrm{poly}(\varepsilon)$.*

The underlying algorithm in the theorem above is polynomial regression w.r.t. the absolute loss function. In the testable setting, a similar result holds. The key difference is that one now needs good approximation w.r.t. *all* distributions accepted by the proposed tester.

**Theorem 7.** *Let $k \in \mathbb{N}$ and $\varepsilon > 0$. Let $\mathcal{T}$ be a tester which accepts $\mathcal{D}$ and which requires time and sample complexity $\tau$. Suppose that, for any $f \in \mathcal{F}$, and for any $\mathcal{D}'$ accepted by $\mathcal{T}$, there exists a polynomial $h$ of degree $k$ such that*

$$\mathbb{E}_{X \sim \mathcal{D}'} [|h(X) - f(X)|] \leq \varepsilon.$$

*Then, $\mathcal{F}$ can be testably learned up to error $\varepsilon$ in time and sample complexity $\tau + n^{O(k)}/\mathrm{poly}(\varepsilon)$.*

The takeaway is that, in order to devise efficient algorithms for agnostic or testable learning, it suffices to study low-degree polynomial approximations of elements of $\mathcal{F}$. Under the assumption that $\mathcal{D}$ is a (standard) Gaussian, one has access to powerful techniques from Fourier analysis to show existence of good polynomial approximators w.r.t. $\mathcal{D}$ for various concept classes. Using Theorem 6, this leads to efficient agnostic learning algorithms for a variety of concept classes w.r.t. $\mathcal{N}(0, I_n)$ [20, 25, 22].

**Testable learning via direct approximation.** In the testable setting, it is not sufficient to approximate with respect to $\mathcal{D}$ alone, and so one cannot rely directly on any of its special structure. In [28], the authors overcome this obstacle to get testable learning guarantees for halfspaces w.r.t. $\mathcal{D} = \mathcal{N}(0, I_n)$ by appealing to more basic properties of the distributions $\mathcal{D}'$ accepted by their tester. Their approach is roughly as follows. First, they use standard results from polynomial approximation theory to find a (univariate) polynomial $q$ which approximates the sign-function well on the interval $[-1, 1]$. For a halfspace $f(x) = \mathrm{sign}(\langle v, x \rangle - \theta)$, they consider the approximator $h(x) = q(\langle v, x \rangle - \theta)$, which satisfies

$$\mathbb{E}_{X \sim \mathcal{D}'} [|h(X) - f(X)|] = \mathbb{E}_{Y \sim \mathcal{D}'_{v,\theta}} [|q(Y) - \mathrm{sign}(Y)|].$$

Here, $\mathcal{D}'_{v,\theta}$ is the (shifted) projection of $\mathcal{D}'$ onto the line $\mathrm{span}(v) \subseteq \mathbb{R}^n$. That is, $Y = \langle v, X \rangle - \theta$. Then, for carefully chosen $k \in \mathbb{N}$ and $\eta > 0$, they show that for any $\mathcal{D}'$ accepted by $\mathcal{T}_{\mathrm{AMM}}(k, \eta)$, the distribution $\mathcal{D}'_{v,\theta}$ satisfies certain *concentration* and *anti-concentration* properties, meaning essentially that $\mathcal{D}'_{v,\theta}$ is distributed somewhat uniformly on $[-1, 1]$. As $q$ approximates the sign-function on $[-1, 1]$, they may conclude that $\mathbb{E}_{Y \sim \mathcal{D}'_{v,\theta}} [|q(Y) - \mathrm{sign}(Y)|]$ is small, and invoke Theorem 7.

**Testable learning via fooling.** It is natural to attempt a generalization of the approach above to PTFs. Indeed, for $f(x) = \mathrm{sign}(p(x))$, one could consider the approximator $h(x) = q(p(x))$. However, as we show below in Section 2.4, this approach cannot work when $\deg(p) \geq 6$. Instead, we will rely on a more indirect technique, proposed in [15]. It connects the well-studied notion of *fooling* to low-degree polynomial approximation, and to testable learning.

**Definition 8** (Fooling). *Let $\varepsilon > 0$. We say a distribution $\mathcal{D}'$ on $\mathbb{R}^n$ fools $\mathcal{F}$ w.r.t. $\mathcal{D}$ up to error $\varepsilon$, if, for all $f \in \mathcal{F}$, we have $|\mathbb{E}_{Y \sim \mathcal{D}} [f(Y)] - \mathbb{E}_{X \sim \mathcal{D}'} [f(X)]| \leq \varepsilon$.*

The main result of [15] shows that fooling implies testable learning when using approximate moment-matching to test the distributional assumptions. It forms the basis of our proof of Theorem 3.

**Theorem 9** ([15, Theorem 4.5]). *Let $k, m \in \mathbb{N}$ and $\varepsilon, \eta > 0$. Suppose that the following hold:*

1. *Any distribution $\mathcal{D}'$ whose moments up to degree $k$ match those of $\mathcal{D}$ with slack $\eta$ fools $\mathcal{F}$ w.r.t. $\mathcal{D}$ up to error $\varepsilon/2$.*

2. *With high probability over $m$ samples from $\mathcal{D}$ the empirical distribution matches moments of degree at most $k$ with $\mathcal{D}$ up to slack $\eta$.*

*Then, using the moment-matching tester $\mathcal{T} = \mathcal{T}_{\mathrm{AMM}}(k, \eta)$, we can learn $\mathcal{F}$ testably with respect to $\mathcal{D}$ up to error $\varepsilon$ in time and sample complexity $m + n^{O(k)}$.*

**Remark 10.** When $\mathcal{D} = \mathcal{N}(0, I_n)$ is the standard Gaussian, then the second condition in Theorem 9 is satisfied for $m = \Theta\big((2kn)^k \cdot \eta^{-2}\big)$, see also Fact 36.

The primary technical argument in the proof of Theorem 9 in [15] is an equivalence between fooling and a type of low-degree polynomial approximation called *sandwiching*. Compared to Theorem 7, the advantage of sandwiching is that one needs to approximate $f$ only w.r.t. $\mathcal{D}$ (rather than any distribution accepted by the tester). However, one needs to find not one, but two low degree approximators $h_1, h_2$ that satisfy $h_1 \leq f \leq h_2$ pointwise (i.e., 'sandwich' $f$). We refer to [15] for details.

**Fooling PTFs.** In light of Theorem 9 and Remark 10, in order to prove our main result Theorem 3, it suffices to show that distributions $\mathcal{D}'$ which approximately match the moments of $\mathcal{N}(0, I_n)$ fool the concept class of PTFs. This is our primary technical contribution (see Proposition 12 below). It can be viewed as a generalization of the following result due to Kane [21].

**Theorem 11** (Informal version of [21, Theorem 1]). *Let $\mathcal{D}'$ be a $k$-independent standard Gaussian, meaning the restriction of $\mathcal{D}'$ to any subset of $k$ variables has distribution $\mathcal{N}(0, I_k)$. Then, $\mathcal{D}'$ fools degree-$d$ PTFs w.r.t. $\mathcal{N}(0, I_n)$ up to error $\varepsilon > 0$ as long as $k = k(d, \varepsilon)$ is large enough.*

Theorem 11 applies to a class of distributions that is (far) more restrictive than what we need. First, note that $k$-independent Gaussians match the moments of $\mathcal{N}(0, I_n)$ up to degree $k$ exactly, whereas we must allow $\mathcal{D}'$ whose moments *match only approximately*. Second, even if $\mathcal{D}'$ would match the moments of a Gaussian exactly up to degree $k$, its $k$-dimensional marginals need not be Gaussian. In fact, we have *no information on its moments of high degree* even if they depend on at most $k$ variables. These two distinctions cause substantial technical difficulties in our proof of Proposition 12 below.

## 2.3 Overview of the proof of Theorem 3: testably learning PTFs

As we have seen, in order to prove Theorem 3, it suffices to show that approximately moment-matching distributions fool PTFs. We obtain the following.

**Proposition 12.** *Let $\varepsilon > 0$. Suppose that $\mathcal{D}'$ approximately matches the moments of $\mathcal{N}(0, I_n)$ up to degree $k$ and slack $\eta$, where $k \geq \Omega_d\big(\varepsilon^{-4d \cdot 7^d}\big)$, and $\eta \leq n^{-\Omega_d(k)} k^{-\Omega_d(k)}$. Then, $\mathcal{D}'$ fools the class of degree-$d$ PTFs w.r.t. $\mathcal{N}(0, I_n)$ up to error $\varepsilon/2$. That is, for any $f \in \mathcal{F}_{\mathrm{PTF}, d}$, we then have*

$$\left| \mathbb{E}_{Y \sim \mathcal{N}(0, I_n)} \left[ f(Y) \right] - \mathbb{E}_{X \sim \mathcal{D}'} \left[ f(X) \right] \right| \leq \varepsilon/2. \tag{1}$$

In the rest of this section, we outline how to obtain Proposition 12. Full details can be found in Appendix A. Structurally, our proof is similar to the proof of Theorem 11 in [21]: First, in Section 2.3.1, we show fooling for the subclass of PTFs defined by *multilinear* polynomials. Then, in Section 2.3.2, we extend this result to general PTFs by relating arbitrary polynomials to multilinear polynomials in a larger number of variables. Our primary contribution is thus to show that the construction of [21] (which considers $k$-independent Gaussians) remains valid for the larger class of distributions that approximately match the moments of a Gaussian.

### 2.3.1 Fooling multilinear PTFs

Let $f(x) = \mathrm{sign}(p(x))$, where $p$ is a multilinear polynomial. Our goal is to establish (1) for $f$ under the assumptions of Proposition 12. We will follow the proof of Kane [21] for Theorem 11, which proceeds as follows. Let $\mathcal{D}'$ be a $k$-independent Gaussian. First, Kane constructs a degree-$k$ polynomial approximation $h$ of $f$, satisfying

$$\mathbb{E}_{Y \sim \mathcal{N}(0, I_n)} \left[ h(Y) \right] \approx \mathbb{E}_{Y \sim \mathcal{N}(0, I_n)} \left[ f(Y) \right], \quad \text{and,} \tag{2}$$

$$\mathbb{E}_{X \sim \mathcal{D}'} \left[ h(X) \right] \approx \mathbb{E}_{X \sim \mathcal{D}'} \left[ f(X) \right]. \tag{3}$$

Since the moments of $\mathcal{D}'$ are exactly equal to those of $\mathcal{N}(0, I_n)$ up to degree $k$, we have $\mathbb{E}_{Y \sim \mathcal{N}(0,I_n)}[h(Y)] = \mathbb{E}_{X \sim \mathcal{D}'}[h(X)]$. We may then conclude the fooling property for $\mathcal{D}'$ (cf. (1)):

$$\mathbb{E}_{Y \sim \mathcal{N}(0,I_n)}[f(Y)] \approx \mathbb{E}_{Y \sim \mathcal{N}(0,I_n)}[h(Y)] = \mathbb{E}_{X \sim \mathcal{D}'}[h(X)] \approx \mathbb{E}_{X \sim \mathcal{D}'}[f(X)].$$

As we see below, Kane relies on a *structure theorem* (see Lemma 21) for multilinear polynomials to construct his low-degree approximation $h$. We wish to extend this proof to our setting, where $\mathcal{D}'$ merely matches the moments of the standard Gaussian up to degree $k$ and slack $\eta$. As we will see, the construction of the polynomial $h$ remains valid (although some care is required in bounding the approximation error). A more serious concern is that, for us, $\mathbb{E}_{Y \sim \mathcal{N}(0,I_n)}[h(Y)] \neq \mathbb{E}_{X \sim \mathcal{D}'}[h(X)]$ in general. Our main technical contribution in this section is dealing with the additional error terms that arise from this fact.

**Constructing a low-degree approximation.** We now give details on the construction of the low-degree approximation $h$ that we use in our proof, which is the same as in [21]. The starting point of the construction is a structure theorem for multilinear polynomials $p$ (see Lemma 21 below). It tells us that $f = \text{sign}(p)$ can be decomposed as $f(x) = F(P(x))$, where $F$ is again a PTF, and $P = (P_i)$ is a vector of multilinear polynomials of degree $d$, whose moments $\mathbb{E}_{Y \sim \mathcal{N}(0,I_n)}[P_i(Y)^\ell]$ are all at most $O_d(\sqrt{\ell})^\ell$. Note that these bounds are much stronger than what we would get from standard arguments (which would only yield $\mathbb{E}_{Y \sim \mathcal{N}(0,I_n)}[P_i(Y)^\ell] \leq O_d(\sqrt{\ell})^{d\ell}$). As in [21], we approximate $F$ by a smooth function $\tilde{F}$ via mollification (see Appendix A.1.1). That is, $\tilde{F}$ is the convolution $F * \rho$ of $F$ with a carefully chosen smooth function $\rho$. Then, we set $h(x) = T(P(x))$, where $T$ is the Taylor approximation of $\tilde{F}$ of appropriate degree (see Appendix A.1.2). Intuitively, taking the Taylor expansion yields a good approximation as the (Gaussian) moments of the $P_i$ are not too large, yielding (2), (3).

**Error analysis of the approximation.** Our goal is now to establish (2), (3) in our setting. Note that, since (2) is only concerned with the Gaussian distribution, there is no difference with [21]. For (3), we have to generalize the proof in [21]. For this, we first bound the probability under $\mathcal{D}'$ that (at least) one of the $P_i$ is large, which we do using Markov's inequality. Then, we need to show a bound on the moments of $P_i$ under $\mathcal{D}'$ (recall that the structure theorem only gives a bound on the Gaussian moments). Using bounds on the coefficients of the $P_i$, we are able to do this under a mild condition on $\eta$ (see Appendices A.1.2 and A.1.3).

**Controlling the additional error terms.** To conclude the argument, we need to show that, for our low-degree approximation $h$, we have $\mathbb{E}_{Y \sim \mathcal{N}(0,I_n)}[h(Y)] \approx \mathbb{E}_{X \sim \mathcal{D}'}[h(X)]$. Recall that in [21], these expectations were simply equal. The main issue lies in the fact that, in our setting, the moment matching is only approximate; equality would still hold if $\mathcal{D}'$ matched the moments of $\mathcal{N}(0, I_n)$ up to degree $k$ exactly. Under $\eta$-*approximate* moment matching, we could say that

$$\left| \mathbb{E}_{Y \sim \mathcal{N}(0,I_n)}[h(Y)] - \mathbb{E}_{X \sim \mathcal{D}'}[h(X)] \right| \leq \eta \cdot \|h\|_1, \tag{4}$$

where $\|h\|_1$ is the 1-norm of the coefficients of $h$. However, there is no way to control this norm directly. Instead, we rely on the fact that $h = T \circ P$ and argue as follows. On the one hand, we show a bound on the coefficients in the Taylor approximation $T$ of $\tilde{F}$. On the other hand, we show bounds on all terms of the form $\left| \mathbb{E}_{Y \sim \mathcal{N}(0,I_n)}[P(Y)^\alpha] - \mathbb{E}_{X \sim \mathcal{D}'}[P(X)^\alpha] \right|$. Combining these bounds yields an estimate on the difference $\left| \mathbb{E}_{Y \sim \mathcal{N}(0,I_n)}[h(Y)] - \mathbb{E}_{X \sim \mathcal{D}'}[h(X)] \right|$, which lets us conclude (1).

Going into more detail, the LHS of (4) is equal to the inner product $|\langle t, u \rangle|$ between the vector $t = (t_\alpha)$ of coefficients of $T$ and the vector $u = (u_\alpha)$, where $u_\alpha = \mathbb{E}_{Y \sim \mathcal{N}(0,I_n)}[P(Y)^\alpha] - \mathbb{E}_{X \sim \mathcal{D}'}[P(X)^\alpha]$. This can be viewed as a 'change of basis' $x \to P(x)$. Then, (4) can bounded by $\|u\|_\infty \cdot \|t\|_1$, where $\|u\|_\infty = \max_\alpha |u_\alpha|$. The coefficients $t_\alpha$ of $T$ are related directly to the partial derivatives of $\tilde{F}$, which in turn depend on the function $\rho$ used in the mollification. After careful inspection of this function, we can bound $\|t\|_1 \leq k^{O_d(k)}$ (see Lemma 27). Finally, for any $|\alpha| \leq k$, it holds that

$$\left| \mathbb{E}_{Y \sim \mathcal{N}(0,I_n)}[P(Y)^\alpha] - \mathbb{E}_{X \sim \mathcal{D}'}[P(X)^\alpha] \right| \leq \eta \cdot \sup_i \left( \|P_i\|_1 \right)^{|\alpha|} \leq \eta \cdot n^{\frac{|\alpha| \cdot d}{2}} \leq \eta \cdot n^{O_d(k)},$$

see Fact 22. Putting things together, we get that

$$\left| \mathbb{E}_{Y \sim \mathcal{N}(0,I_n)}[h(Y)] - \mathbb{E}_{X \sim \mathcal{D}'}[h(X)] \right| \leq \|u\|_\infty \cdot \|t\|_1 \leq \eta \cdot k^{O_d(k)} n^{O_d(k)} \ll \varepsilon/2,$$

using the fact that $\eta \leq n^{-\Omega_d(k)} k^{-\Omega_d(k)}$ and $k \gg 1/\varepsilon$ for the last inequality.

### 2.3.2 Fooling arbitrary PTFs

Now, let $f(x) = \text{sign}(p(x))$ be an arbitrary PTF. As before, we want to establish (1). Following [21], the idea is to reduce this problem to the multilinear case as follows. Let $Y \sim \mathcal{N}(0, I_n)$ and let $X$ be a random variable that matches the moments of $Y$ up to degree $k$ and with slack $\eta$. For $N \in \mathbb{N}$ to be chosen later, we construct new random variables $\hat{X}$ and $\hat{Y}$, and a *multilinear* PTF $\hat{f} = \text{sign}(\hat{p})$, all in $n \cdot N$ variables, such that $\hat{Y} \sim \mathcal{N}(0, I_{nN})$, $\hat{X}$ matches moments of $\hat{Y}$ up to degree $k$ with slack $\hat{\eta}$, and

$$\left| \mathbb{E}_Y \left[ f(Y) \right] - \mathbb{E}_X \left[ f(X) \right] \right| \approx \left| \mathbb{E}_{\hat{Y}} \left[ \hat{f}(\hat{Y}) \right] - \mathbb{E}_{\hat{X}} \left[ \hat{f}(\hat{X}) \right] \right|. \tag{5}$$

Assuming $\hat{\eta}$ is not much bigger than $\eta$, and the approximation above is sufficiently good, we may then apply the result of Section 2.3.1 to $\hat{f}$ to conclude (1) for $f$. Our construction of $\hat{X}, \hat{Y}$ and $\hat{f}$ will be the same as in [21]. However, with respect to his proof, we face two difficulties. First, we need to control the slack parameter $\hat{\eta}$ in terms $\eta$. More seriously, Kane's proof of (5) breaks in our setting: He relies on the fact that $X$ is $k$-independent Gaussian in his setting to bound *high degree* moments of $\hat{X}$ which depend on at most $k$ variables. In our setting, we have *no information* on such moments at all (even if $X$ matched the moments of $\mathcal{N}(0, I_n)$ up to degree $k$ exactly).

**Construction of $\hat{X}$ and $\hat{Y}$.** For $i \in [n]$, let $Z^{(i)}$ be an $N$-dimensional Gaussian random variable with mean 0, variances $1 - 1/N$ and covariances $-1/N$, independent from $X$ and all other $Z^{(i')}$. We define $\hat{X}_{ij} \coloneqq X_i/\sqrt{N} + Z_j^{(i)}$, and set $\hat{X} = (\hat{X}_{ij})$. We define $\hat{Y}$ analogously. This ensures that $\hat{Y} \sim \mathcal{N}(0, I_{nN})$. Furthermore, given that $X$ matches the moments of $Y$ with slack $\eta$, it turns out that $\hat{X}$ matches the moments of $\hat{Y}$ with slack $\hat{\eta} = (2k)^{k/2} \cdot \eta$. This follows by direct computation after expanding the moments of $\hat{X}$ in terms of those of $X$ and of the $Z^{(i)}$, see Lemma 32.

**Construction of the multilinear PTF.** We want to construct a *multilinear* polynomial $\hat{p}$ in $nN$ variables so that $p(X) \approx \hat{p}(\hat{X})$. For $\hat{x} \in \mathbb{R}^{nN}$, write $\varphi(\hat{x}) \coloneqq (\sum_{j=1}^N \hat{x}_{ij}/\sqrt{N})_{i \in [n]} \in \mathbb{R}^n$. Since $\varphi(Z^{(i)}) = 0$ holds deterministically, $\varphi(\hat{X}) = X$. So, if we were to set $\hat{p} = p \circ \varphi$, it would satisfy $\hat{p}(\hat{X}) = p(X)$. However, it would clearly not be multilinear. To fix this, we write $p(\varphi(\hat{x})) = \sum_\alpha \lambda_\alpha \hat{x}^\alpha$ and replace each non-multilinear term $\lambda_\alpha \hat{x}^\alpha$ by a multilinear one as follows: If the largest entry of $\alpha$ is at least three, we remove the term completely. If the largest entry of $\alpha$ is two, we replace the term by $\lambda_\alpha \hat{x}^{\alpha'}$, where $\alpha'_{ij} = 1$ if $\alpha_{ij} = 1$ and 0 otherwise. This is identical to the construction in [21]. Now, to show that $p(X) \approx \hat{p}(\hat{X})$ we need to bound the effect of these modifications. It turns out that it suffices to control the following expressions in terms of $N$:

$$a_i \coloneqq \left| \sum_{j=1}^N \frac{\hat{X}_{i,j}}{\sqrt{N}} \right|, \quad b_i \coloneqq \left| \sum_{j=1}^N \left( \frac{\hat{X}_{i,j}}{\sqrt{N}} \right)^2 - 1 \right|, \quad c_{i,\ell} \coloneqq \left| \sum_{j=1}^N \left( \frac{\hat{X}_{i,j}}{\sqrt{N}} \right)^\ell \right| \quad (i \in [n], 3 \le \ell \le d).$$

For the $b_i$ and $c_{i,\ell}$, we can do so using a slight modification of the arguments in [21]. For the $a_i$, however, Kane [21] exploits the fact that in his setting, $X_i$ is standard Gaussian for each fixed $i \in [n]$, meaning the $\hat{X}_{i,j}$ are jointly standard Gaussian over $j$. This gives him access to strong concentration bounds. To get such concentration bounds in our setting, we would need information on the moments of the $X_i$ up to degree roughly $\log n$. However, we only have access to moments up to degree $k$, which is not allowed to depend on $n$ (as our tester uses time $n^{\Omega(k)}$). Instead, we use significantly weaker concentration bounds based on moments of constant degree. By imposing stronger conditions on the $b_i, c_{i,\ell}$, we are able to show that the remainder of the argument in [21] still goes through in our setting, see Appendix A.2.2. Finally, for $N$ sufficiently large, this allows us to conclude (5) for $\hat{f} = \text{sign}(\hat{p})$.

### 2.4 Impossibility result: learning PTFs via the push-forward

In this section, we show that the approach of [28] to prove testable learning guarantees for halfspaces w.r.t. the standard Gaussian cannot be generalized to PTFs. Namely, we show that in general, PTFs $f(x) = \text{sign}(p(x))$ with $\deg(p) \ge 3$ cannot be approximated up to arbitrary error w.r.t. $\mathcal{N}(0, I_n)$ by a polynomial of the form $h(x) = q(p(x))$, regardless of the degree of $q$.[2] Importantly, we show

---

[2]Note that this even excludes proving an *agnostic* learning guarantee w.r.t. $\mathcal{N}(0, I_n)$ using this approach.

that this is the case even if one makes certain typical structural assumptions on $p$ which only change the PTF $f = \text{sign}(p)$ on a set of negligible Gaussian volume; namely that $p$ is square-free and that $\{p \geq 0\} \subseteq \mathbb{R}^n$ is compact. Our main technical contribution is an extension of a well-known inapproximability result due to Bun & Steinke (Theorem 15 below) to distributions 'with a single heavy tail' (see Theorem 18).

**Approximating the sign-function on the real line**   Let $p_{\#}\mathcal{N}(0, I_n)$ be the *push-forward* of the standard Gaussian distribution by $p$, which is defined by

$$\mathbb{P}_{Y \sim p_{\#}\mathcal{N}(0,I_n)} \left[ Y \in A \right] := \mathbb{P}_{X \sim \mathcal{N}(0,I_n)} \left[ X \in p^{-1}(A) \right] \quad (A \subseteq \mathbb{R}). \tag{6}$$

Note that, if $h(x) = q(p(x))$, we then have

$$\mathbb{E}_{X \sim \mathcal{N}(0,I_n)} \left[ |h(X) - f(X)| \right] = \mathbb{E}_{Y \sim p_{\#}\mathcal{N}(0,I_n)} \left[ |q(Y) - \text{sign}(Y)| \right].$$

Finding a good approximator $h \approx f$ of the form $h = q \circ p$ is thus equivalent to finding a (univariate) polynomial $q$ which approximates the sign-function on $\mathbb{R}$ well under the push-forward distribution $p_{\#}\mathcal{N}(0, I_n)$. In light of this observation, we are interested in the following question: Let $\mathcal{D}$ be a distribution on the real line. Is it possible to find for each $\varepsilon > 0$ a polynomial $q$ such that $\mathbb{E}_{Y \sim \mathcal{D}} \left[ |q(Y) - \text{sign}(Y)| \right] \leq \varepsilon$? This question is well-understood for distributions $\mathcal{D}$ whose density is of the form $w_\gamma(x) := C_\gamma \exp(-|x|^\gamma)$, $\gamma > 0$. Namely, when $\gamma \geq 1$, these distributions are *log-concave*, and the question can be answered in the affirmative. On the other hand, when $\gamma < 1$, they are *log-superlinear*, and polynomial approximation of the sign function is not possible.

**Theorem 13** (see, e.g. [20]). *Let $\mathcal{D}$ be a log-concave distribution on $\mathbb{R}$. Then, for any $\varepsilon > 0$ there exists a polynomial $q$ such that $\mathbb{E}_{Y \sim \mathcal{D}} \left[ |q(Y) - \text{sign}(Y)| \right] \leq \varepsilon$.*

**Definition 14.** *Let $\mathcal{D}$ be a distribution on $\mathbb{R}$ whose density function $w$ satisfies*

$$w(x) \geq C \cdot w_\gamma(x) \quad \forall \, x \in \mathbb{R}$$

*for some $\gamma < 1$ and $C > 0$. Then we say $\mathcal{D}$ is* log-superlinear (LSL).

**Theorem 15** (Bun-Steinke [6]). *Let $\mathcal{D}$ be an LSL-distribution on $\mathbb{R}$. Then there exists an $\varepsilon > 0$ such that, for any polynomial $q$, we have $\mathbb{E}_{Y \sim \mathcal{D}} \left[ |q(Y) - \text{sign}(Y)| \right] > \varepsilon$.*

When $p$ is of degree 1 (i.e., when $f = \text{sign}(p)$ defines a halfspace), the push-forward distribution $\mathcal{D} = p_{\#}\mathcal{N}(0, I_n)$ defined in (6) is itself a (shifted) Gaussian. In particular, it is log-concave and by Theorem 13 approximation of the sign-function w.r.t. $\mathcal{D}$ is possible. On the other hand, when $p$ is of higher degree, $\mathcal{D}$ could be an LSL-distribution, meaning approximation of the sign-function w.r.t. $\mathcal{D}$ is not possible by Theorem 15. For instance, consider $p(x) = x^3$. The density $w$ of $p_{\#}\mathcal{N}(0, 1)$ is given by $w(x) = C \cdot |x|^{-2/3} \cdot \exp(-|x|^{2/3})$, and so $p_{\#}\mathcal{N}(0, 1)$ is log-superlinear.

**Choice of description.**   The example $p(x) = x^3$ is artificial: We have $\text{sign}(p(x)) = \text{sign}(x)$, and so the issue is not with the concept $f = \text{sign}(p)$, but rather with our choice of description $p$. In general, one can (and should) assume that $p$ is *square-free*, meaning it is not of the form $p = p_1^2 \cdot p_2$. Indeed, note that for such a polynomial, we have $\text{sign}(p(x)) = \text{sign}(p_2(x))$ almost everywhere. Square-freeness plays an important role in the analysis of learning algorithms for PTFs, see, e.g., [22, Appendix A]. It turns out that even if $p$ is square-free, the distribution $p_{\#}\mathcal{N}(0, I_n)$ can still be log-superlinear, e.g., when $p(x) = x(x-1)(x-2)$. Note that, for this example, $\text{sign}(p)$ describes a non-compact subset of $\mathbb{R}$. This is crucial to find that $p_{\#}\mathcal{N}(0, 1)$ is LSL. Indeed, if $\{p \geq 0\} \subseteq \mathbb{R}$ were compact, then $p_{\max} = \sup_{x \in \mathbb{R}} p(x) < \infty$, and so the density $w$ of $p_{\#}\mathcal{N}(0, 1)$ would satisfy $w(x) = 0$ for all $x > p_{\max}$. In particular, $w$ would not be log-superlinear. One could therefore hope that assuming $\{p \geq 0\}$ is compact might fix our issues. This assumption is reasonable as $\{p \geq 0\}$ can be approximated arbitrarily well by a compact set (in terms of Gaussian volume). On the contrary, we show the following.

**Theorem 16.** *There exists a square-free polynomial $p$, so that $\{p \geq 0\} \subseteq \mathbb{R}$ is compact, but for which there exists $\varepsilon > 0$ so that, for any polynomial $q$, $\mathbb{E}_{X \sim \mathcal{N}(0,1)} \left[ |q(p(X)) - \text{sign}(p(X))| \right] > \varepsilon$.*

To establish Theorem 16, we prove a 'one-sided' analog of Theorem 15 in Appendix B.1, which we believe to be of independent interest. It shows impossibility of approximating the sign-function under a class of distributions related to, but distinct from, those considered by Bun & Steinke [6]. The key difference is that the densities in our result need only to have a single heavy tail. However, this tail must be 'twice as heavy' ($\gamma < 1/2$ vs. $\gamma < 1$). We emphasize that [6] does not cover compact PTFs.

**Definition 17.** *Let $\mathcal{D}$ be a distribution on $\mathbb{R}$ whose density function $w$ satisfies*

$$w(x) \geq C \cdot w_\gamma(x) \quad \forall\, x \in (-\infty, 1]$$

*for some $\gamma < 1/2$ and $C > 0$. Then we say $\mathcal{D}$ is* one-sided *log-superlinear.*

**Theorem 18.** *Let $\mathcal{D}$ be a one-sided LSL-distribution on $\mathbb{R}$. Then there exists an $\varepsilon > 0$ such that, for any polynomial $q$, we have $\mathbb{E}_{Y \sim \mathcal{D}}\left[|q(Y) - \mathrm{sign}(Y)|\right] > \varepsilon$.*

*Proof of Theorem 16.* It suffices to find a square-free polynomial $p$ for which $\{p \geq 0\}$ is compact, and the push-forward distribution $p_\# \mathcal{N}(0, 1)$ is one-sided LSL. A direct computation shows that

$$p(x) \coloneqq -x(x - 1)(x - 2)(x - 3)(x - 4)(x - 5)$$

meets the criteria, see Appendix B.2 for details. $\qquad\square$

## Acknowledgments

We thank the anonymous reviewers for their valuable comments and suggestions. We thank Arsen Vasilyan for helpful discussions. This work is supported by funding from the European Research Council (ERC) under the European Union's Horizon 2020 research and innovation programme (grant agreement No 815464).

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

# A  Testable learning of polynomial threshold functions

In this section, we give a formal proof of our main result, Theorem 3, that we restate here.

**Theorem 19** (Formal version of Theorem 3). *Let $d \in \mathbb{N}$. For any $\varepsilon > 0$, the concept class of degree-$d$ polynomial threshold functions in $n$ variables can be testably learned up to error $\varepsilon$ w.r.t. the standard Gaussian $\mathcal{N}(0, I_n)$ in time and sample complexity*

$$n^{O_d\left(\varepsilon^{-4d \cdot 7^d}\right)} \varepsilon^{-O_d\left(\varepsilon^{-4d \cdot 7^d}\right)},$$

*In particular, if $d$ is constant, the time and sample complexity is $n^{\mathrm{poly}(1/\varepsilon)}$.*

Our goal to show this result is to apply Theorem 9. We first focus on proving the fooling condition in this theorem. Recall that for this we need to show that there are $k$ and $\eta$ such that if $\mathcal{D}'$ matches the moments of $\mathcal{N}(0, I_n)$ up to degree $k$ and slack $\eta$, then we have

$$\left|\mathbb{E}_{X \sim \mathcal{D}'}[f(X)] - \mathbb{E}_{Y \sim \mathcal{N}(0, I_n)}[f(Y)]\right| \le \varepsilon/2.$$

In order to show this, we follow the result of [21] (see Theorem 11). This paper shows this condition for any distribution $\mathcal{D}'$ that is a $k$-independent Gaussian, i.e. for which the marginal of every subset of $k$ coordinates is $\mathcal{N}(0, I_k)$.

The reason why it is enough to only focus on satisfying the fooling condition is that for any $\eta$, we can find $m$ large enough that the first condition of Theorem 9 is satisfied for $\mathcal{N}(0, I_n)$ (see Remark 10 and Fact 36). For $k$ we choose the same value as in [21], namely

$$k = \Theta_d\left(\varepsilon^{-4d \cdot 7^d}\right). \tag{7}$$

We first show how to choose $\eta$ for the case of multilinear polynomial polynomials in Appendix A.1. In Appendix A.2, we generalize the fooling result to arbitrary PTFs. Finally, in Appendix A.3, we show how to apply Theorem 9 to get testable learning for PTFs.

## A.1  Fooling for multlinear PTFs

Thus, let $f \in \mathcal{C}$ and let $p$ be the multilinear polynomial (of degree $d$) such that $f(x) = \mathrm{sign}(p(x))$. Note that this notation is different than the one used in [21]. There, the roles of $f$ and $F$ are interchanged. We use $f$ for the PTF throughout to be consistent with the previous work on (testable) learning. Without loss of generality, we assume that the sum of the squares of the coefficients of $p$ is 1. We can make this assumption since rescaling does not change the PTF.

The main result of this section is the following proposition.

**Proposition 20** (Fooling for multilinear PTFs). *Let $\varepsilon > 0$. Suppose that $\mathcal{D}'$ approximately matches the moments of $\mathcal{N}(0, I_n)$ up to degree $k$ and slack $\eta$, where $k \ge \Omega_d\left(\varepsilon^{-4d \cdot 7^d}\right)$, and $\eta \le n^{-\Omega_d(k)} k^{-\Omega_d(k)}$. Then, we have that, for any multilinear $f \in \mathcal{F}_{\mathrm{PTF}, d}$,*

$$\left|\mathbb{E}_{X \sim \mathcal{D}'}[f(X)] - \mathbb{E}_{Y \sim \mathcal{N}(0, I_n)}[f(Y)]\right| \le O_d(\varepsilon).$$

Note that we only show $O_d(\varepsilon)$ here instead of $\varepsilon/2$, which is needed to apply Theorem 9. This simplifies the notation in our proof of this proposition. We later apply this proposition to $\varepsilon' = \varepsilon/\Omega_d(1)$ to conclude the fooling result we need.

As mentioned earlier, the general strategy to do this is based on [21]. We want to find a function $\tilde{f} : \mathbb{R}^n \to \mathbb{R}$ that approximates $f$. We will define this function in Appendix A.1.1. In Appendix A.1.2, we show that the expectation of $\tilde{f}$ is close under $\mathcal{D}'$ and $\mathcal{N}(0, I_n)$. More precisely, in Lemma 25, we show that under the above assumption on $\eta$, we have that

$$\left|\mathbb{E}_{Y \sim \mathcal{N}(0, I_n)}[\tilde{f}(Y)] - \mathbb{E}_{X \sim \mathcal{D}'}[\tilde{f}(X)]\right| \le O(\varepsilon).$$

In Appendix A.1.3, we then show that under both $\mathcal{D}'$ and $\mathcal{N}(0, I_n)$ the expectation of $f$ and $\tilde{f}$ are close and complete the proof of Proposition 20. More precisely, from [21, Proposition 14] (restated in Lemma 28), we get that

$$\left|\mathbb{E}_{Y \sim \mathcal{N}(0, I_n)}[f(Y)] - \mathbb{E}_{Y \sim \mathcal{N}(0, I_n)}[\tilde{f}(Y)]\right| \le O(\varepsilon).$$

In Appendix A.1.3, we then also show that this also holds for $X \sim \mathcal{D}'$ instead of $Y \sim \mathcal{N}(0, I_n)$. More precisely, we show in Lemma 29 that Lemmas 25 and 28 contain already enough information about the moment-matching distribution $\mathcal{D}'$ to conclude Proposition 20.

### A.1.1  Set up and definition of the function $\tilde{f}$

In this section, we want to define the function $\tilde{f}$ that should be thought of as a smooth approximation of the PTF $f$. In order to define this function, we first restate the following structural theorem from [21].

**Lemma 21** ([21, Proposition 4]). *Let $m_1 \leq m_2 \leq \cdots \leq m_d$ be integers. Then there exists integers $n_1, n_2, \ldots, n_d$, where $n_i \leq O_d(m_1 m_2 \ldots m_{i-1})$ and (non-constant homogeneous multilinear) polynomials $h_1, \ldots, h_d, P_{i,j}$ ($1 \leq i \leq d$, $1 \leq j \leq n_i$) such that:*

1. *The sum of the squares of the coefficients of $P_{i,j}$ is 1.*

2. *If $Y \sim \mathcal{N}(0, I_n)$ and $\ell \leq m_i$, then $\mathbb{E}\left[|P_{i,j}(Y)|^\ell\right] \leq O_d(\sqrt{\ell})^\ell$.*

3. $p(Y) = \sum_{i=1}^d h_i(P_{i,1}(Y), P_{i,2}(Y), \ldots, P_{i,n_i}(Y)).$

The values of $m_i$ we want to choose are as in [21], i.e. we let $m_i = \Theta_d\left(\varepsilon^{-3.7^i d}\right)$. Given this structure theorem, we introduce now the following notation that we use throughout the remainder of this section, analogous to [21]. As before, we use $p : \mathbb{R}^n \to \mathbb{R}$ to be denote the multilinear polynomial and $f : \mathbb{R}^n \to [-1, 1]$ to be the PTF we are interested in, i.e. $f(x) = \text{sign}(p(x))$. Furthermore, the $P_{i,j} : \mathbb{R}^n \to \mathbb{R}$ for $i \in [n]$ and $j \in [n_i]$ are the polynomials in the structure theorem. We denote by $P_i : \mathbb{R}^n \to \mathbb{R}^{n_i}$ for $i \in [n]$ the vector $(P_{i,1}, \ldots, P_{i,n_i})$ and by $P : \mathbb{R}^n \to \mathbb{R}^{n_1} \times \cdots \times \mathbb{R}^{n_d}$ the vector $(P_1, \ldots, P_d)$. Finally, we define $F : \mathbb{R}^{n_1} \times \cdots \times \mathbb{R}^{n_d} \to \mathbb{R}$ as the function $F(y_1, \ldots, y_d) = \sum_{i=1}^d h_i(y_i)$, i.e. we get $f(x) = F(P(x))$.

Similar to Condition 1 in the above lemma, we can also get a bound on the sum of the absolute values of the coefficients of the $P_{i,j}$. We need this later in the proof of Lemma 25.

**Fact 22.** *For any $i \in [n]$ and $j \in [n_i]$, the sum of the absolute values of the coefficients of $P_{i,j}$ is at most $n^{d/2}$.*

The idea to prove this is to bound the number of coefficients we have and use an inequality between the 1- and 2-norm. The detailed argument can be found in Appendix C.1. Furthermore, we have an analogous result to Item 2 for the moment-matching distributions, which is again proved in Appendix C.1. We need this result later for concluding that $\tilde{f}$ is a good approximation under the moment matching distribution $\mathcal{D}'$.

**Fact 23.** *For any $i \in [d]$, $j \in [n_i]$ and $\ell \leq k/d$, we have that*

$$\mathbb{E}_{X \sim \mathcal{D}'}\left[P_{i,j}(X)^\ell\right] \leq \mathbb{E}_{Y \sim \mathcal{N}(0, I_n)}\left[P_{i,j}(Y)^\ell\right] + \eta n^{d\ell/2}.$$

As in [21], we now consider the function $\rho_C : \mathbb{R}^n \to \mathbb{R}$ defined in the following lemma.

**Lemma 24** ([21, Lemma 5]). *Let*

$$B(\xi) = \begin{cases} 1 - \|\xi\|_2^2 & \text{if } \|\xi\|_2 \leq 1 \\ 0 & \text{else} \end{cases} \quad \text{and} \quad \rho_2(x) = \frac{|\hat{B}(x)|^2}{\|B\|_{L^2}^2},$$

*where where $\hat{B}$ is the Fourier transform of $B$. Then, the function $\rho_C$ defined by*

$$\rho_C(x) = \left(\frac{C}{2}\right)^n \rho_2\left(\frac{Cx}{2}\right)$$

*satisfies the following conditions*

1. $\rho_C \geq 0,$

2. $\int_{\mathbb{R}^n} \rho(x)\, \mathrm{d}x = 1,$

3. *for any unit vector $v$ and any non-negative integer $\ell$ we have*

$$\int_{\mathbb{R}^n} |D_v^\ell \rho(C)(x)|\, \mathrm{d}x \leq C^\ell,$$

4. for $D > 0$, $\int_{\|x\|_2 \geq D} |\rho_C(x)| \, dx = O\left(\left(\frac{n}{CD}\right)^2\right)$.

We now define the following three functions $\rho$, $\tilde{F}$ and in particular $\tilde{f}$, in the same way as in [21]. We let $\rho : \mathbb{R}^{n_1} \times \cdots \times \mathbb{R}^{n_d} \to \mathbb{R}$ be defined as $\rho(y_1, \ldots, y_d) = \rho_{C_1}(y_1) \cdots \rho_{C_d}(y_d)$, where the $C_i$ are the same as in [21], i.e. we let $C_i = \Theta_d\left(\varepsilon^{-7^i d}\right)$. Using this function, we define an approximation $\tilde{F} : \mathbb{R}^{n_1} \times \cdots \times \mathbb{R}^{n_d} \to \mathbb{R}$ to $F$ as the convolution $\tilde{F} = F * \rho$ and an approximation $\tilde{f} : \mathbb{R}^n \to \mathbb{R}$ to $f$ as $\tilde{f}(x) = \tilde{F}(P(x))$.

The general strategy to prove Proposition 20 is show the following three steps, where as usual $Y \sim \mathcal{N}(0, I_n)$ and $X \sim \mathcal{D}'$,

$$\mathbb{E}_Y\left[f(Y)\right] \overset{\text{Lem. } 28}{\approx} \mathbb{E}_Y\left[\tilde{f}(Y)\right] \overset{\text{Lem. } 25}{\approx} \mathbb{E}_X\left[\tilde{f}(X)\right] \overset{\text{Lem. } 29}{\approx} \mathbb{E}_X\left[f(X)\right]. \tag{8}$$

### A.1.2 Expectations of $\tilde{f}$ under the Gaussian and the moment-matching distribution are close

In this section, we want to prove the middle approximation of (8). More precisely, we show the following lemma.

**Lemma 25.** *Let $\varepsilon > 0$. Suppose that $\mathcal{D}'$ approximately matches the moments of $\mathcal{N}(0, I_n)$ up to degree $k$ and slack $\eta$, where $k \geq \Omega_d\left(\varepsilon^{-4d \cdot 7^d}\right)$, and $\eta \leq n^{-\Omega_d(k)} k^{-\Omega_d(k)}$. Then we have that*

$$\left|\mathbb{E}_{Y \sim \mathcal{N}(0, I_n)}\left[\tilde{f}(Y)\right] - \mathbb{E}_{X \sim \mathcal{D}'}\left[\tilde{f}(X)\right]\right| \leq O(\varepsilon).$$

We want to do this similar to [21, Section 6]. For this, let $T$ be the Taylor approximation of $\tilde{F}$ around 0. We use degree $m_i$ for the $i$th batch of coordinates (recall that $\tilde{F} : \mathbb{R}^{n_1} \times \cdots \times \mathbb{R}^{n_d} \to \mathbb{R}$). The strategy to show Lemma 25 is to proceed in the following three steps, where again $Y \sim \mathcal{N}(0, I_n)$ and $X \sim \mathcal{D}'$,

$$\mathbb{E}_Y\left[\tilde{f}(Y)\right] \overset{[21]}{\approx} \mathbb{E}_Y\left[T(P(Y))\right] \overset{\text{Pf. of Lem. } 25}{\approx} \mathbb{E}_X\left[T(P(X))\right] \overset{\text{Lem. } 26}{\approx} \mathbb{E}_X\left[\tilde{f}(X)\right]. \tag{9}$$

The first approximation above only involves the Gaussian $Y$, so we get directly from [21, Proof of Proposition 8] that $\mathbb{E}_{Y \sim \mathcal{N}(0, I_n)}\left[|\tilde{f}(Y) - T(P(Y))|\right] \leq O(\varepsilon)$. We now want to extend this also to moment-matching distribution, which we do in the following lemma, i.e. show the third approximation in (9).

**Lemma 26.** *Let $\varepsilon > 0$. Suppose that $\mathcal{D}'$ approximately matches the moments of $\mathcal{N}(0, I_n)$ up to degree $k$ and slack $\eta$, where $k \geq \Omega_d\left(\varepsilon^{-4d \cdot 7^d}\right)$ and $\eta \leq \frac{1}{kd}$. Then, we have that*

$$\mathbb{E}_{X \sim \mathcal{D}'}\left[|\tilde{f}(X) - T(P(X))|\right] \leq O(\varepsilon)$$

This proof follows closely [21, Proof of Proposition 8], which is why we defer the proof to Appendix C.2.

Note that the condition on $\eta$ in this lemma is also satisfied for the $\eta$ from Lemma 25. Thus, it remains to prove

$$\left|\mathbb{E}_{Y \sim \mathcal{N}(0, I_n)}\left[T(P(Y))\right] - \mathbb{E}_{X \sim \mathcal{D}'}\left[T(P(X))\right]\right| \leq O(\varepsilon)$$

to complete the proof of Lemma 25.

Note that in contrast to [21], this quantity is not 0 in our case since we only have *approximate* moment matching. This is the main technical difficulty in our proof for the multilinear case. We need to give a different argument here and argue that this is small even if $X$ is only approximately moment matching a Gaussian and not a $k$-independent Gaussian. We can write the Taylor expansion as follows

$$T(x) = \sum_{\alpha = (\alpha_1, \ldots, \alpha_d): |\alpha_i| \leq m_i} \frac{1}{\alpha!} \partial^\alpha \tilde{F}(0) x^\alpha,$$

where the $\alpha_i$ are multi-indices in $\mathbb{N}^{n_i}$. We want to prove the following lemma about the coefficients in the Taylor expansion.

**Lemma 27.** *For any multi-index $\alpha = (\alpha_1, \ldots, \alpha_d) \in \mathbb{N}^{n_1} \times \cdots \times \mathbb{N}^{n_d}$, we have*

$$\partial^\alpha \tilde{F}(0) \leq \prod_{i=1}^d C_i^{|\alpha_i|}.$$

The proof of this lemma is in Appendix C.2. The ideas of this proof are based on [21, Proof of Lemmas 5 and 7]. We can now prove Lemma 25.

*Proof of Lemma 25.* As argued above, we already know that

$$\mathbb{E}_{Y \sim \mathcal{N}(0, I_n)} \left[|\tilde{f}(Y) - T(P(Y))|\right] \leq O(\varepsilon) \quad \text{and} \quad \mathbb{E}_{X \sim \mathcal{D}'} \left[|\tilde{f}(X) - T(P(X))|\right] \leq O(\varepsilon).$$

It thus remains to argue that

$$\left|\mathbb{E}_{Y \sim \mathcal{N}(0, I_n)} \left[T(P(Y))\right] - \mathbb{E}_{X \sim \mathcal{D}'} \left[T(P(X))\right]\right| \leq O(\varepsilon).$$

Define the vectors $t = (t_\alpha)$ and $u = (u_\alpha)$ by

$$t_\alpha := \frac{1}{\alpha!} \partial^\alpha \tilde{F}(0) \quad \text{and} \quad u_\alpha := \mathbb{E}_{Y \sim \mathcal{N}(0, I_n)} \left[P(Y)^\alpha\right] - \mathbb{E}_{X \sim \mathcal{D}'} \left[P(X)^\alpha\right].$$

We then have that

$$\left|\mathbb{E}_{Y \sim \mathcal{N}(0, I_n)} \left[T(P(Y))\right] - \mathbb{E}_{X \sim \mathcal{D}'} \left[T(P(X))\right]\right| = |\langle t, u \rangle| \leq \|t\|_1 \cdot \|u\|_\infty.$$

We now want to bound $\|t\|_1$ and $\|u\|_\infty$ separately. For the bound on $\|t\|_1$, note that, by Lemma 27, we have

$$|t_\alpha| = \frac{1}{\alpha!} \partial^\alpha \tilde{F}(0) \leq \prod_{i=1}^d C_i^{|\alpha_i|}.$$

Plugging in the definition of $C_i = \Theta_d \left(\varepsilon^{-7^i d}\right) \leq O_d(k)$ and using $|\alpha| = \sum_{i=1}^d |\alpha_i| \leq k$, we get that

$$|t_\alpha| \leq k^{O_d(k)}.$$

Since we have that $n_i \leq k$, we can conclude that the number of multi-indices $\alpha$ with $|\alpha| \leq k$ is at most $(dk)^k \leq k^{O_d(k)}$ and thus, we have that

$$\|t\|_1 = \sum_\alpha |t_\alpha| \leq k^{O_d(k)}.$$

We now move on to bound $\|u\|_\infty$. We write $P_{i,j}$ as follows

$$P_{i,j}(x) = \sum_\beta a_{i,j}^{(\beta)} x^\beta.$$

Here, the $\beta$ in the sum goes over all multi-indices with $|\beta|$ being the degree of $P_{i,j}$, which is at most $d$. By Fact 22, we have that

$$\sum_\beta |a_{i,j}^{(\beta)}| \leq n^{d/2}.$$

Let $|\alpha| = \ell$ be such that $|\alpha_i| \leq m_i$ (i.e. the term appears in the Taylor expansion) and let $(i_1, j_1), \ldots, (i_\ell, j_\ell)$ be such that

$$P(x)^\alpha = P_{i_1, j_1}(x) \ldots P_{i_\ell, j_\ell}(x).$$

Note that if some $(\alpha_i)_j > 1$, we include the corresponding factor $P_{i,j}$ multiple times. We can now expand $P(x)^\alpha$ as follows

$$P(x)^\alpha = \left(\sum_{\beta_1} a_{i_1, j_1}^{(\beta_1)} x^{\beta_1}\right) \cdots \left(\sum_{\beta_\ell} a_{i_\ell, j_\ell}^{(\beta_\ell)} x^{\beta_\ell}\right)$$

$$= \sum_{\beta_1} \cdots \sum_{\beta_\ell} a_{i_1, j_1}^{(\beta_1)} \ldots a_{i_\ell, j_\ell}^{(\beta_\ell)} x^{\beta_1 + \cdots + \beta_\ell}.$$

Now, note that $k \geq d(m_1 + \cdots + m_d)$ (this condition is in addition to the conditions on $k$ in [21] but it does not change the asymptotic value of $k$ as stated in (7)). We get that the degree of the terms appearing in the sum is $|\beta_1| + \cdots + |\beta_\ell| \leq d|\alpha| \leq d(m_1 + \ldots m_d) \leq k$ and thus that

$$|\mathbb{E}_{Y \sim \mathcal{N}(0, I_n)} [P(Y)^\alpha] - \mathbb{E}_{X \sim \mathcal{D}'} [P(X)^\alpha]| \leq \sum_{\beta_1} \cdots \sum_{\beta_\ell} |a_{i_1, j_1}^{(\beta_1)}| \ldots |a_{i_\ell, j_\ell}^{(\beta_\ell)}| \eta.$$

Here, we used the triangle inequality and the fact that

$$|\mathbb{E}_{Y \sim \mathcal{N}(0, I_n)} \left[ Y^{\beta_1 + \cdots + \beta_\ell} \right] - \mathbb{E}_{X \sim \mathcal{D}'} \left[ X^{\beta_1 + \cdots + \beta_\ell} \right]| \leq \eta.$$

Thus, we can compute

$$
\begin{aligned}
|\mathbb{E}_{Y \sim \mathcal{N}(0, I_n)} [P(Y)^\alpha] - \mathbb{E}_{X \sim \mathcal{D}'} [P(X)^\alpha]| &\leq \sum_{\beta_1} \cdots \sum_{\beta_\ell} |a_{i_1, j_1}^{(\beta_1)}| \ldots |a_{i_\ell, j_\ell}^{(\beta_\ell)}| \eta \\
&= \left( \sum_{\beta_1} |a_{i_1, j_1}^{(\beta_1)}| \right) \cdots \left( \sum_{\beta_\ell} |a_{i_\ell, j_\ell}^{(\beta_\ell)}| \right) \eta \\
&= \|a_{i_1, j_1}\|_1 \ldots \|a_{i_\ell, j_\ell}\|_1 \eta \\
&\leq n^{\ell d/2} \eta \\
&= n^{|\alpha| d/2} \eta.
\end{aligned}
$$

Thus, we get

$$\|u\|_\infty \leq n^{O_d(k)} \eta.$$

Finally, we can conclude that

$$\left| \mathbb{E}_{Y \sim \mathcal{N}(0, I_n)} [T(P(Y))] - \mathbb{E}_{X \sim \mathcal{D}'} [T(P(X))] \right| \leq \|t\|_1 \cdot \|u\|_\infty \leq n^{O_d(k)} k^{O_d(k)} \eta \leq O(\varepsilon)$$

since we have the conditions $\eta \leq n^{-\Omega_d(k)} k^{-\Omega_d(k)}$ and $k^{-1} \leq O(\varepsilon)$. $\qquad \square$

### A.1.3 The functions $f$ and $\tilde{f}$ are close in expectation

In this section, we want to complete the proof of Proposition 20 that shows

$$\left| \mathbb{E}_{X \sim \mathcal{D}'} [f(X)] - \mathbb{E}_{Y \sim \mathcal{N}(0, I_n)} [f(Y)] \right| \leq O_d(\varepsilon).$$

So far, we already showed in Lemma 25 that

$$\left| \mathbb{E}_{Y \sim \mathcal{N}(0, I_n)} \left[ \tilde{f}(Y) \right] - \mathbb{E}_{X \sim \mathcal{D}'} \left[ \tilde{f}(X) \right] \right| \leq O(\varepsilon).$$

Thus, it remains to show that under both $X \sim \mathcal{D}'$ and $Y \sim \mathcal{N}(0, I_n)$, the expectation of $f$ and $\tilde{f}$ differ by at most $O_d(\varepsilon)$. For $Y$, we directly get the following. Note that the approximation $\tilde{f}$ depends on $\varepsilon$ via the numbers $m_i$ in the structure theorem Lemma 21 and the $C_i$ in the definition of $\rho$.

**Lemma 28** ([21, Proposition 14]). *Let $\varepsilon > 0$. Then, we have that*

$$\left| \mathbb{E}_{Y \sim \mathcal{N}(0, I_n)} [f(Y)] - \mathbb{E}_{Y \sim \mathcal{N}(0, I_n)} \left[ \tilde{f}(Y) \right] \right| \leq O(\varepsilon).$$

The reason why we get this directly from [21] is that this lemma only concerns the Gaussian and not the moment matching distribution. Combining this with the above, we have now shown that

$$\left| \mathbb{E}_{Y \sim \mathcal{N}(0, I_n)} [f(Y)] - \mathbb{E}_{X \sim \mathcal{D}'} \left[ \tilde{f}(X) \right] \right| \leq O(\varepsilon).$$

To conclude Proposition 20 we want to use the following lemma. It is analogous to [21, Proof of Proposition 2] and we use the same definition of $B_i$, i.e. we define $B_i = \Theta_d(\sqrt{\log(1/\varepsilon)})$. We prove this lemmas in Appendix C.2.

**Lemma 29** (analogous to [21, Proof of Proposition 2]). *Let $\varepsilon > 0$. Suppose that $\mathcal{D}'$ is a distribution such that the following holds*

- $\left| \mathbb{E}_{Y \sim \mathcal{N}(0, I_n)} [f(Y)] - \mathbb{E}_{X \sim \mathcal{D}'} \left[ \tilde{f}(X) \right] \right| \leq O(\varepsilon)$, *and*

- $\mathbb{P}_{X \sim \mathcal{D}'} \left[ \exists i, j : |P_{i,j}(X)| > B_i \right] \leq O(\varepsilon)$.

*Then, we have that*

$$\left| \mathbb{E}_{Y \sim \mathcal{N}(0, I_n)} \left[ f(Y) \right] - \mathbb{E}_{X \sim \mathcal{D}'} \left[ f(X) \right] \right| \leq O_d(\varepsilon).$$

We are now ready to prove Proposition 20.

*Proof of Proposition 20.* Using Lemmas 25, 28 and 29, it remains to argue that

$$\mathbb{P}_{X \sim \mathcal{D}'} \left[ \exists i, j : |P_{i,j}(X)| > B_i \right] \leq O(\varepsilon).$$

This is true by Markov's inequality and looking at the $\log(dn_i/\varepsilon) = \ell$-th moment since this implies that (assuming without loss of generality that $\ell$ is even)

$$
\begin{aligned}
\mathbb{P}_{X \sim \mathcal{D}'} \left[ |P_{i,j}(Y)| \geq B_i \right] &\leq \frac{\mathbb{E}_{X \sim \mathcal{D}'} \left[ P_{i,j}(X)^\ell \right]}{B_i^\ell} \\
&\leq \frac{\mathbb{E}_{Y \sim \mathcal{N}(0, I_n)} \left[ P_{i,j}(Y)^\ell \right] + \eta n^{d\ell/2}}{B_i^\ell} \\
&\leq \frac{\mathbb{E}_{Y \sim \mathcal{N}(0, I_n)} \left[ P_{i,j}(Y)^\ell \right] + 1}{B_i^\ell} \\
&\leq \frac{O_d(\sqrt{\ell})^\ell}{B_i^\ell} \\
&\leq e^{-\ell} \\
&= \frac{\varepsilon}{dn_i}.
\end{aligned}
$$

In the second step we used Fact 23. Note that $\ell = \log(dn_i/\varepsilon) \leq k/d$ is ensured by the choice of $k$ as in [21]. In the third step we used that since $d\ell/2 \leq k$ and thus the condition on $\eta$ in the statement of this proposition ensures $\eta \leq \frac{1}{n^{d\ell/2}}$. In the fourth step we used Lemma 21. In the fifth step we used the condition $B_i \geq \Omega_d(\sqrt{\ell})$. This is true by the choice of $B_i = \Theta_d(\sqrt{\log(1/\varepsilon)})$ and the fact that $\log(n_i) \leq \sum_{j=1}^{i-1} \log(m_j) \leq O_d(\log(1/\varepsilon))$. In the last step, we then used the definition of $\ell$. Taking a union bound over $j$ and then over $i$ gives

$$\mathbb{P}_{X \sim \mathcal{D}'} \left[ \exists i, j : |P_{i,j}(X)| > B_i \right] \leq O(\varepsilon),$$

which completes the proof. $\qquad\square$

### A.2 Fooling for arbitrary PTFs

In this section, we want to prove a result similar to Proposition 20 for arbitrary PTFs and not just multilinear ones. Namely, we show Proposition 12, which we restate with a slight modification below. Namely, we only prove $\left| \mathbb{E}_{Y \sim \mathcal{N}(0, I_n)} \left[ f(Y) \right] - \mathbb{E}_{X \sim \mathcal{D}'} \left[ f(X) \right] \right| \leq O_d(\varepsilon)$ instead of $\varepsilon/2$. The reason for this is, as for Proposition 20, this simplifies the proof and we take care of this difference when we apply Theorem 9 in Appendix A.3.

**Proposition 30** (Restatement of Proposition 12). *Let $\varepsilon > 0$. Suppose that $\mathcal{D}'$ approximately matches the moments of $\mathcal{N}(0, I_n)$ up to degree $k$ and slack $\eta$, where $k \geq \Omega_d \left( \varepsilon^{-4d \cdot 7^d} \right)$, and $\eta \leq n^{-\Omega_d(k)} k^{-\Omega_d(k)}$. Then, we have that for any $f \in \mathcal{F}_{\mathrm{PTF},d}$*

$$\left| \mathbb{E}_{Y \sim \mathcal{N}(0, I_n)} \left[ f(Y) \right] - \mathbb{E}_{X \sim \mathcal{D}'} \left[ f(X) \right] \right| \leq O_d(\varepsilon).$$

The general strategy for this will be to, given a polynomial $p$, find another polynomial $p_\delta$ and reduce to Proposition 20. This strategy and the construction described in what follows are the same as used in [21, Lemma 15]. The following lemma is an analog of this lemma for our case. However, there is one key part of the proof of [21] that breaks in our setting, as explained in Section 2.3.2. Specifically, in [21] all restrictions to coordinates are *exactly* Gaussian, and in particular we have access to moments of all orders. The proof in [21] exploits this since it considers a number of moments depending on the dimension, whereas we only have access to a constant number of moments.

**Lemma 31.** *Let $\delta > 0$. Suppose $X \sim \mathcal{D}'$ approximately matches the moments of $\mathcal{N}(0, I_n)$ up to degree $k$ and slack $\eta$, where $\eta \leq \delta^{O_d(1)} n^{-\Omega_d(1)} k^{-k}$ and $k \geq \Omega_d(1)$. Let $Y \sim \mathcal{N}(0, I_n)$. Then there are a polynomial $p_\delta$ and random variables $\hat{X}$ and $\hat{Y}$, all in more variables, such that*

- *$\hat{Y}$ is a Gaussian with mean $0$ and covariance identity,*

- *$\hat{X}$ is approximately moment-matching $\hat{Y}$ up to degree $k$ and slack $\hat{\eta} = \eta(2k)^{k/2}$,*

- $\mathbb{P}_{Y,\hat{Y}} \left[ |p(Y) - p_\delta(\hat{Y})| > \delta \right] < \delta$, *and,*

- $\mathbb{P}_{X,\hat{X}} \left[ |p(X) - p_\delta(\hat{X})| > \delta \right] < \delta$.

Given this lemma, we can prove Proposition 30. Since this proof follows closely [21, Proof of Theorem 1], we defer it to Appendix C.2

It remains to prove Lemma 31 and to explain how we construct $p_\delta$ as well as $\hat{X}$ and $\hat{Y}$. We do the latter in Appendix A.2.1. Namely, we show there how to construct the random variables $\hat{X}$ and $\hat{Y}$ from $X$ and $Y$. In this section, we also make precise in how many variables the polynomial $p_\delta$ is (and thus also the random variable $\hat{X}$ and $\hat{Y}$). The proof that they satisfy the condition required by Lemma 31 can be found in Appendix C.2. In Appendix A.2.2 we then state a lemma about how we want to replace a factor $X_i^\ell$ in $p$ by a multilinear polynomial in $\hat{X}$ (whose proof is in Appendix C.2) and use it construct $p_\delta$ and prove Lemma 31.

### A.2.1 Construction of the random variables $\hat{X}$ and $\hat{Y}$

To show Lemma 31, we want, given a polynomial $p$ and $\delta > 0$ as well as two random variables $X$ and $Y$, where $Y \sim \mathcal{N}(0, I_n)$ and $X$ matches the moments of $\mathcal{N}(0, I_n)$ up to degree $k$ and slack $\eta$, to construct a multilinear polynomial $p_\delta$ in more variables and two random variables $\hat{X}$ and $\hat{Y}$ such that $\hat{Y}$ is a again Gaussian with mean $0$ and covariance identity and $\hat{X}$ matches the moments of $\hat{Y}$ up to degree $k$ and slack $\hat{\eta}$. The guarantee we then want to show in Lemma 31 is that

$$\mathbb{P}_{Y,\hat{Y}} \left[ |p(Y) - p_\delta(\hat{Y})| > \delta \right] < \delta$$

and

$$\mathbb{P}_{X,\hat{X}} \left[ |p(X) - p_\delta(\hat{X})| > \delta \right] < \delta,$$

where the probability is over the joint distribution of $Y$ and $\hat{Y}$ respectively $X$ and $\hat{X}$.

Let $N$ be a (large) positive integer that will be chosen later. Then the number of variables of the new polynomial $p_\delta$ is $n \cdot N$, i.e. we replace every variable of $p$ by $N$ variables for $p_\delta$. We make the following definition. For $i \in \{1, \ldots, n\}$ and $j \in \{1, \ldots, N\}$, we define

$$\hat{X}_{i,j} := \frac{1}{\sqrt{N}} X_i + Z_j^{(i)},$$

where $Z^{(i)}$ is are multivariate Gaussians with mean $0$, variance $1 - \frac{1}{N}$ and covariance $-\frac{1}{N}$, independent for different $i$ and independent from $X$. In particular, the choice of the covariance matrix ensures that we deterministically have $Z_{i,1} + \ldots Z_{i,N} = 0$ and thus $X_i = \sum_{j=1}^{N} \hat{X}_{i,j}$. The construction for $\hat{Y}$ is the same, i.e.

$$\hat{Y}_{i,j} := \frac{1}{\sqrt{N}} Y_i + Z_j'^{(i)},$$

where $Z'^{(i)}$ are again multivariate Gaussians with mean $0$, variance $1 - \frac{1}{N}$ and covariance $-\frac{1}{N}$, independent for different $i$ and also independent from $Y$ (as well as $X$ and $Z$).

We now have the following two lemmas that relate $\hat{X}$ to $X$ and $\hat{Y}$ to $Y$. The proofs of these lemmas are in Appendix C.2.

**Lemma 32.** *Suppose $X$ approximately matches the moments of $\mathcal{N}(0, I_n)$ up to degree $k$ and slack $\eta$. Then, $\hat{X}$ approximately matches the moments of $\mathcal{N}(0, I_{nN})$ up to degree $k$ and slack $\hat{\eta} = (2k)^{k/2}\eta$.*

**Lemma 33.** *If $Y \sim \mathcal{N}(0, I_n)$, then $\hat{Y} \sim \mathcal{N}(0, I_{nN})$.*

Lemma 33 is already proven in [21, Lemma 15], but for completeness we also make it explicit in Appendix C.2.

### A.2.2 Proof of Lemma 31

After constructing the random variables $\hat{X}$ and $\hat{Y}$, we now move on to construct the polynomial $p_\delta$. As in [21, Proof of Lemma 15], the goal is to replace every factor $x_i^\ell$ of $p$ by a multilinear polynomial in variables $(\hat{x}_{i,j})_j$ such that $X_i^\ell$ is close to this polynomial evaluated in $(\hat{X}_{i,j})_j$ with large probability. Doing this for all factors $x_i^\ell$ appearing in $p$ and combining the new multilinear terms, this then gives a multilinear polynomial $p_\delta$ of degree $d$. Note that the polynomial is in fact multilinear since for replacing $x_i$ we only use the variables $\hat{x}_{i',j}$ where $i' = i$.

Note that it is enough to show

$$\mathbb{P}\left[ |p(X) - p_\delta(\hat{X})| > \delta \right] < \delta.$$

The reason for this is that, if $Y \sim \mathcal{N}(0, I_n)$, then $Y$ in particular matches the moments of $\mathcal{N}(0, I_n)$ (exactly) and the proof for $X$ applies and we can conclude Lemma 31, using also Lemmas 32 and 33.

In order to get the above, we let $\delta'$ be a small positive number (depending on $\delta$, $n$, $d$) to be chosen later. We need the following lemma.

**Lemma 34.** *Let $\delta' > 0$. Let $i \in [n]$ and $\ell \in [d]$. Assume that each of the following conditions holds with probability $1 - \frac{\delta'}{d}$*

$$\left| \sum_{j=1}^{N} \frac{\hat{X}_{i,j}}{\sqrt{N}} \right| \leq \frac{1}{\delta'} \tag{10}$$

$$\left| \sum_{j=1}^{N} \left( \frac{\hat{X}_{i,j}}{\sqrt{N}} \right)^2 - 1 \right| \leq \delta'^{d+1} \tag{11}$$

$$\left| \sum_{j=1}^{N} \left( \frac{\hat{X}_{i,j}}{\sqrt{N}} \right)^a \right| \leq \delta'^{d+1} \qquad \forall\, 3 \leq a \leq d. \tag{12}$$

*Then, there is a multilinear polynomial in $\hat{X}_{i,j}$ that is within $O_d(\delta')$ of $X_i^\ell$ with probability $1 - \delta'$.*

The proof of this lemma is analogous to [21, Proof of Lemma 15] and is deferred to Appendix C.2. However, in contrast to [21], there is a key difference in this lemma. The bound on the RHS of (10) is much weaker than the one used by Kane. The reason for that is that the bounds used there do not hold in our case, since we only have that $X$ (approximately) matches the moments of $\mathcal{N}(0, I_n)$. By using stronger bounds for (11) and (12), we are able to generalize the proof from Kane to our setting. For a more detailed explanation why we need to change the bounds, we refer to Section 2.3.2.

We now define

$$\delta' := \min\left\{ \frac{\delta}{2dn}, \Theta_d\left( \frac{\delta^{d+1}}{n^{3d/2}} \right) \right\} = \Theta_d\left( \frac{\delta^{d+1}}{n^{3d/2}} \right). \tag{13}$$

Why we make this choice will become clear in the proof of Lemma 31 below. In Appendix C.2, we prove the following lemma that states that this choice of $\delta'$ and the condition on $\eta$ from Lemma 31 ensure that we can apply Lemma 34.

**Lemma 35.** *Let $\delta'$ as in (13). Assuming $\eta \leq \delta^{O_d(1)} n^{-\Omega_d(1)} k^{-k}$, there is a choice of $N$ (independent of $i$ or $\ell$) such that (10), (11) and (12) hold, each with probability $1 - \frac{\delta'}{d}$.*

*Proof of Lemma 31.* Lemma 34 (together with Lemma 35) shows that we can replace $X_i^\ell$ by a multilinear polynomial in $\hat{X}_{i,j}$ that is within $O_d(\delta')$ of $X_i^\ell$ with probability $1 - \delta'$ for our choice of $\delta'$ as in (13). Since $\delta' \leq \frac{\delta}{2dn}$, we can union bound these events over all $i \in [n]$ and $\ell \in [d]$ and get

that with probability $1 - \frac{\delta}{2}$, we have that for any $i \in [n]$ and $\ell \in [d]$, we have that the replacement polynomial for $X_i^\ell$ is within $O_d(\delta')$ of $X_i^\ell$.

Furthermore, for any $i \in [n]$, we have that with probability $1 - \frac{\delta}{2n}$ that $|X_i| \leq 3\frac{n}{\delta}$. This is true since we match $k \geq 2$ moments (and $\eta \leq \frac{1}{4}$), and thus

$$\mathbb{P}\left[|X_i| \geq 3\frac{n}{\delta}\right] \leq \frac{\mathbb{E}\left[X_i^2\right]}{\left(3\frac{n}{\delta}\right)^2} \leq \frac{(2+\eta)\delta^2}{9n^2} \leq \frac{\delta}{4n}.$$

Hence, we can again apply the union bound to show that with probability $1 - \frac{\delta}{2}$, we have for any $i \in [n]$ that $|X_i| \leq 3\frac{n}{\delta}$.

Thus, with probability $1 - \delta$, all the above events holds. Conditioned on that, we have that the replacement polynomial $p_\delta$ is off by at most $\left(3\frac{n}{\delta}\right)^d O_d(\delta')$ multiplied by the sum of the coefficients of $p$. The later is at most $n^{d/2}$ by Fact 22 applied to $p$ instead of $P_{i,j}$. Hence, the replacement polynomial is off by at most $O_d\left(\frac{n^{3d/2}}{\delta^d}\right)\delta'$. Thus, since $\delta' \leq \frac{\delta}{\Omega_d\left(\frac{n^{3d/2}}{\delta^d}\right)}$, we get that with probability $1 - \delta$, $p_\delta$ is off by at most $\delta$, which is what we wanted to show. $\qquad\square$

### A.3   Proof of testable learning of PTFs

In this section, we prove Theorem 19. As already mentioned in the beginning of this section, we want to apply Theorem 9 for this. In Proposition 30, we have shown that if we have moment matching up to error $\eta \leq n^{-\Omega_d(k)}k^{-\Omega_d(k)}$, then we have the fooling condition of Theorem 9.

Note that the fooling condition requires $|\mathbb{E}_{Y \sim \mathcal{N}(0, I_n)}[f(Y)] - \mathbb{E}_{X \sim \mathcal{D}'}[f(X)]| \leq \frac{\varepsilon}{2}$ but Proposition 30 only gives $O_d(\varepsilon)$. Thus, technically, we apply Proposition 30 for $\varepsilon' = \frac{\varepsilon}{\Omega_d(1)}$. However, this does not change the asymptotic condition on $\eta$ described above. In summary, if $\eta$ satisfies the condition as described above, we get indeed the fooling condition as needed for Theorem 9.

The remaining part to prove Theorem 19 is to find an $m$ such that with high probability over $m$ samples from $\mathcal{N}(0, I_n)$ we have that the empirical distribution matches the moments up to degree $k$ with error at most $\eta$. Then, we get testable learning of PTFs with respect to Gaussian in time and sample complexity $m + n^{O(k)}$ by Theorem 9.

To get $m$, we use the following fact, which we prove in Appendix C.1. Using this, we can then prove Theorem 19.

**Fact 36.** *Given $m \geq \Omega((2kn)^k\eta^{-2})$ samples of $\mathcal{N}(0, I_n)$, we have that with high probability the empirical distribution matches the moments of $\mathcal{N}(0, I_n)$ up to degree $k$ and slack $\eta$.*

*Proof of Theorem 19.* Using Theorem 9, as noted before, by Proposition 30, we get testable learning of degree $d$ PTFs with respect to Gaussian in time and sample complexity bounded by $m + n^{O(k)}$, where $m$ is such that with high probability over $m$ samples from $\mathcal{N}(0, I_n)$ we have that the empirical distribution matches the moments up to degree $k$ with error at most $\eta$. It remains to determine $m$. By Fact 36, we get that the choice of $m = \Theta((2kn)^k\eta^{-2})$ is enough. Now, in order to apply Proposition 30, we need to choose $\eta = n^{-\Theta_d(k)}k^{-\Theta_d(k)}$. Plugging in the value $k = \Theta_d\left(\varepsilon^{-4d\cdot 7^d}\right)$ we get $\eta = \varepsilon^{\Theta_d\left(\varepsilon^{-4d\cdot 7^d}\right)} n^{-\Theta_d\left(\varepsilon^{-4d\cdot 7^d}\right)}$ and hence

$$m = \Theta\left((2kn)^k n^{\Theta_d(\varepsilon^{-4d\cdot 7^d})}\varepsilon^{-\Theta_d(\varepsilon^{-4d\cdot 7^d})}\right).$$

Thus, the time and sample complexity for testably learning PTFs is

$$O\left((2kn)^k n^{O_d\left(\varepsilon^{-4d\cdot 7^d}\right)}\varepsilon^{-\Omega_d\left(\varepsilon^{-4d\cdot 7^d}\right)} n^{O(k)}\right).$$

Again, by plugging in the value $k = \Theta_d\left(\varepsilon^{-4d\cdot 7^d}\right)$, we can simplify this to get that the sample and time complexity for testably learning PTFs is

$$n^{O_d\left(\varepsilon^{-4d\cdot 7^d}\right)}\varepsilon^{-\Omega_d\left(\varepsilon^{-4d\cdot 7^d}\right)},$$

which completes the proof of this theorem. $\qquad\square$

# B  Impossibility of approximating PTFs via the push-forward

In this section, we provide further details on the results claimed in Section 2.4. Specifically, we prove our impossibility result Theorem 16 below in Appendix B.2. For this, we first need to establish our 'one-sided' analog Theorem 18 of the inapproximability result for LSL-distributions of Bun & Steinke (Theorem 15), which we do in Appendix B.1.

## B.1  Proof of Theorem 18

Let us begin by restating some important definitions and results from Section 2.4 for convenience. For $\gamma > 0$, we write $w_\gamma(x) := C_\gamma \cdot \exp(-|x|^\gamma)$, where $C_\gamma$ is a normalizing constant which ensures $w_\gamma$ is a probability density. A distribution $\mathcal{D}$ on $\mathbb{R}$ is called *log-superlinear (LSL)* if its density function[3] satisfies $w(x) \geq C \cdot w_\gamma(x)$ for all $x \in \mathbb{R}$, for some $\gamma \in (0, 1)$ and $C > 0$. Recall the following.

**Theorem** (Restatement of Theorem 15). *Let $\mathcal{D}$ be an LSL-distribution on $\mathbb{R}$. Then there exists an $\varepsilon > 0$ such that, for any polynomial $q$, we have $\mathbb{E}_{Y \sim \mathcal{D}} \left[ |q(Y) - \mathrm{sign}(Y)| \right] > \varepsilon$.*

We defined in Section 2.4 a 'one-sided' analog of LSL-distributions as follows.

**Definition** (Restatement of Definition 17). *Let $\mathcal{D}$ be a distribution on $\mathbb{R}$ whose density function $w$ satisfies*

$$w(x) \geq C \cdot w_\gamma(x) \quad \forall\, x \in (-\infty, 1]$$

*for some $\gamma < 1/2$ and $C > 0$. Then we say $\mathcal{D}$ is* one-sided *log-superlinear.*

In this section, we prove Theorem 18, which is an analog of Theorem 15 for one-sided LSL-distributions, and the basis of our proof of Theorem 16 in Appendix B.2.

**Theorem** (Restatement of Theorem 18). *Let $\mathcal{D}$ be a one-sided LSL-distribution on $\mathbb{R}$. Then there exists an $\varepsilon > 0$ such that, for any polynomial $q$, we have $\mathbb{E}_{Y \sim \mathcal{D}} \left[ |q(Y) - \mathrm{sign}(Y)| \right] > \varepsilon$.*

For our proof, it is useful to first recall the main ingredient of the proof of Theorem 15 in [6], which is the following inequality.

**Proposition 37** ([6, Lemma 20]). *Let $q$ be a univariate polynomial, and let $\gamma \in (0, 1)$. Then, there exists a constant $M_\gamma > 0$, depending only on $\gamma$, so that*

$$\sup_{x \in \mathbb{R}} |q'(x) w_\gamma(x)| \leq M_\gamma \cdot \int_{\mathbb{R}} |q(x)| w_\gamma(x)\, dx.$$

Given Proposition 37, the intuition for the proof of Theorem 15 is that, if $q$ is a good approximation of the sign-function on $\mathbb{R}$, it must have very large derivative near the origin. On the other hand, $\mathbb{E}_{Y \sim \mathcal{D}} \left[ |q(Y)| \right]$ is bounded from above (as $\mathbb{E}_{Y \sim \mathcal{D}} \left[ |\mathrm{sign}(Y)| \right] = 1$), leading to a contradiction.

The key technical tool in our proof of Theorem 18 is a version of Proposition 37 that applies to *one-sided* LSL distributions. That is, a bound on the derivative of a polynomial in terms of its $L_1$-norm on $(-\infty, 1]$ w.r.t. the weight $w_\gamma(x) = C_\gamma \cdot \exp(-|x|^\gamma)$, $\gamma < 1/2$. We will only be able to obtain such a bound in a small neighborhood of 0 (Proposition 39 below), but this will turn out to be sufficient. We first need the following lemma, which bounds the derivative near 1. One can think of it as a one-sided *Bernstein*-Nikolskii-type inequality (whereas Proposition 37 is a *Markov*-Nikolskii-type inequality).

**Lemma 38.** *Let $q$ be a univariate polynomial, and let $\gamma < 1/2$. Then, there exists a constant $M_\gamma' > 0$, depending only on $\gamma$, so that*

$$\sup_{|x-1| \leq \frac{1}{4}} |q'(x)| \leq M_\gamma' \cdot \int_0^\infty |q(x)| w_\gamma(x)\, dx.$$

*Proof.* By a substitution $u^2 = x$, and using the fact that $w_\gamma(x) := C_\gamma \exp(-|x|^\gamma)$ is even, we find

$$\int_0^\infty |q(x)| w_\gamma(x)\, dx = \int_0^\infty |q(u^2)| w_\gamma(u^2) \cdot 2u\, du = \frac{C_\gamma}{C_{2\gamma}} \cdot \int_{-\infty}^\infty |q(u^2)u| w_{2\gamma}(u)\, du.$$

---

[3]Technically, density functions are defined only up to measure-zero sets. Therefore, one should read statements of the form '$w(x) \geq \ldots$ for all $x \in \ldots$' as only holding a.e. throughout.

Now, since $2\gamma < 1$, we may apply Proposition 37 to the polynomial $\hat{q}(u) = q(u^2)u$, yielding an upper bound on its derivative for all $u \in \mathbb{R}$; namely

$$w_{2\gamma}(u) \cdot \left| \frac{\mathrm{d}}{\mathrm{d}u} \hat{q}(u) \right| \leq M_{2\gamma} \cdot \int_{-\infty}^{\infty} |q(u^2)u| w_{2\gamma}(u) \, du$$
$$= \frac{M_{2\gamma} C_{2\gamma}}{C_\gamma} \cdot \int_{0}^{\infty} |q(x)| w_\gamma(x) \, dx.$$

As $w_{2\gamma}$ is monotonically decreasing, we have $w_{2\gamma}(x) \geq w_{2\gamma}(5/4)$ for all $x \leq 5/4$, and so we find

$$\sup_{|u^2 - 1| \leq \frac{1}{4}} \left| \frac{\mathrm{d}}{\mathrm{d}u} \hat{q}(u) \right| \leq \hat{M}_\gamma \cdot \int_{0}^{\infty} |q(x)| w_\gamma(x) \, dx, \quad \hat{M}_\gamma := \frac{M_{2\gamma} C_{2\gamma}}{C_\gamma \cdot w_{2\gamma}(\frac{5}{4})}. \tag{14}$$

We wish to transform this bound on the derivative of $\hat{q}$ into a bound on the derivative of $q$. Note that

$$\frac{\mathrm{d}}{\mathrm{d}u} \hat{q}(u) = \frac{\mathrm{d}}{\mathrm{d}u} [q(u^2)u] = 2q'(u^2)u^2 + q(u^2).$$

We can bound this as follows

$$\sup_{|u^2 - 1| \leq \frac{1}{4}} |2q'(u^2)u^2 + q(u^2)| \geq \frac{3}{2} \cdot \sup_{|u^2 - 1| \leq \frac{1}{4}} |q'(u^2)| - \sup_{|u^2 - 1| \leq \frac{1}{4}} |q(u^2)|.$$

Now, switching our notation back to the variable $x = u^2$, and using (14), we have

$$\frac{3}{2} \cdot \sup_{|x - 1| \leq \frac{1}{4}} |q'(x)| \leq \hat{M}_\gamma \cdot \int_{0}^{\infty} |q(x)| w_\gamma(x) \, dx + \sup_{|x - 1| \leq \frac{1}{4}} |q(x)|. \tag{15}$$

It remains to bound the rightmost term in this inequality. Write $c_q > 0$ for the constant (depending on $q$) satisfying:

$$\sup_{|x - 1| \leq \frac{1}{4}} |q(x)| = \frac{c_q}{w_\gamma(\frac{5}{4})} \cdot \int_{0}^{\infty} |q(x)| w_\gamma(x) \, dx. \tag{16}$$

If $c_q \leq 2$, we are done immediately by (15), as then

$$\frac{3}{2} \cdot \sup_{|x - 1| \leq \frac{1}{4}} |q'(x)| \leq \left( \hat{M}_\gamma + \frac{2}{w_\gamma(\frac{5}{4})} \right) \cdot \int_{0}^{\infty} |q(x)| w_\gamma(x) \, dx.$$

So assume that $c_q > 2$. Note that

$$\int_{0}^{\infty} |q(x)| w_\gamma(x) \, dx \geq w_\gamma(\tfrac{5}{4}) \cdot \int_{\frac{3}{4}}^{\frac{5}{4}} |q(x)| \, dx \geq w_\gamma(\tfrac{5}{4}) \cdot \inf_{|x - 1| \leq \frac{1}{4}} |q(x)|,$$

and so

$$\inf_{|x - 1| \leq \frac{1}{4}} |q(x)| \leq \frac{1}{w_\gamma(\frac{5}{4})} \cdot \int_{0}^{\infty} |q(x)| w_\gamma(x) \, dx. \tag{17}$$

Applying the mean value theorem to (16) and (17) on the interval $[\frac{3}{4}, \frac{5}{4}]$, we find that

$$\sup_{|x - 1| \leq \frac{1}{4}} |q'(x)| \geq 2 \cdot \frac{c_q - 1}{w_\gamma(\frac{5}{4})} \cdot \int_{0}^{\infty} |q(x)| w_\gamma(x) \, dx = \frac{2(c_q - 1)}{c_q} \cdot \sup_{|1 - x| \leq \frac{1}{4}} |q(x)|.$$

As $c_q > 2$ by assumption, this yields

$$\sup_{|x - 1| \leq \frac{1}{4}} |q'(x)| \geq \sup_{|x - 1| \leq \frac{1}{4}} |q(x)|,$$

and we may conclude from (15) that

$$\left( \frac{3}{2} - 1 \right) \cdot \sup_{|x - 1| \leq \frac{1}{4}} |q'(x)| \leq \hat{M}_\gamma \cdot \int_{0}^{\infty} |q(x)| w_\gamma(x) \, dx. \qquad \square$$

Combining the cases $c_q \leq 2$, $c_q > 2$ finishes the proof with $M'_\gamma = \max\{\hat{M}_\gamma + \frac{2}{w_\gamma(5/4)}, 2\hat{M}_\gamma\}$.

**Proposition 39.** *Let $\mathcal{D}$ be a one-sided LSL distribution on $\mathbb{R}$. Then there exists a constant $C_{\mathcal{D}} > 0$ such that, for any polynomial $q$, we have*

$$\sup_{|x| \leq \frac{1}{4}} |q'(x)| \leq C_{\mathcal{D}} \cdot \mathbb{E}_{X \sim \mathcal{D}} \left[ |q(X)| \right].$$

*Proof.* We wish to use Lemma 38, which gives us a bound on the derivative $q'(x)$ of $q$ when $x \approx 1$. To transport this bound to the origin, we consider the shifted polynomial $\hat{q}(x) := q(1-x)$. Let $w$ be the density function of $\mathcal{D}$. Since $\mathcal{D}$ is a one-side LSL-distribution, there exists a constant $C > 0$ and a $\gamma \in (0, 1/2)$ such that $w(x) \geq C \cdot w_\gamma(x)$ for all $x \in (-\infty, 1]$. As $w_\gamma$ is even, bounded from above and below on $[0, 1]$, and $w_\gamma(x-1) \leq w_\gamma(x)$ for $x \leq 0$, we can find a constant $C' > 0$ such that

$$w(x) \geq C' \cdot w_\gamma(1-x) \quad \forall\, x \in (-\infty, 1].$$

Now, we find that

$$\int_0^\infty |\hat{q}(x)| w_\gamma(x)\, dx = \int_{-\infty}^1 |\hat{q}(1-x)| w_\gamma(1-x)\, dx$$

$$\leq \frac{1}{C'} \cdot \int_{-\infty}^1 |q(x)| w(x)\, dx \leq \frac{1}{C'} \cdot \mathbb{E}_{X \sim \mathcal{D}} \left[ |q(X)| \right].$$

As $\gamma < 1/2$, we may apply Lemma 38 to find that

$$\sup_{|x-1| \leq \frac{1}{4}} |(\hat{q})'(x)| \leq M'_\gamma \cdot \int_0^\infty |\hat{q}(x)| w_\gamma(x)\, dx \leq \frac{M'_\gamma}{C'} \cdot \mathbb{E}_{X \sim \mathcal{D}} \left[ |q(X)| \right].$$

To finish the proof (with $C_{\mathcal{D}} = M'_\gamma / C'$), it remains to note that

$$\sup_{|x-1| \leq \frac{1}{4}} |(\hat{q})'(x)| = \sup_{|x| \leq \frac{1}{4}} |q'(x)|. \qquad \square$$

We are now ready to prove Theorem 18. Our approach is similar to the proof of Theorem 15 in [6].

*Proof of Theorem 18.* Let $\mathcal{D}$ be a one-side LSL-distribution, and suppose that $q$ is a univariate polynomial satisfying

$$\mathbb{E}_{X \sim \mathcal{D}} \left[ |\operatorname{sign}(X) - q(X)| \right] \leq 1.$$

Since $\mathbb{E}_{X \sim \mathcal{D}} \left[ |\operatorname{sign}(X)| \right] = 1$, we may use the triangle inequality to find that

$$\mathbb{E}_{X \sim \mathcal{D}} \left[ |q(X)| \right] \leq 1 + 1 = 2.$$

By Proposition 39, this means that, for some constant $C_{\mathcal{D}} > 1$ depending only on $\mathcal{D}$,

$$\sup_{|x| \leq \frac{1}{4}} |q'(x)| \leq 2 \cdot C_{\mathcal{D}}.$$

Set $\eta = (10 C_{\mathcal{D}})^{-1}$. It follows that

$$\sup_{|x| \leq \eta} |q(x) - q(0)| \leq (10 C_{\mathcal{D}})^{-1} \cdot 2 C_{\mathcal{D}} = \frac{1}{5}.$$

Assuming first that $q(0) \leq 0$, this means that

$$|q(x) - \operatorname{sign}(x)| \geq \frac{4}{5} \quad \forall\, x \in [0, \eta].$$

Let $w$ be the density function $\mathcal{D}$. As $\mathcal{D}$ is one-sided LSL, and $\eta < 1$, we know there is a constant $C > 0$, independent of $\eta$, such that $w(x) \geq C$ for $x \in [0, \eta]$. But, this implies that

$$\mathbb{E}_{X \sim \mathcal{D}} \left[ |\operatorname{sign}(X) - q(X)| \right] \geq C \int_0^\eta |\operatorname{sign}(x) - q(x)|\, dx \geq \frac{4}{5} \eta \cdot C,$$

giving a uniform lower bound on $\mathbb{E}_{X \sim \mathcal{D}} \left[ |\operatorname{sign}(X) - q(X)| \right]$ for all polynomials $q$. If, on the other hand $q(0) \geq 0$, the same argument works after replacing $[0, \eta]$ by $[-\eta, 0]$. $\qquad \square$

### B.2 Proof details for Theorem 16

In this section, we complete the proof of Theorem 16 started in Section 2.4. Recall that, in light of Theorem 18, it remained to prove the following.

**Proposition 40.** *Let $p(x) = -x(x-1)(x-2)(x-3)(x-4)(x-5)$. Then, $p$ is square-free, $\{p \geq 0\} \subseteq \mathbb{R}$ is compact, and $p_\# \mathcal{N}(0,1)$ is one-sided log-superlinear.*

*Proof.* The first two properties follow directly. For the third, let $w$ be the density function of $p_\# \mathcal{N}(0,1)$. Applying the inverse function rule to $w_2(x) = C_2 \cdot \exp(-|x|^2)$, we have

$$w(y) = C_2 \cdot \sum_{x \in p^{-1}(y)} \exp(-|x|^2) \cdot \frac{1}{|p'(x)|} \quad \forall y \in \mathbb{R}.$$

We need to show that, for some $C > 0$ and $\gamma \in (0, 1/2)$,

$$w(y) \geq C \cdot \exp(-|y|^\gamma) \quad \forall y \in (-\infty, 1]. \tag{18}$$

Qualitatively, the behavior of $w$ can be described as follows. When $y \geq \sup_{x \in \mathbb{R}} p(x) \approx 17$, then $w(y) = 0$. When $y \ll 0$, then $x \in p^{-1}(y)$ implies $|x| \approx |y|^{1/6}$, and so $w(y) \approx \exp(-|y|^{1/3})$. Finally, for any $K > 0$, there is a uniform lower bound on $w(y)$ for all $y \in [-K, 1]$. To show (18), it remains to make this description quantitative.

As the leading term of $p$ is $-x^6$, there is a $K > 0$ such that $|p^{-1}(y)| = 2$ for all $y \leq -K$, and

$$x \in p^{-1}(y) \implies |x| \leq 2|y|^{1/6} \quad \forall y \leq -K.$$

This means that, for a (possibly larger) constant $K' > 0$, and some $\gamma \in (1/3, 1/2)$, we have

$$w(y) \geq 2 \cdot C_2 \cdot \exp\left(-4|y|^{1/3}\right) \cdot \frac{1}{|p'(2|y|^{1/6})|} \geq \exp(-|y|^\gamma) \quad \forall y \leq -K'. \tag{19}$$

Now, there is an $M > 0$ so that

$$x \in p^{-1}(y) \implies |x| \leq M \quad \forall y \in [-K', 1].$$

It follows, as $p^{-1}(y)$ is non-empty for $y \leq 1$, that

$$w(y) \geq \inf_{|x| \leq M} \left[ C_2 \cdot \exp(-|x|^2) \cdot \frac{1}{|p'(x)|} \right] > 0 \quad \forall y \in [-K', 1]. \tag{20}$$

Combining (19) and (20), and choosing $C > 0$ small enough, yields (18). $\qquad \square$

## C Technical details on the proof of testable learning

### C.1 Facts and computations

In this section, we prove several facts that we used in the proofs in Appendix A about the moments and other properties of a Gaussian random variable $Y$ and a random variable $X$ that matches the moments of a Gaussian up to degree $k$ and slack $\eta$.

First, we want to prove Fact 22, which we restate below, about the sum of the absolute values of the coefficients of the polynomials $P_{i,j}$. Recall from Lemma 21 that the sum of the squares of the coefficients of the polynomials $P_{i,j}$ is 1.

**Fact** (Restatement of Fact 22). *For any $i \in [n]$ and $j \in [n_i]$, the sum of the absolute values of the coefficients of $P_{i,j}$ is at most $n^{d/2}$.*

*Proof.* Similar to before, we write the $P_{i,j}$ as follows

$$P_{i,j}(x) = \sum_\beta a_{i,j}^{(\beta)} x^\beta.$$

The $\beta$ in the sum goes over all multi-indices with $|\beta|$ being the degree of $P_{i,j}$, which is at most $d$. Thus, there are at most $n^d$ terms in the sum. By the structure theorem (Lemma 21), we know that

$\|a_{i,j}\|_2 = \sum_\beta (a_{i,j}^{(\beta)})^2 = 1$. We can now get a bound on the 1-norm as follows (since the number of coefficients $a_{i,j}^{(\beta)}$ is at most $n^d$)

$$\sum_\beta |a_{i,j}^{(\beta)}| = \|a_{i,j}\|_1 \leq n^{d/2}\|a_{i,j}\|_2 \leq n^{d/2}. \qquad \square$$

Next, we want to show two facts about the moments of $X$. On the one hand, we want to show that under very mild assumptions on $\eta$, we can bound the moments of $X$ similar as the moments of a Gaussian $Y$. We also prove Fact 23, which we also restate below, about the expectation of $P_{i,j}(X)$. For the Gaussian, we get a bound by Lemma 21 and we generalize this in this fact to the moment-matching distribution.

**Fact 41.** *Let $\eta \leq 1$ and let $\mathcal{D}'$ be a distribution that matches the moments of $\mathcal{N}(0, I_n)$ up to degree $k$ and slack $\eta$. Then, we have that for any multi-index $\alpha$ with $|\alpha| \leq k$ we have*

$$|\mathbb{E}_{X \sim \mathcal{D}'}[X^\alpha]| \leq \sqrt{|\alpha|}^{|\alpha|}.$$

*Proof.* We have that

$$|\mathbb{E}_{X \sim \mathcal{D}'}[X^\alpha]| \leq |\mathbb{E}_{Y \sim \mathcal{N}(0,I_n)}[Y^\alpha]| + 1$$

$$\leq |\mathbb{E}_{Z \sim \mathcal{N}(0,1)}[Z^{|\alpha|}]| + 1$$

$$\leq 1 + \begin{cases} 0 & \text{if } |\alpha| \text{ is odd} \\ (|\alpha| - 1)!! & \text{otherwise} \end{cases}$$

$$\leq \sqrt{|\alpha|}^{|\alpha|},$$

which is what we wanted to proof. $\qquad \square$

**Fact** (Restatement of Fact 23). *For any $i \in [d]$, $j \in [n_i]$ and $\ell \leq k/d$, we have that*

$$\mathbb{E}_{X \sim \mathcal{D}'}[P_{i,j}(X)^\ell] \leq \mathbb{E}_{Y \sim \mathcal{N}(0,I_n)}[P_{i,j}(Y)^\ell] + \eta n^{d\ell/2}.$$

*Proof.* Writing $P_{i,j}$ as in Fact 22

$$P_{i,j}(x) = \sum_\beta a_{i,j}^{(\beta)} x^\beta,$$

we can compute $\mathbb{E}_{X \sim \mathcal{D}'}[P_{i,j}(X)^\ell]$ as follows

$$\mathbb{E}_{X \sim \mathcal{D}'}[P_{i,j}(X)^\ell] = \sum_{\beta_1,\ldots,\beta_\ell} a_{i,j}^{(\beta_1)} \ldots a_{i,j}^{(\beta_\ell)} \mathbb{E}_{X \sim \mathcal{D}'}[X^{\beta_1 + \ldots \beta_\ell}].$$

Now, since $X$ is $\eta$-approximately moment matching, we have that (note that $P_{i,j}$ has degree at most $d$ and thus any term in the sum that degree at most $d\ell \leq k$ and thus the moment matching applies)

$$\mathbb{E}_{X \sim \mathcal{D}'}[X^{\beta_1 + \ldots \beta_\ell}] = \mathbb{E}_{Y \sim \mathcal{N}(0,I_n)}[Y^{\beta_1 + \ldots \beta_\ell}] \pm \eta.$$

Combining this with the above, we get

$$\mathbb{E}_{X \sim \mathcal{D}'}[P_{i,j}(X)^\ell] \leq \mathbb{E}_{Y \sim \mathcal{N}(0,I_n)}[P_{i,j}(Y)^\ell] + \eta \left(\sum_\beta |a_{i,j}^{(\beta)}|\right)^\ell.$$

By Fact 22, we thus get that

$$\mathbb{E}_{X \sim \mathcal{D}'}[P_{i,j}(X)^\ell] \leq \mathbb{E}_{Y \sim \mathcal{N}(0,I_n)}[P_{i,j}(Y)^\ell] + \eta n^{d\ell/2}$$

as wanted. $\qquad \square$

Finally, we show two facts about the Gaussian distribution. First, we want to give a bound on the moments of a Gaussian random vector that has not necessarily independent entries and for which we only know that the variances are at most 1. If $Z$ were independent, then the bound we show would follow directly by the formulas for the moments of a standard Gaussian. Second, we show Fact 36 that shows how many samples $m$ of $\mathcal{N}(0, I_n)$ we need such that the empirical moments up to degree $k$ match the actual moments of $\mathcal{N}(0, I_n)$ up to slack $\eta$.

**Fact 42.** *Let $Z$ be a (multivariate) Gaussian random variable with mean $0$ and variances at most $1$. Then, for any multi-index $\beta$, we have that*

$$\left| \mathbb{E}\left[ Z^\beta \right] \right| \leq |\beta|^{|\beta|/2}.$$

*Proof.* We want to show that for any random variables $W_1, \ldots, W_\ell$ we have

$$\left( \mathbb{E}\left[ \prod_{j=1}^\ell W_j \right] \right)^\ell \leq \prod_{j=1}^\ell \mathbb{E}\left[ W_j^\ell \right].$$

To prove this, we use induction and Hölder's inequality. The case $\ell = 1$ follows directly and for $\ell \geq 2$ we can compute using Hölder's inequality, since $\frac{1}{\frac{\ell}{\ell-1}} + \frac{1}{\ell} = 1$,

$$\mathbb{E}\left[ \prod_{j=1}^\ell W_j \right] \leq \left( \mathbb{E}\left[ \prod_{j=1}^{\ell-1} W_j^{\frac{\ell}{\ell-1}} \right] \right)^{\frac{\ell-1}{\ell}} \cdot \left( \mathbb{E}\left[ W_\ell^\ell \right] \right)^{\frac{1}{\ell}}.$$

Thus, we get, using the induction hypothesis,

$$\left( \mathbb{E}\left[ \prod_{j=1}^\ell W_j \right] \right)^\ell \leq \left( \mathbb{E}\left[ \prod_{j=1}^{\ell-1} W_j^{\frac{\ell}{\ell-1}} \right] \right)^{\ell-1} \cdot \mathbb{E}\left[ W_\ell^\ell \right] \leq \prod_{j=1}^\ell \mathbb{E}\left[ W_j^\ell \right].$$

We now use this result with $\ell = |\beta|$ and the $W_1, \ldots, W_\ell$ being the $Z_j$, where every entry $Z_j$ occurs $\beta_j$ times. Then we get

$$\left| \mathbb{E}\left[ Z^\beta \right] \right| \leq \left( \prod_j \left( \mathbb{E}\left[ Z_j^\ell \right] \right)^{\beta_j} \right)^{\frac{1}{\ell}}.$$

Since $\mathbb{E}\left[ Z_j \right] = 0$ and $\operatorname{Var}\left[ Z_j \right] \leq 1$, we have that

$$\mathbb{E}\left[ Z_j^\ell \right] \leq \ell^{\ell/2}$$

and thus we get

$$\left| \mathbb{E}\left[ Z^\beta \right] \right| \leq \ell^{\ell/2} = |\beta|^{|\beta|/2}. \qquad \square$$

**Fact** (Restatement of Fact 36). *Given $m \geq \Omega((2kn)^k \eta^{-2})$ samples of $\mathcal{N}(0, I_n)$, we have that with high probability the empirical distribution matches the moments of $\mathcal{N}(0, I_n)$ up to degree $k$ and slack $\eta$.*

*Proof.* Given $m$ samples from $\mathcal{N}(0, I_n)$, we want to compute the probability that for some $\alpha$ with $|\alpha| \leq k$ we have that the empirical moment is close to the moment with high probability. We can compute using Chebyshev's inequality, where $c_\alpha$ is the $\alpha$-moment of $\mathcal{N}(0, I_n)$ and $\hat{c}_\alpha$ is the empirical moment, that

$$\mathbb{P}\left[ |\hat{c}_\alpha - c_\alpha| > \eta \right] \leq \frac{\operatorname{Var}\left[ \hat{c}_\alpha \right]}{\eta^2} = \frac{1}{m} \frac{\operatorname{Var}\left[ Y^\alpha \right]}{\eta^2} \leq \frac{1}{m} \frac{\operatorname{Var}\left[ Y_1^{|\alpha|} \right]}{\eta^2} \leq \frac{1}{m} \frac{(2|\alpha|)^{|\alpha|}}{\eta^2} \leq \frac{1}{m} \frac{(2k)^k}{\eta^2}.$$

To be able to use a union-bound, we need this to be smaller than $O(n^{-k})$, i.e. we need

$$m \geq \Omega\left( \frac{(2kn)^k}{\eta^2} \right). \qquad \square$$

## C.2 Remaining proofs

In this section, we prove several lemmas that follow closely [21]. Some of these lemmas are also proven in [21], but we include them here for completeness. We first prove Lemmas 26 and 27, which we need in order to show that the expectation of $\tilde{f}$ is close under the moment-matching distribution and the Gaussian distribution.

**Lemma** (Restatement of Lemma 26). *Let $\varepsilon > 0$. Suppose that $\mathcal{D}'$ approximately matches the moments of $\mathcal{N}(0, I_n)$ up to degree $k$ and slack $\eta$, where $k \geq \Omega_d\left(\varepsilon^{-4d \cdot 7^d}\right)$ and $\eta \leq \frac{1}{kd}$. Then, we have that*

$$\mathbb{E}_{X \sim \mathcal{D}'}\left[|\tilde{f}(X) - T(P(X))|\right] \leq O(\varepsilon)$$

As mentioned earlier, this proof follows closely [21, Proof of Proposition 8]. In particular, we also need the following lemma from [21].

**Lemma 43** ([21, Proposition 6]). *We have that, for $x \in \mathbb{R}^n$,*

$$|T(P(x)) - \tilde{F}(P(x))| \leq \prod_{i=1}^{d}\left(1 + \frac{C_i^{m_i}\|P_i(x)\|_2^{m_i}}{m_i!}\right) - 1.$$

The $C_i$ and $m_i$ are again as in [21], i.e. we choose $C_i = \Theta_d\left(\varepsilon^{-7^i d}\right)$ and $m_i = \Theta_d\left(\varepsilon^{-3 \cdot 7^i d}\right)$. For the proof, we furthermore need the bound on the expectation of $|P_{i,j}(X)|$ from Fact 23.

*Proof of Lemma 26.* We have by Lemma 43 that

$$|T(P(x)) - \tilde{F}(P(x))| \leq \prod_{i=1}^{d}\left(1 + \frac{C_i^{m_i}\|P_i(x)\|_2^{m_i}}{m_i!}\right) - 1.$$

As in [21, Proof of Proposition 8], the RHS of this inequality is the sum over all non-empty subsets $S \subseteq \{1, 2, \ldots, d\}$ of the following term

$$\prod_{i \in S}\left(\frac{C_i^{m_i}\|P_i(x)\|_2^{m_i}}{m_i!}\right).$$

Continuing exactly as in [21, Proof of Proposition 8], we want to show that any of these $2^d - 1$ terms is at most $O(\varepsilon/2^d)$. By the inequality of arithmetic and geometric means, we get that this term is at most

$$\frac{1}{|S|}\sum_{i \in S}\left(\frac{C_i^{m_i}\|P_i(x)\|_2^{m_i}}{m_i!}\right)^{|S|}.$$

Thus, it remains to bound $\mathbb{E}_{X \sim \mathcal{D}'}\left[\|P_i(X)\|_2^{\ell_i}\right]$, where $\ell_i = m_i|S|$. We do this as follows

$$\mathbb{E}_{X \sim \mathcal{D}'}\left[\|P_i(X)\|_2^{\ell_i}\right] \leq \sqrt{\mathbb{E}_{X \sim \mathcal{D}'}\left[\|P_i(X)\|_2^{2\ell_i}\right]}$$

$$= \sqrt{\mathbb{E}_{X \sim \mathcal{D}'}\left[\left(\sum_{j=1}^{n_i} P_{i,j}(X)^2\right)^{\ell_i}\right]},$$

where we used the Cauchy-Schwarz inequality in the first step. Continuing further by applying Jensen's inequality to the (convex) function $g(a) = a^{\ell_i}$, we get

$$\left(\sum_{j=1}^{n_i} P_{i,j}(X)^2\right)^{\ell_i} = n_i^{\ell_i}\left(\frac{1}{n_i}\sum_{j=1}^{n_i} P_{i,j}(X)^2\right)^{\ell_i} \leq n_i^{\ell_i}\frac{1}{n_i}\sum_{j=1}^{n_i} P_{i,j}(X)^{2\ell_i}.$$

Combining these two, we get

$$\mathbb{E}_{X \sim \mathcal{D}'}\left[\|P_i(X)\|_2^{\ell_i}\right] \leq \sqrt{n_i^{\ell_i}\frac{1}{n_i}\sum_{j=1}^{n_i}\mathbb{E}_{X \sim \mathcal{D}'}\left[P_{i,j}(X)^{2\ell_i}\right]}.$$

The next step is now different to [21] since we only have an approximately moment-matching distribution. By Fact 23, we get that

$$\mathbb{E}_{X \sim \mathcal{D}'}\left[P_{i,j}(X)^{2\ell_i}\right] \leq \mathbb{E}_{Y \sim \mathcal{N}(0, I_n)}\left[P_{i,j}(Y)^{2\ell_i}\right] + \eta n^{d\ell_i}$$

assuming that $k \geq 2d^2 m_i$ (this ensures that $2\ell_i = 2m_i|S| \leq 2m_i d$ is at most $k/d$ as required in Fact 23). This condition is slightly different to [21], but it does not change the (asymptotic) definition of $k$ as in (7). By Lemma 21, we now have that

$$\mathbb{E}_{Y \sim \mathcal{N}(0,I_n)} \left[ P_{i,j}(Y)^{2\ell_i} \right] \leq O_d \left( \sqrt{2\ell_i} \right)^{2\ell_i} = O_d \left( \sqrt{\ell_i} \right)^{2\ell_i}$$

for any $j$. Combining this with the above, we can conclude that

$$\mathbb{E}_{X \sim \mathcal{D}'} \left[ \|P_i(X)\|_2^{\ell_i} \right] \leq \sqrt{n_i}^{\ell_i} O_d \left( \sqrt{\ell_i} \right)^{\ell_i} + \sqrt{n_i^{\ell_i} \eta n^{d\ell_i}}$$
$$\leq O_d \left( \sqrt{n_i m_i |S|} \right)^{m_i|S|},$$

using that $\eta \leq \frac{1}{n^{d\ell_i}}$. This is true since $\ell_i = m_i|S| \leq m_i d \leq k$ and thus $\eta \leq \frac{1}{n^{dk}}$ implies $\eta \leq \frac{1}{n^{d\ell_i}}$. The rest of this proof is again the same as in [21, Proof of Proposition 8]. We can compute

$$\mathbb{E}_{X \sim \mathcal{D}'} \left[ |T(P(X)) - \tilde{F}(P(X))| \right]$$
$$\leq 2^d \max_{\emptyset \neq S \subseteq [d]} \left\{ \mathbb{E}_{X \sim \mathcal{D}'} \left[ \frac{1}{|S|} \sum_{i \in S} \left( \frac{C_i^{m_i} \|P_i(X)\|_2^{m_i}}{m_i!} \right)^{|S|} \right] \right\}$$
$$\leq 2^d \max_{\emptyset \neq S \subseteq [d]} \left\{ \max_{i \in S} \frac{C_i^{m_i|S|} O_d \left( \sqrt{n_i m_i |S|} \right)^{m_i|S|}}{m_i^{m_i|S|}} \right\}$$
$$\leq 2^d \max_{i \in [d], s \in [d]} \left\{ O_d \left( \frac{C_i \sqrt{n_i}}{\sqrt{m_i}} \right)^{m_i s} \right\}.$$

By choice of $m_i$, exactly as in [21], we can conclude that

$$\mathbb{E}_{X \sim \mathcal{D}'} \left[ |T(P(X)) - \tilde{f}(P(X))| \right] \leq 2^d \exp \left( -\min_{i \in [d]} m_i \right) \leq O(\varepsilon),$$

which completes the proof. $\qquad \square$

**Lemma** (Restatement of Lemma 27). *For any multi-index $\alpha = (\alpha_1, \ldots, \alpha_d) \in \mathbb{N}^{n_1} \times \cdots \times \mathbb{N}^{n_d}$, we have*

$$\partial^\alpha \tilde{F}(0) \leq \prod_{i=1}^d C_i^{|\alpha_i|}.$$

*Proof.* As mentioned earlier, the ideas of the following proof are based on [21, Proof of Lemmas 5 and 7]. We have that

$$|\partial^\alpha (F * \rho)| = |(F * \partial^\alpha \rho)(0)| \qquad \text{(property of convolution)}$$
$$\leq \|F\|_{L^\infty} \|\partial^\alpha \rho\|_{L^1}$$
$$\leq \|\partial^\alpha \rho\|_{L^1} \qquad (F \text{ maps to } [-1, 1]).$$

Thus, it remains to bound $\|\partial^\alpha \rho\|_{L^1}$. Using the product structure of $\rho$, we have that

$$\|\partial^\alpha \rho\|_{L^1} = \prod_{i=1}^n \|\partial^{\alpha_i} \rho_{C_i}\|_{L^1}. \tag{21}$$

First, we compute a bound for $\|\partial^{\alpha_i} \rho_2\|_{L^1}$. We generalize this afterwards to $\rho_{C_i}$. Recall that

$$B(\xi) = \begin{cases} 1 - \|\xi\|_2^2 & \text{if } \|\xi\|_2 \leq 1 \\ 0 & \text{else} \end{cases}$$

and

$$\rho_2(x) = \frac{|\hat{B}(x)|^2}{\|B\|_{L^2}^2},$$

where $\hat{B}$ is the Fourier transform of $B$. We first note that

$$\hat{B}(x) \cdot \overline{\hat{B}(x)} = |\hat{B}(x)|^2.$$

Thus, we can apply the product rule to get the following formula for the derivative

$$\partial^{\alpha_i} \rho_2 = \frac{1}{\|B\|_{L^2}^2} \sum_{\beta \leq \alpha_i} \binom{\alpha_i}{\beta} (\partial^\beta \hat{B})(\partial^{\alpha_i - \beta} \overline{\hat{B}})$$

$$= \frac{1}{\|B\|_{L^2}^2} \sum_{\beta \leq \alpha_i} \binom{\alpha_i}{\beta} (\partial^\beta \hat{B})(\overline{\partial^{\alpha_i - \beta} \hat{B}}),$$

where we used that the conjugate of the derivative is the derivative of the conjugate. We thus get the following by triangle inequality

$$\|(\partial^{\alpha_i} \rho_2)(x)\|_{L^1} \leq \frac{1}{\|B\|_{L^2}^2} \sum_{\beta \leq \alpha_i} \binom{\alpha_i}{\beta} \left\| (\partial^\beta \hat{B})(x) \overline{(\partial^{\alpha_i - \beta} \hat{B})(x)} \right\|_{L^1}.$$

We now want to analyze $(\partial^\beta \hat{B})(x)$. We have

$$(\partial^\beta \hat{B})(x) = \mathcal{F}(x^\beta B(x)),$$

where $\mathcal{F}(\cdot) = \hat{\cdot}$ stands for the Fourier transform and we used a fact about the derivative of the Fourier transform. Thus, we get furthermore that

$$\|(\partial^{\alpha_i} \rho_2)(x)\|_{L^1} \leq \frac{1}{\|B\|_{L^2}^2} \sum_{\beta \leq \alpha_i} \binom{\alpha_i}{\beta} \left\| \mathcal{F}(x^\beta B(x)) \overline{\mathcal{F}(x^{\alpha_i - \beta} B(x))} \right\|_{L^1}$$

$$= \frac{1}{\|B\|_{L^2}^2} \sum_{\beta \leq \alpha_i} \binom{\alpha_i}{\beta} \left\| x^\beta B(x) \overline{x^{\alpha_i - \beta} B(x)} \right\|_{L^1}$$

$$\leq \frac{1}{\|B\|_{L^2}^2} \sum_{\beta \leq \alpha_i} \binom{\alpha_i}{\beta} \left\| B(x) \overline{B(x)} \right\|_{L^1}$$

$$= \sum_{\beta \leq \alpha_i} \binom{\alpha_i}{\beta}$$

$$= 2^{|\alpha_i|}.$$

The second step uses the Parseval identity; the third step uses that $B(x) \neq 0$ implies that $\|x\|_\infty \leq \|x\|_2 \leq 1$; the fourth step uses that $\|B(x)\overline{B(x)}\|_{L^1} = \|B(x)\|_{L^2}^2$.

For an arbitrary $C_i$, we can bound $\|\partial^{\alpha_i} \rho_{C_i}\|_{L^1}$ as follows. Recall that

$$\rho_{C_i}(x) = \left(\frac{C_i}{2}\right)^n \rho_2\left(\frac{C_i}{2} x\right).$$

We can compute, using the chain rule that

$$(\partial^\alpha \rho_{C_i})(x) = \left(\frac{C_i}{2}\right)^n (\partial^\alpha \rho_2)\left(\frac{C_i}{2} x\right) \left(\frac{C_i}{2}\right)^{|\alpha_i|}.$$

To illustrate how the chain rule implies this, we can compute

$$\left(\frac{\partial}{\partial x_1} \rho_{C_i}\right)(x) = \left(\frac{C_i}{2}\right)^n \sum_{j=1}^n \left(\frac{\partial}{\partial x_j} \rho_2\right)\left(\frac{C_i}{2} x\right) \left(\frac{\partial}{\partial x_1}\left[x \mapsto \frac{C_i}{2} x_j\right]\right)(x)$$

$$= \left(\frac{C_i}{2}\right)^n \sum_{j=1}^n \left(\frac{\partial}{\partial x_j} \rho_2\right)\left(\frac{C_i}{2} x\right) \cdot \begin{cases} \frac{C_i}{2} & \text{if } j = 1 \\ 0 & \text{if } j \neq 1 \end{cases}$$

$$= \left(\frac{C_i}{2}\right)^{n+1} \left(\frac{\partial}{\partial x_1} \rho_2\right)\left(\frac{C_i}{2} x\right).$$

Doing this iteratively, we get the formula above. Now, we want to compute the $L^1$-norm of that function. We get

$$
\begin{aligned}
\|\partial^{\alpha_i}\rho_{C_i}\|_1 &= \int_{\mathbb{R}^n} (\partial^{\alpha_i}\rho_{C_i})(x)\,\mathrm{d}x \\
&= \left(\frac{C_i}{2}\right)^{n+|\alpha_i|} \int_{\mathbb{R}^n} (\partial^{\alpha_i}\rho_2)\left(\frac{C_i}{2}x\right)\mathrm{d}x \\
&= \left(\frac{C_i}{2}\right)^{|\alpha_i|} \int_{\mathbb{R}^n} (\partial^{\alpha_i}\rho_2)(y)\,\mathrm{d}y \qquad\qquad \left(y = \frac{C_i}{2}x\right) \\
&= \left(\frac{C_i}{2}\right)^{|\alpha_i|} \|\partial^{\alpha_i}\rho_2\|_1.
\end{aligned}
$$

Using the bound from above on $\|\partial^{\alpha_i}\rho_2\|_1$, we thus get

$$
\|\partial^{\alpha_i}\rho_{C_i}\|_1 \leq C_i^{|\alpha_i|}. \tag{22}
$$

Combining (21) and (22) completes the proof. $\qquad\square$

Next, we prove how we can generalize from

$$
\left|\mathbb{E}_{Y\sim\mathcal{N}(0,I_n)}[f(Y)] - \mathbb{E}_{X\sim\mathcal{D}'}\left[\tilde{f}(X)\right]\right| \leq O(\varepsilon)
$$

to the fooling condition we need

$$
\left|\mathbb{E}_{Y\sim\mathcal{N}(0,I_n)}[f(Y)] - \mathbb{E}_{X\sim\mathcal{D}'}[f(X)]\right| \leq O_d(\varepsilon).
$$

This proof is based on [21, Proof of Proposition 2]. It turns out that this part of [21] does not need that $X$ is $k$-Gaussian but works on the following, weaker assumptions.

**Lemma** (Restatement of Lemma 29). *Let $\varepsilon > 0$. Suppose that $\mathcal{D}'$ is a distribution such that the following holds*

- $\left|\mathbb{E}_{Y\sim\mathcal{N}(0,I_n)}[f(Y)] - \mathbb{E}_{X\sim\mathcal{D}'}\left[\tilde{f}(X)\right]\right| \leq O(\varepsilon)$, *and*

- $\mathbb{P}_{X\sim\mathcal{D}'}[\exists i,j : |P_{i,j}(X)| > B_i] \leq O(\varepsilon)$.

*Then, we have that*

$$
\left|\mathbb{E}_{Y\sim\mathcal{N}(0,I_n)}[f(Y)] - \mathbb{E}_{X\sim\mathcal{D}'}[f(X)]\right| \leq O_d(\varepsilon).
$$

In the proof of this lemma, we use the following theorem about anti-concentration of a polynomial of a Gaussian random variable. Importantly, it is enough to have this result for the Gaussian random variable $Y$ and we do not need it for the moment-matching distribution.

**Theorem 44** (Carbery and Wright, see [21, Theorem 13] or [7, Theorem 8]). *Let $p$ be a degree $d$ polynomial. Suppose that $\mathbb{E}_{Y\sim\mathcal{N}(0,I_n)}\left[p(Y)^2\right] = 1$. Then, for $\varepsilon > 0$,*

$$
\mathbb{P}_{Y\sim\mathcal{N}(0,I_n)}[|p(Y)| < \varepsilon] \leq O_d(d\varepsilon^{1/d}).
$$

*Proof of Lemma 29.* Note that [21, Lemma 12 and proof of Proposition 14] and the second condition in the lemma imply that

$$
\mathbb{E}[f(X)] \geq \mathbb{E}\left[\tilde{f}(X)\right] - O(\varepsilon) - 2\mathbb{P}\left[-O_d(\varepsilon^d) < p(X) < 0\right]
$$

and

$$
\mathbb{E}[f(X)] \leq \mathbb{E}\left[\tilde{f}(X)\right] + O(\varepsilon) + 2\mathbb{P}\left[0 < p(X) < O_d(\varepsilon^d)\right].
$$

Using these and the first assumption of the lemma we get that

$$
\mathbb{E}[f(X)] \geq \mathbb{E}[f(Y)] - O(\varepsilon) - 2\mathbb{P}\left[-O_d(\varepsilon^d) < p(X) < 0\right]
$$

and

$$
\mathbb{E}[f(X)] \leq \mathbb{E}[f(Y)] + O(\varepsilon) + 2\mathbb{P}\left[0 < p(X) < O_d(\varepsilon^d)\right].
$$

We have furthermore that

$$\mathbb{E}\left[\mathrm{sign}(p(X))\right] + 2\mathbb{P}\left[-O_d(\varepsilon^d) < p(X) < 0\right] = \mathbb{E}\left[\mathrm{sign}(p(X) + O_d(\varepsilon^d))\right]$$

and

$$\mathbb{E}\left[\mathrm{sign}(p(X))\right] - 2\mathbb{P}\left[0 < p(X) < O_d(\varepsilon^d)\right] = \mathbb{E}\left[\mathrm{sign}(p(X) - O_d(\varepsilon^d))\right].$$

The reason for this is that adding or subtracting the two probability terms can be interpreted as changing the sign for values of $X$ in $(-O_d(\varepsilon^d), 0)$ respectively $(0, O_d(\varepsilon^d))$, which the same as shifting the polynomial. Thus, when combining this with the above we get that

$$\mathbb{E}\left[\mathrm{sign}(p(X) + O_d(\varepsilon^d))\right] \geq \mathbb{E}\left[\mathrm{sign}(p(Y))\right] - O(\varepsilon)$$

and

$$\mathbb{E}\left[\mathrm{sign}(p(X) - O_d(\varepsilon^d))\right] \leq \mathbb{E}\left[\mathrm{sign}(p(Y))\right] + O(\varepsilon).$$

Now we apply this result not to the polynomial $p$, but to the polynomial $p \mp O_d(\varepsilon^d)$. This shifts the additional factor from the $X$-side to the $Y$-side and we get

$$\mathbb{E}\left[\mathrm{sign}(p(X))\right] \geq \mathbb{E}\left[\mathrm{sign}(p(Y) - O_d(\varepsilon^d))\right] - O(\varepsilon)$$

as well as

$$\mathbb{E}\left[\mathrm{sign}(p(X))\right] \leq \mathbb{E}\left[\mathrm{sign}(p(Y) + O_d(\varepsilon^d))\right] + O(\varepsilon).$$

Combining these two inequalities we get that

$$\mathbb{E}\left[\mathrm{sign}(p(Y) - O_d(\varepsilon^d))\right] - O(\varepsilon) \leq \mathbb{E}\left[f(X)\right] \leq \mathbb{E}\left[\mathrm{sign}(p(Y) + O_d(\varepsilon^d))\right] + O(\varepsilon).$$

For $Y$, we have that (since the inequality hold point-wise)

$$\mathbb{E}\left[\mathrm{sign}(p(Y) - O_d(\varepsilon^d))\right] \leq \mathbb{E}\left[f(Y)\right] \leq \mathbb{E}\left[\mathrm{sign}(p(Y) + O_d(\varepsilon^d))\right].$$

Now, the two function $\mathrm{sign}(p(Y) - O_d(\varepsilon^d))$ and $\mathrm{sign}(p(Y) + O_d(\varepsilon^d))$ differ by at most 2 and only when $|p(Y)| \leq O_d(\varepsilon^d)$. We now use an anti-concentration result for $Y$ (the standard Gaussian). Namely, we can use Theorem 44 to conclude that this happens with probability at most $O_d(\varepsilon)$. Note that we have $\mathbb{E}_{Y \sim \mathcal{N}(0, I_n)}\left[p(Y)^2\right] = 1$ since we assumed that the sum of the squares of the coefficients of $p$, which is exactly $\mathbb{E}_{Y \sim \mathcal{N}(0, I_n)}\left[p(Y)^2\right]$ for multilinear $p$, is 1. Thus,

$$\left|\mathbb{E}\left[\mathrm{sign}(p(Y) + O_d(\varepsilon^d))\right] - \mathbb{E}\left[\mathrm{sign}(p(Y) - O_d(\varepsilon^d))\right]\right| \leq 2O_d(\varepsilon) = O_d(\varepsilon).$$

Thus, since both $\mathbb{E}\left[f(X)\right]$ and $\mathbb{E}\left[f(Y)\right]$ are between the values $\mathbb{E}\left[\mathrm{sign}(p(Y) - O_d(\varepsilon^d))\right]$ and $\mathbb{E}\left[\mathrm{sign}(p(Y) + O_d(\varepsilon^d))\right]$ (up to $O(\varepsilon)$ for $\mathbb{E}\left[f(X)\right]$), we can conclude that also

$$\left|\mathbb{E}\left[f(X)\right] - \mathbb{E}\left[f(Y)\right]\right| \leq O_d(\varepsilon),$$

as wanted. $\qquad\square$

Next, we want to prove Proposition 30 that follows closely [21, Proof of Theorem 1]. This proposition shows the fooling condition for arbitrary PTFs. In the proof we need Proposition 20 about the fooling condition for multilinear PTFs and Lemma 31 that, given an arbitrary PTF $p$, constructs a multilinear PTF $p_\delta$ that is close to $p$.

**Proposition** (Restatement of Proposition 30). *Let $\varepsilon > 0$. Suppose that $\mathcal{D}'$ approximately matches the moments of $\mathcal{N}(0, I_n)$ up to degree $k$ and slack $\eta$, where $k \geq \Omega_d\left(\varepsilon^{-4d \cdot 7^d}\right)$, and $\eta \leq n^{-\Omega_d(k)} k^{-\Omega_d(k)}$. Then, we have that for any $f \in \mathcal{F}_{\mathrm{PTF},d}$*

$$\left|\mathbb{E}_{Y \sim \mathcal{N}(0, I_n)}\left[f(Y)\right] - \mathbb{E}_{X \sim \mathcal{D}'}\left[f(X)\right]\right| \leq O_d(\varepsilon).$$

*Proof.* As already before, we assume without loss of generality that the polynomial is normalized in the sense that the sum of the squares of the coefficients is 1 (this does not change the PTF). We set

$$\delta = \left(\frac{\varepsilon}{d}\right)^d.$$

We first want to show that the condition on $\eta$ in this lemma ensures that we can apply both Lemma 31 to construct the multilinear polynomial $p_\delta$ and Proposition 20 to get the fooling condition for $p_\delta$.

The condition for Lemma 31 is $\eta \le \delta^{O_d(1)} n^{-\Omega_d(1)} k^{-k}$. Note that by our choice of $k$, we have $\varepsilon^{O_d(1)} \le k^{-\Omega_d(1)}$. Thus, for our choice of $\delta$, the condition on $\eta$ needed for Lemma 31 is satisfied by our assumption on $\eta$ in this lemma. For Proposition 20, we need $\hat{\eta} = (2k)^{k/2}\eta \le k^{-\Omega_d(k)} n^{-\Omega_d(k)}$, which is also satisfied by our condition on $\eta$. Note that we need this condition for $\hat{\eta}$ (and not $\eta$) since the new random variable $\hat{X}$ is only moment-matching up to slack $\hat{\eta}$.

We now want to show that $|\mathbb{P}[p(X) > 0] - \mathbb{P}[p(Y) > 0]| \le O_d(\varepsilon)$. Note that
$$\mathbb{E}[\text{sign}(p(X))] = \mathbb{P}[p(X) \ge 0] - \mathbb{P}[p(X) < 0]$$
$$= \mathbb{P}[p(X) \ge 0] - (1 - \mathbb{P}[p(X) \ge 0])$$
$$= 2\mathbb{P}[p(X) \ge 0] - 1$$
and the same holds for $Y$. Thus, $|\mathbb{P}[p(X) \ge 0] - \mathbb{P}[p(Y) \ge 0]| \le O_d(\varepsilon)$ will be enough since then
$$|\mathbb{E}[\text{sign}(p(Y))] - \mathbb{E}[\text{sign}(p(X))]| = 2|\mathbb{P}[p(Y) \ge 0] - \mathbb{P}[p(X) \ge 0]| \le O_d(\varepsilon).$$
By Lemma 31, we have that
$$\mathbb{P}[p(X) \ge 0] \ge \mathbb{P}\left[p_\delta(\hat{X}) \ge \delta\right] - \delta.$$
Since $p_\delta$ is multilinear, we can apply Proposition 20 to $p_\delta - \delta$ to get that
$$\mathbb{P}[p(X) \ge 0] \ge \mathbb{P}\left[p_\delta(\hat{Y}) \ge \delta\right] - O_d(\varepsilon)$$
since $\delta \le O(\varepsilon)$. Note that we can apply Proposition 20 to $\mathbb{P}[p(X) \ge 0]$ instead of $\mathbb{E}[\text{sign}(p(X))]$ by the relation $\mathbb{E}[\text{sign}(p(X))] = 2\mathbb{P}[p(X) \ge 0] - 1$ from above. By Carbery-Wright (Theorem 44), we get that $\mathbb{P}\left[|p_\delta(\hat{Y})| \le \delta\right] \le O(\varepsilon)$, thus we further have that
$$\mathbb{P}[p(X) \ge 0] \ge \mathbb{P}\left[p_\delta(\hat{Y}) \ge -\delta\right] - O_d(\varepsilon).$$
Applying again Lemma 31, we get that
$$\mathbb{P}[p(X) \ge 0] \ge \mathbb{P}[p(Y) \ge 0] - O_d(\varepsilon).$$
By an analogous calculation, we get that $\mathbb{P}[p(X) \ge 0] \le \mathbb{P}[p(Y) \ge 0] + O_d(\varepsilon)$, which completes the proof. $\qquad\square$

We now want to prove Lemmas 32 and 33 that show that the construction of $\hat{X}$ respectively $\hat{Y}$ preserve the assumptions on the distribution. The latter lemma is also proved in [21, Lemma 15], but we include it here for completeness.

**Lemma** (Restatement of Lemma 32). *Suppose $X$ approximately matches the moments of $\mathcal{N}(0, I_n)$ up to degree $k$ and slack $\eta$. Then, $\hat{X}$ approximately matches the moments of $\mathcal{N}(0, I_{nN})$ up to degree $k$ and slack $\hat{\eta} = (2k)^{k/2}\eta$.*

*Proof.* For a multi-index $\alpha$, let $c_\alpha$ be the $\alpha$-moment of a Gaussian. We want to show that $\left|\mathbb{E}\left[\hat{X}^\alpha\right] - c_\alpha\right| \le \hat{\eta}$ for any $\alpha$ with $|\alpha| \le k$. We can compute the following, by writing $Z_{i,j} = Z_j^{(i)}$,

$$\left|\mathbb{E}\left[\hat{X}^\alpha\right] - c_\alpha\right| = \left|\mathbb{E}\left[\prod_{i=1}^n \prod_{j=1}^N \hat{X}_{i,j}^{\alpha_{i,j}}\right] - c_\alpha\right|$$

$$= \left|\mathbb{E}\left[\prod_{i=1}^n \prod_{j=1}^N \left(\frac{1}{\sqrt{N}}X_i + Z_{i,j}\right)^{\alpha_{i,j}}\right] - c_\alpha\right|$$

$$= \left|\mathbb{E}\left[\prod_{i=1}^n \prod_{j=1}^N \sum_{r=0}^{\alpha_{i,j}} \binom{\alpha_{i,j}}{r}\left(\frac{1}{\sqrt{N}}X_i\right)^r Z_{i,j}^{\alpha_{i,j}-r}\right] - c_\alpha\right|$$

$$= \left|\mathbb{E}\left[\sum_{\beta \le \alpha} \binom{\alpha}{\beta}\left(\frac{X}{\sqrt{N}}\right)^\beta Z^{\alpha-\beta}\right] - c_\alpha\right|$$

$$= \left|\sum_{\beta \le \alpha} \binom{\alpha}{\beta}\mathbb{E}\left[\left(\frac{X}{\sqrt{N}}\right)^\beta\right]\mathbb{E}\left[Z^{\alpha-\beta}\right] - c_\alpha\right|.$$

In the second step we used the definition of $\hat{X}$ and in the last step, we used that $Z$ and $X$ are independent. Now,

$$\left| \mathbb{E}\left[\left(\frac{X}{\sqrt{N}}\right)^{\beta}\right] - \mathbb{E}\left[\left(\frac{Y}{\sqrt{N}}\right)^{\beta}\right] \right| \leq \eta$$

since $X$ matches the moments of $\mathcal{N}(0, I_n)$ up to degree $k$ and slack $\eta$. Thus, we get

$$\left| \mathbb{E}\left[\hat{X}^{\alpha}\right] - c_{\alpha} \right| = \left| \sum_{\beta \leq \alpha} \binom{\alpha}{\beta} \mathbb{E}\left[\left(\frac{X}{\sqrt{N}}\right)^{\beta}\right] \mathbb{E}\left[Z^{\alpha-\beta}\right] - c_{\alpha} \right|$$

$$\leq \left| \sum_{\beta \leq \alpha} \binom{\alpha}{\beta} \left(\mathbb{E}\left[\left(\frac{X}{\sqrt{N}}\right)^{\beta}\right] - \mathbb{E}\left[\left(\frac{Y}{\sqrt{N}}\right)^{\beta}\right]\right) \mathbb{E}\left[Z^{\alpha-\beta}\right] \right|$$

$$+ \left| \sum_{\beta \leq \alpha} \binom{\alpha}{\beta} \mathbb{E}\left[\left(\frac{Y}{\sqrt{N}}\right)^{\beta}\right] \mathbb{E}\left[Z^{\alpha-\beta}\right] - c_{\alpha} \right|$$

$$\leq \eta \sum_{\beta \leq \alpha} \binom{\alpha}{\beta} \left| \mathbb{E}\left[Z^{\alpha-\beta}\right] \right| + \left| \sum_{\beta \leq \alpha} \binom{\alpha}{\beta} \mathbb{E}\left[\left(\frac{Y}{\sqrt{N}}\right)^{\beta}\right] \mathbb{E}\left[Z^{\alpha-\beta}\right] - c_{\alpha} \right|$$

$$= \eta \sum_{\beta \leq \alpha} \binom{\alpha}{\beta} \left| \mathbb{E}\left[Z^{\alpha-\beta}\right] \right| + \left| \mathbb{E}\left[Y^{\alpha}\right] - c_{\alpha} \right|$$

$$= \eta \sum_{\beta \leq \alpha} \binom{\alpha}{\beta} \left| \mathbb{E}\left[Z^{\alpha-\beta}\right] \right|.$$

The second-to-last step is the same computation from above but backwards (note that the moments of $Z'$ used for the construction of $\hat{Y}$ are the same as those of $Z$ used for the construction of $\hat{X}$) and the last step used that $Y$ is Gaussian and thus $\mathbb{E}[Y^{\alpha}] = c_{\alpha}$. We can furthermore compute

$$\left| \mathbb{E}\left[\hat{X}^{\alpha}\right] - c_{\alpha} \right| = \eta \sum_{\beta \leq \alpha} \binom{\alpha}{\beta} \left| \mathbb{E}\left[Z^{\alpha-\beta}\right] \right|$$

$$= \eta \sum_{\beta \leq \alpha} \binom{\alpha}{\beta} \left| \mathbb{E}\left[Z^{\beta}\right] \right|$$

$$= \eta \sum_{\beta \leq \alpha} \binom{\alpha}{\beta} \prod_{i=1}^{n} \left| \mathbb{E}\left[\prod_{j=1}^{N} Z_{i,j}^{\beta_{i,j}}\right] \right|,$$

where the third step uses that the $Z_{i,j}$ are independent for different $i$. We now get that, using Fact 42,

$$\left| \mathbb{E}\left[\prod_{j=1}^{N} Z_{i,j}^{\beta_{i,j}}\right] \right| \leq |\beta_{i,\cdot}|^{|\beta_{i,\cdot}|/2} \leq \sqrt{k}^{|\beta_{i,\cdot}|}.$$

Thus, continuing with the above, we get

$$\left| \mathbb{E}\left[\hat{X}^{\alpha}\right] - c_{\alpha} \right| = \eta \sum_{\beta \leq \alpha} \binom{\alpha}{\beta} \prod_{i=1}^{n} \left| \mathbb{E}\left[\prod_{i=1}^{N} Z_{i,j}^{\beta_{i,j}}\right] \right|$$

$$\leq \eta \sum_{\beta \leq \alpha} \binom{\alpha}{\beta} \prod_{i=1}^{n} \sqrt{k}^{|\beta_{i,\cdot}|}$$

$$= \eta \sum_{\beta \leq \alpha} \binom{\beta}{\alpha} (\sqrt{k}\mathbb{1})^{\beta}$$

$$= \eta(\sqrt{k} + 1)^{|\alpha|}$$

$$\leq (2k)^{k/2}\eta = \hat{\eta},$$

where $\mathbb{1}$ is the all-ones vector. This completes the proof. $\qquad\square$

**Lemma** (Restatement of Lemma 33). *If $Y \sim \mathcal{N}(0, I_n)$, then $\hat{Y} \sim \mathcal{N}(0, I_{nN})$.*

*Proof.* Since $\hat{Y}$ is a linear transformation of the Gaussian vector $(Y, Z')$, it is jointly Gaussian and thus to show the lemma, it is enough to show that the expectation of $\hat{Y}$ is 0 and the covariance matrix is the identity.

Writing as above $Z'_{i,j} = Z'^{(i)}_j$, we have for any $i \in [n]$, $j \in [N]$ that

$$\mathbb{E}\left[\hat{Y}_{i,j}\right] = \mathbb{E}\left[\frac{1}{\sqrt{N}}Y_i + Z'_{i_j}\right] = \frac{1}{\sqrt{N}} \cdot 0 + 0 = 0.$$

Furthermore, for any $i \in [n]$, $j \in [N]$ we have that, by independence of $Y$ and $Z'$,

$$\text{Var}\left[\hat{Y}_{i,j}\right] = \frac{1}{N}\text{Var}\left[Y_i\right] + \text{Var}\left[Z'_{i,j}\right] = \frac{1}{N} \cdot 1 + \left(1 - \frac{1}{N}\right) = 1.$$

For any $i \in [n]$, $j_1 \neq j_2 \in [N]$ we get that

$$\text{Cov}\left[\hat{Y}_{i,j_1}, \hat{Y}_{i,j_2}\right] = \mathbb{E}\left[\left(\frac{1}{\sqrt{N}}Y_i + Z'_{i,j_1}\right)\left(\frac{1}{\sqrt{N}}Y_i + Z'_{i,j_2}\right)\right]$$

$$= \mathbb{E}\left[\frac{1}{N}Y_i^2\right] + \mathbb{E}\left[\frac{1}{\sqrt{N}}Y_i Z'_{i,j_1}\right] + \mathbb{E}\left[\frac{1}{\sqrt{N}}Y_i Z'_{i,j_2}\right] + \mathbb{E}\left[Z'_{i,j_1}Z'_{i,j_2}\right]$$

$$= \frac{1}{N} + 0 + 0 - \frac{1}{N}$$

$$= 0.$$

Finally, for any $i_1 \neq i_2 \in [n]$, $j_1, j_2 \in [N]$ we have that

$$\text{Cov}\left[\hat{Y}_{i_1,j_1}, \hat{Y}_{i_2,j_2}\right] = 0$$

by independence of $\hat{Y}_{i_1,j_1} = Y_{i_1} + Z'_{i_1,j_1}$ and $\hat{Y}_{i_2,j_2} = Y_{i_2} + Z'_{i_2,j_2}$ for $i_1 \neq i_2$.

Thus, as argued in the beginning, this shows that $\hat{Y} \sim \mathcal{N}(0, I_{nN})$ and completes the proof. $\qquad\square$

Finally, we want to provide the details of the missing parts in the proof of Lemma 31 about the existence of the polynomial $p_\delta$. For this, it remains to prove Lemmas 34 and 35, which we restate below. First, recall our definition of $\delta'$ in (13).

$$\delta' := \min\left\{\frac{\delta}{2dn}, \Theta_d\left(\frac{\delta^{d+1}}{n^{3d/2}}\right)\right\} = \Theta_d\left(\frac{\delta^{d+1}}{n^{3d/2}}\right). \qquad\text{(restatement of 13)}$$

In the proof of Lemma 31, we then used the following lemma that allows to replace a factor $X_i^\ell$ by a multilinear polynomial in the new variable $\hat{X}$.

**Lemma** (Restatement of Lemma 34). *Let $\delta' > 0$. Let $i \in [n]$ and $\ell \in [d]$. Assume that each of the following conditions holds with probability $1 - \frac{\delta'}{d}$*

$$\left|\sum_{j=1}^{N} \frac{\hat{X}_{i,j}}{\sqrt{N}}\right| \leq \frac{1}{\delta'} \qquad\text{(restatement of 10)}$$

$$\left|\sum_{j=1}^{N} \left(\frac{\hat{X}_{i,j}}{\sqrt{N}}\right)^2 - 1\right| \leq \delta'^{d+1} \qquad\text{(restatement of 11)}$$

$$\left|\sum_{j=1}^{N} \left(\frac{\hat{X}_{i,j}}{\sqrt{N}}\right)^a\right| \leq \delta'^{d+1} \qquad \forall\, 3 \leq a \leq d. \qquad\text{(restatement of 12)}$$

*Then, there is a multilinear polynomial in $\hat{X}_{i,j}$ that is within $O_d(\delta')$ of $X_i^\ell$ with probability $1 - \delta'$.*

We prove this lemma below. Before that, we want to prove Lemma 35 that states that we can use the above lemma for our choice of $\delta'$.

**Lemma** (Restatement of Lemma 35). *Let $\delta'$ as in* (13)*. Assuming $\eta \leq \delta^{O_d(1)} n^{-\Omega_d(1)} k^{-k}$, there is a choice of $N$ (independent of $i$ or $\ell$) such that* (10)*,* (11) *and* (12) *hold, each with probability $1 - \frac{\delta'}{d}$.*

The proof of this lemma is a combination of the following lemmas.

**Lemma 45.** *Assuming*

$$\eta \leq \frac{\frac{1}{\delta'd} - 1}{N(2k)^{k/2}},$$

*we have with probability $1 - \frac{\delta'}{d}$ that*

$$\left| \sum_{j=1}^{N} \frac{\hat{X}_{i,j}}{\sqrt{N}} \right| \leq \frac{1}{\delta'}.$$

**Lemma 46.** *Assuming*

$$\eta \leq \frac{\frac{\delta'^{2d+3}}{d} - \frac{2}{N}}{3(2k)^{k/2}},$$

*we have with probability $1 - \frac{\delta'}{d}$ that*

$$\left| \sum_{j=1}^{N} \left( \frac{\hat{X}_{i,j}}{\sqrt{N}} \right)^2 - 1 \right| \leq \delta'^{d+1}.$$

Note that for the above condition on $\eta$ to be meaningful (we need $\eta > 0$), we also need to ensure that

$$N > \frac{2d}{\delta'^{2d+3}}.$$

**Lemma 47.** *Assuming*

$$\eta \leq \frac{1}{(2k)^{k/2}}$$

*and*

$$N \geq \frac{100d^2}{\delta'^{2d+3}}$$

*we have for any $3 \leq a \leq d$ with probability $1 - \frac{\delta'}{d}$ that*

$$\left| \sum_{j=1}^{N} \left( \frac{\hat{X}_{i,j}}{\sqrt{N}} \right)^a \right| \leq \delta'^{d+1}.$$

Using these three lemmas, we are now able to prove Lemma 35.

*Proof of Lemma 35.* It remains to argue that there is a choice of $N$ such that the condition on $\eta$ in the lemma statement ensures that the conditions on $\eta$ in Lemmas 45, 46 and 47 are satisfied. We argue below that the following choice of $N$ is enough, which will complete the proof,

$$N := \Theta_d \left( \frac{1}{\delta'^{2d+3}} \right) = \Theta_d \left( \frac{n^{3d(2d+3)/2}}{\delta^{(d+1)(2d+3)}} \right). \tag{23}$$

For Lemma 45, we need $\eta \leq \frac{\frac{1}{\delta'd} - 1}{N(2k)^{k/2}}$. Plugging in the value of $N$ and $\delta'$, it is enough for $\eta$ to satisfy (note that $\frac{1}{\delta'd} - 1 \geq 1$ since $\delta'$ is small) $\eta \leq O_d \left( \frac{\delta^{O_d(1)}}{n^{\Omega_d(1)}(2k)^{k/2}} \right)$, which holds since by assumption $\eta \leq \delta^{O_d(1)} n^{-\Omega_d(1)} k^{-k}$.

For Lemma 46, we need $\eta \leq \frac{\frac{\delta'^{2d+3}}{d} - \frac{2}{N}}{3(2k)^{k/2}}$ and $N > \frac{2d}{\delta'^{2d+3}}$. The latter is clearly satisfied by the choice of $N$ as in (23). For the former, plugging in again the values of $N$ and $\delta'$, it is enough for $\eta$ to satisfy $\eta \leq O_d\left(\frac{\delta^{O_d(1)}}{n^{\Omega_d(1)}(2k)^{k/2}}\right)$, which again holds since $\eta \leq \delta^{O_d(1)}n^{-\Omega_d(1)}k^{-k}$.

For Lemma 47, we need $N \geq \frac{100d^2}{\delta'^{2d+3}}$ and $\eta \leq \frac{1}{(2k)^{k/2}}$. The former is again directly satisfied by the choice of $N$. Furthermore, also the latter is true, again since we assume $\eta \leq \delta^{O_d(1)}n^{-\Omega_d(1)}k^{-k}$. $\qquad\square$

Next, we want to prove Lemmas 45, 46 and 47.

*Proof of Lemma 45.* Using Markov's inequality, we get that

$$\mathbb{P}\left[\left|\sum_{j=1}^N \frac{\hat{X}_{i,j}}{\sqrt{N}}\right| > \frac{1}{\delta'}\right] \leq \frac{\mathbb{E}\left[\frac{1}{N}\left(\sum_{j=1}^N \hat{X}_{i,j}\right)^2\right]}{\left(\frac{1}{\delta'}\right)^2}.$$

Since, $\hat{X}$ approximates the moments of $\mathcal{N}(0, I_{nN})$ up to degree $k$ and error $\hat{\eta}$, we can compute the above expectation as follows

$$\mathbb{E}\left[\left(\sum_{j=1}^N \hat{X}_{i,j}\right)^2\right] = \sum_{|\alpha|=2}\binom{2}{\alpha}\mathbb{E}\left[\hat{X}_{i,\cdot}^\alpha\right] \leq \hat{\eta}N(N-1) + (1+\hat{\eta})N = \hat{\eta}N^2 + N.$$

Here, we used that the expectation $\mathbb{E}\left[\hat{X}_{i,\cdot}^\alpha\right]$ is at most $\hat{\eta}$ if some $\alpha_j = 1$ (for such $\alpha$ the expectation of $\mathcal{N}(0, I_{nN})$ is 0) and at most $1 + \hat{\eta}$ otherwise (since the expectation of $\mathcal{N}(0, I_{nN})$ is 1 for such $\alpha$). Thus, we get that

$$\mathbb{P}\left[\left|\sum_{j=1}^N \frac{\hat{X}_{i,j}}{\sqrt{N}}\right| > \frac{1}{\delta'}\right] \leq (\hat{\eta}N + 1)\delta'^2 \leq \frac{\delta'}{d}$$

since by assumption we have that $\hat{\eta} = (2k)^{k/2}\eta \leq \frac{\frac{1}{\delta'd}-1}{N}$. $\qquad\square$

*Proof of Lemma 46.* We again use Markov's inequality to get

$$\mathbb{P}\left[\left|\frac{1}{N}\left(\sum_{j=1}^N \hat{X}_{i,j}^2\right) - 1\right| > \delta'^{d+1}\right] \leq \frac{\mathbb{E}\left[\left(\frac{1}{N}\left(\sum_{j=1}^N \hat{X}_{i,j}^2\right) - 1\right)^2\right]}{\delta'^{2d+2}}.$$

We are thus interested in $\mathbb{E}\left[\left(\sum_{j=1}^N \left(\hat{X}_{i,j}^2 - 1\right)\right)^2\right]$. We can expand this as follows

$$\mathbb{E}\left[\left(\sum_{j=1}^N \left(\hat{X}_{i,j}^2 - 1\right)\right)^2\right] = \sum_{|\alpha|=2}\binom{2}{\alpha}\mathbb{E}\left[\left(\hat{X}_{i,\cdot}^2 - 1\right)^\alpha\right].$$

If some $\alpha_j = 1$, then the expectation will be, for some $j_1 \neq j_2 \in [N]$,

$$\mathbb{E}\left[\left(\hat{X}_{i,j_1}^2 - 1\right)\left(\hat{X}_{i,j_2}^2 - 1\right)\right] = \mathbb{E}\left[\hat{X}_{i,j_1}^2\hat{X}_{i,j_2}^2\right] - \mathbb{E}\left[\hat{X}_{i,j_1}^2\right] - \mathbb{E}\left[\hat{X}_{i,j_2}^2\right] + 1$$
$$\leq 1 + \hat{\eta} - (1 - \hat{\eta}) - (1 - \hat{\eta}) + 1$$
$$= 3\hat{\eta}.$$

As for the proof of Lemma 45, this is true since the corresponding Gaussian moments are all 1. If no $\alpha_j = 1$, we get

$$\mathbb{E}\left[\left(\hat{X}_{i,j}^2 - 1\right)^2\right] = \mathbb{E}\left[\hat{X}_{i,j}^4\right] - 2\mathbb{E}\left[\hat{X}_{i,j}^2\right] + 1$$
$$\leq 3 + \hat{\eta} - 2(1 - \hat{\eta}) + 1$$
$$= 2 + 2\hat{\eta},$$

since the second and fourth moment of a Gaussian are 1 and 3 respectively. Summarized we get that

$$\mathbb{E}\left[\left(\sum_{j=1}^{N}\left(\hat{X}_{i,j}^{2}-1\right)\right)^{2}\right] \leq N(2+2\hat{\eta}) + N(N-1)3\hat{\eta}$$

$$= 2N - N\hat{\eta} + 3N^{2}\hat{\eta}.$$

Together with the above we get that

$$\mathbb{P}\left[\left|\frac{1}{N}\left(\sum_{j=1}^{N}\hat{X}_{i,j}^{2}\right) - 1\right| > \delta'^{d+1}\right] \leq \frac{2N - N\hat{\eta} + 3N^{2}\hat{\eta}}{N^{2}\delta'^{2d+2}}$$

$$\leq \frac{\frac{2}{N} + 3\hat{\eta}}{\delta'^{2d+2}}$$

$$\leq \frac{\delta'}{d},$$

since by assumption we have that $\hat{\eta} = (2k)^{k/2}\eta \leq \frac{\frac{\delta'^{2d+3}}{d} - \frac{2}{N}}{3}$. $\qquad\square$

*Proof of Lemma 47.* For any $3 \leq a \leq d$, similar to before, we compute using Markov's inequality

$$\mathbb{P}\left[\left|\frac{1}{N^{a/2}}\sum_{j=1}^{N}\hat{X}_{i,j}^{a}\right| > \delta'^{d+1}\right] \leq \frac{\mathbb{E}\left[\left(\frac{1}{N^{a/2}}\sum_{j=1}^{N}\hat{X}_{i,j}^{a}\right)^{2}\right]}{\delta'^{2d+2}}.$$

We thus need to bound $\mathbb{E}\left[\left(\sum_{j=1}^{N}\hat{X}_{i,j}^{a}\right)^{2}\right]$. We get

$$\mathbb{E}\left[\left(\sum_{j=1}^{N}\hat{X}_{i,j}^{a}\right)^{2}\right] = \sum_{|\alpha|=2}\binom{2}{\alpha}\mathbb{E}\left[\hat{X}_{i,\cdot}^{a\cdot\alpha}\right] \leq N^{2}(2a)^{a}$$

since for any $\alpha$ we have $\mathbb{E}\left[\hat{X}_{i,\cdot}^{a\cdot\alpha}\right] \leq (2a)^{a}$ by Fact 41, since by assumption we have $\eta \leq \frac{1}{(2k)^{k/2}}$ or in other words $\hat{\eta} \leq 1$. Combining this with the above, we get

$$\mathbb{P}\left[\left|\frac{1}{N^{a/2}}\sum_{j=1}^{N}\hat{X}_{i,j}^{a}\right| > \delta'^{d+1}\right] \leq \frac{N^{2}(2a)^{a}}{N^{a}\delta'^{2d+2}} = \frac{(2a)^{a}}{N^{a-2}\delta'^{2d+2}}.$$

We have for any $a \geq 3$ that $((2a)^{a})^{\frac{1}{a-2}} \leq 100a$ and thus we get that

$$\mathbb{P}\left[\left|\frac{1}{N^{a/2}}\sum_{j=1}^{N}\hat{X}_{i,j}^{a}\right| > \delta'^{d+1}\right] \leq \frac{(100a)^{a-2}}{N^{a-2}\delta'^{2d+2}}$$

$$= \left(\frac{100a}{N}\right)^{a-2}\frac{1}{\delta'^{2d+2}}$$

$$\leq \left(\frac{\delta'^{2d+3}}{d}\right)^{a-2}\frac{1}{\delta'^{2d+2}}$$

$$\leq \frac{\delta'}{d}$$

In the third step we used that by assumption we have that $N \geq \frac{100d^{2}}{\delta'^{2d+3}}$ and $d \geq a$. In the last step, we then used that $a - 2 \geq 1$, together with $\frac{\delta'^{2d+3}}{d} \leq 1$. $\qquad\square$

Finally, it remains to prove Lemma 34.

*Proof of Lemma 34.* By the assumptions, the conditions (10), (11) and (12) hold simultaneously with probability $1 - d\frac{\delta'}{d} = 1 - \delta'$. We want to construct a multilinear polynomial in $\hat{X}_{i,j}$ such that conditioned on (10), (11) and (12) it is within $O_d(\delta')$ of $X_i^\ell$. This will complete the proof.

The construction follows [21, Proof of Lemma 15]. First, note that by construction we have

$$X_i^\ell = N^{-\ell/2} \left( \sum_{j=1}^N \hat{X}_{i,j} \right)^\ell.$$

Expanding the sum and grouping the terms according together according how the power of $\ell$ is partitioned to different $\hat{X}_{i,j}$, we get

$$X_i^\ell = \sum_{r=1}^\ell \sum_{\substack{1 \leq a_1 \leq \cdots \leq a_r \\ \sum a_s = \ell}} c(a_1, \ldots, a_r) \sum_{\substack{j_1, \ldots, j_r \in [N] \\ \text{distinct}}} \prod_{s=1}^r \left( \frac{\hat{X}_{i,j_s}}{\sqrt{N}} \right)^{a_s},$$

where the $c(a_1, \ldots, a_r)$ are constants that capture how often the latter terms occur if we multiply the product out. More precisely,

$$c(a_1, \ldots, a_r) = \binom{\ell}{a_1, \ldots, a_r} \prod_{t=1}^\ell \frac{1}{|\{s : a_s = t\}|}.$$

The strategy is now as follows. We want to approximate the terms

$$\sum_{\substack{j_1, \ldots, j_r \in [N] \\ \text{distinct}}} \prod_{s=1}^r \left( \frac{\hat{X}_{i,j_s}}{\sqrt{N}} \right)^{a_s}$$

separately by multilinear polynomials. If $a_r = 1$, then the terms is already multilinear and there is nothing to do. Note that if $\ell = 1$, then we only have this case, so from now on we assume $\ell \geq 2$. If $a_r = 2$, then we want to show that

$$\left| \sum_{\substack{j_1, \ldots, j_r \in [N] \\ \text{distinct}}} \prod_{s=1}^r \left( \frac{\hat{X}_{i,j_s}}{\sqrt{N}} \right)^{a_s} - \sum_{\substack{j_1, \ldots, j_{\hat{r}} \in [N] \\ \text{distinct}}} \prod_{s=1}^{\hat{r}} \frac{\hat{X}_{i,j_s}}{\sqrt{N}} \right| \leq O_d(\delta'), \qquad (24)$$

where here and also later $\hat{r}$ is the largest $s \in [r]$ such that $a_s = 1$ (we assume here and throughout this proof that in case no $a_s = 1$, i.e. we have an empty sum and an empty product, the second term on the LHS is 1; the reason for this is that, intuitively, we want to make the term multilinear by removing all powers higher than 1, which leaves 1 in case no $a_s = 1$). If $a_r \geq 3$, then we want to show that

$$\left| \sum_{\substack{j_1, \ldots, j_r \in [N] \\ \text{distinct}}} \prod_{s=1}^r \left( \frac{\hat{X}_{i,j_s}}{\sqrt{N}} \right)^{a_s} \right| \leq O_d(\delta'). \qquad (25)$$

Once we have shown these, the idea is to remove all terms with $a_r \geq 3$ and replace the terms for $a_r = 2$ by the multilinear term on the LHS of (24). We get that $X_i^\ell$ is within $O_d(\delta')$ of

$$\sum_{r=1}^\ell \sum_{\substack{1 \leq a_1 \leq \cdots \leq a_r \\ \sum a_s = \ell \\ a_r \leq 2}} c(a_1, \ldots, a_r) \sum_{\substack{j_1, \ldots, j_{\hat{r}} \in [N] \\ \text{distinct}}} \prod_{s=1}^{\hat{r}} \frac{\hat{X}_{i,j_s}}{\sqrt{N}},$$

which is multilinear. The $O_d$ here directly also covers that we need to multiply the $O_d(\delta')$ from above with the constants $c(a_1, \ldots, a_r)$ and then sum over the choices of $a_1, \ldots, a_r$ and over $r$. This then completes the proof.

Thus, it remains to prove (24) and (25). We do this using the assumptions of the lemma and by induction on $r$ (and technically also over $\ell$; the base case for $\ell = 1$ was already covered above). We also inductively show that

$$\left| \sum_{\substack{j_1, \ldots, j_{\hat{r}} \in [N] \\ \text{distinct}}} \prod_{s=1}^{\hat{r}} \frac{\hat{X}_{i,j_s}}{\sqrt{N}} \right| \leq O_d \left( \left( \frac{1}{\delta'} \right)^r \right). \tag{26}$$

This will be needed to prove (24) and (25).

**Base case $r = 1$.** If $r = 1$, then

$$\sum_{\substack{j_1, \ldots, j_r \in [N] \\ \text{distinct}}} \prod_{s=1}^{r} \left( \frac{\hat{X}_{i,j_s}}{\sqrt{N}} \right)^{a_s} = \sum_{j_1} \left( \frac{\hat{X}_{i,j_1}}{\sqrt{N}} \right)^{\ell}.$$

In this case, we have $a_r = \ell$. If $\ell = 2$, we need to show (24) and this follows directly from (11) (since $\delta'^{d+1} \leq O_d(\delta')$). If $\ell \geq 3$, then we need to show (25), which follows again directly from (12). Also note that (26) holds since $1 \leq O_d \left( \frac{1}{\delta'} \right)$.

**Induction step.** We now assume that $r \geq 2$ and that we have proven the result for all values smaller than $r$. The goal is to now show the result for $r$. We compute

$$\sum_{\substack{j_1, \ldots, j_r \in [N] \\ \text{distinct}}} \prod_{s=1}^{r} \left( \frac{\hat{X}_{i,j_s}}{\sqrt{N}} \right)^{a_s}$$

$$= \left( \sum_{\substack{j_1, \ldots, j_{r-1} \in [N] \\ \text{distinct}}} \prod_{s=1}^{r-1} \left( \frac{\hat{X}_{i,j_s}}{\sqrt{N}} \right)^{a_s} \right) \left( \sum_{j_r=1}^{N} \left( \frac{\hat{X}_{i,j_r}}{\sqrt{N}} \right)^{a_r} - \sum_{j_r \in \{j_1, \ldots, j_{r-1}\}} \left( \frac{\hat{X}_{i,j_r}}{\sqrt{N}} \right)^{a_r} \right)$$

$$= \left( \sum_{\substack{j_1, \ldots, j_{r-1} \in [N] \\ \text{distinct}}} \prod_{s=1}^{r-1} \left( \frac{\hat{X}_{i,j_s}}{\sqrt{N}} \right)^{a_s} \right) \left( \sum_{j_r=1}^{N} \left( \frac{\hat{X}_{i,j_r}}{\sqrt{N}} \right)^{a_r} \right)$$

$$- \left( \sum_{\substack{j_1, \ldots, j_{r-1} \in [N] \\ \text{distinct}}} \prod_{s=1}^{r-1} \left( \frac{\hat{X}_{i,j_s}}{\sqrt{N}} \right)^{a_s} \right) \left( \sum_{j_r \in \{j_1, \ldots, j_{r-1}\}} \left( \frac{\hat{X}_{i,j_r}}{\sqrt{N}} \right)^{a_r} \right). \tag{27}$$

We first want to analyze the second term. Note that this term is equal to

$$\sum_{t=1}^{r-1} \sum_{\substack{j_1, \ldots, j_{r-1} \in [N] \\ \text{distinct}}} \prod_{s=1}^{r-1} \left( \frac{\hat{X}_{i,j_s}}{\sqrt{N}} \right)^{a_s + \mathbb{1}_{[s=t]} a_r}.$$

Now, note that all terms in this sum have been considered in the induction hypothesis. Also note that $a_t + a_r \geq 3$ (since $a_r \geq 2$, otherwise there is nothing to prove) and thus every term in the above sum over $t$ is at most $O_d(\delta')$ in absolute value by (25). Thus, we also get that

$$\left| \left( \sum_{\substack{j_1, \ldots, j_{r-1} \in [N] \\ \text{distinct}}} \prod_{s=1}^{r-1} \left( \frac{\hat{X}_{i,j_s}}{\sqrt{N}} \right)^{a_s} \right) \left( \sum_{j_r \in \{j_1, \ldots, j_{r-1}\}} \left( \frac{\hat{X}_{i,j_r}}{\sqrt{N}} \right)^{a_r} \right) \right| \leq O_d(\delta'). \tag{28}$$

In the following it thus remains to analyze the first term in order to show (24) respectively (25)

$$\left( \sum_{\substack{j_1, \ldots, j_{r-1} \in [N] \\ \text{distinct}}} \prod_{s=1}^{r-1} \left( \frac{\hat{X}_{i,j_s}}{\sqrt{N}} \right)^{a_s} \right) \left( \sum_{j_r=1}^{N} \left( \frac{\hat{X}_{i,j_r}}{\sqrt{N}} \right)^{a_r} \right).$$

We now need to distinguish the cases $a_r \geq 3$ (in which case we need to show (25)) and $a_r = 2$ (in which case we need to show (24)).

**Case I:** $a_r \geq 3$, **proof of** (25). In this case, we have, by (12), that

$$\left| \sum_{j_r=1}^{N} \left( \frac{\hat{X}_{i,j_r}}{\sqrt{N}} \right)^{a_r} \right| \leq \delta'^{d+1}.$$

To analyze the term

$$\sum_{\substack{j_1,\ldots,j_{r-1}\in[N] \\ \text{distinct}}} \prod_{s=1}^{r-1} \left( \frac{\hat{X}_{i,j_s}}{\sqrt{N}} \right)^{a_s}$$

we want to apply the induction hypothesis. We need to again distinguish three cases, namely $a_{r-1} \geq 3$ (in which case we can use (25)), $a_{r-1} = 2$ (in which case we can use (24)) and $a_{r-1} = 1$ (in which case the term on the LHS of (24) are in fact equal). We do this in the following.

If $a_{r-1} \geq 3$, then, by the induction hypothesis for (25),

$$\left| \sum_{\substack{j_1,\ldots,j_{r-1}\in[N] \\ \text{distinct}}} \prod_{s=1}^{r-1} \left( \frac{\hat{X}_{i,j_s}}{\sqrt{N}} \right)^{a_s} \right| \leq O_d(\delta')$$

and thus we get, using the decomposition (27) and the bound (28) on the second term,

$$\left| \sum_{\substack{j_1,\ldots,j_r\in[N] \\ \text{distinct}}} \prod_{s=1}^{r} \left( \frac{\hat{X}_{i,j_s}}{\sqrt{N}} \right)^{a_s} \right| \leq O_d(\delta')\delta'^{d+1} + O_d(\delta') \leq O_d(\delta').$$

If $a_{r-1} = 2$, then, by the induction hypothesis for (24), we have that

$$\left| \sum_{\substack{j_1,\ldots,j_{r-1}\in[N] \\ \text{distinct}}} \prod_{s=1}^{r-1} \left( \frac{\hat{X}_{i,j_s}}{\sqrt{N}} \right)^{a_s} - \sum_{\substack{j_1,\ldots,j_{\hat{r}}\in[N] \\ \text{distinct}}} \prod_{s=1}^{\hat{r}} \frac{\hat{X}_{i,j_s}}{\sqrt{N}} \right| \leq O_d(\delta').$$

By the induction hypothesis on (26), we get

$$\left| \sum_{\substack{j_1,\ldots,j_{\hat{r}}\in[N] \\ \text{distinct}}} \prod_{s=1}^{\hat{r}} \frac{\hat{X}_{i,j_s}}{\sqrt{N}} \right| \leq O_d \left( \left( \frac{1}{\delta'} \right)^{r-1} \right).$$

Combining these two, we get

$$\left| \sum_{\substack{j_1,\ldots,j_{r-1}\in[N] \\ \text{distinct}}} \prod_{s=1}^{r-1} \left( \frac{\hat{X}_{i,j_s}}{\sqrt{N}} \right)^{a_s} \right| \leq O_d(\delta') + O_d \left( \left( \frac{1}{\delta'} \right)^{r-1} \right).$$

Thus, we get (note that $r \leq d$), using again the decomposition (27) and the bound (28) on the second term,

$$\left| \sum_{\substack{j_1,\ldots,j_r\in[N] \\ \text{distinct}}} \prod_{s=1}^{r} \left( \frac{\hat{X}_{i,j_s}}{\sqrt{N}} \right)^{a_s} \right| \leq \left( O_d(\delta') + O_d \left( \left( \frac{1}{\delta'} \right)^{r-1} \right) \right) \delta'^{d+1} + O_d(\delta') \leq O_d(\delta').$$

If $a_{r-1} = 1$, then $\hat{r} = r-1$ and thus

$$\sum_{\substack{j_1,\ldots,j_{r-1}\in[N] \\ \text{distinct}}} \prod_{s=1}^{r-1} \left( \frac{\hat{X}_{i,j_s}}{\sqrt{N}} \right)^{a_s} = \sum_{\substack{j_1,\ldots,j_{\hat{r}}\in[N] \\ \text{distinct}}} \prod_{s=1}^{\hat{r}} \frac{\hat{X}_{i,j_s}}{\sqrt{N}}.$$

As above, we get that, by the induction hypothesis on (26),

$$\left| \sum_{\substack{j_1,\ldots,j_{\hat{r}}\in[N]\\\text{distinct}}} \prod_{s=1}^{\hat{r}} \frac{\hat{X}_{i,j_s}}{\sqrt{N}} \right| \le O_d\left(\left(\frac{1}{\delta'}\right)^{r-1}\right).$$

Again by using the decomposition (27) and the bound (28) on the second term, we get

$$\left| \sum_{\substack{j_1,\ldots,j_r\in[N]\\\text{distinct}}} \prod_{s=1}^{r} \left(\frac{\hat{X}_{i,j_s}}{\sqrt{N}}\right)^{a_s} \right| \le O_d\left(\left(\frac{1}{\delta'}\right)^{r-1}\right)\delta'^{d+1} + O_d(\delta') \le O_d(\delta').$$

Thus, for all three cases, we get that (25) still holds for $r$.

**Case II:** $a_r = 2$, **proof of** (24). In this case, we have, by (11), that

$$\left| \sum_{j_r=1}^{N} \left(\frac{\hat{X}_{i,j_r}}{\sqrt{N}}\right)^{a_r} - 1 \right| \le \delta'^{d+1}.$$

For the term

$$\sum_{\substack{j_1,\ldots,j_{r-1}\in[N]\\\text{distinct}}} \prod_{s=1}^{r-1} \left(\frac{\hat{X}_{i,j_s}}{\sqrt{N}}\right)^{a_s},$$

we again need to distinguish two cases, namely $a_{r-1} = 2$ and $a_{r-1} = 1$.

If $a_{r-1} = 2$, then we have, as above, by the induction hypothesis for (24),

$$\left| \sum_{\substack{j_1,\ldots,j_{r-1}\in[N]\\\text{distinct}}} \prod_{s=1}^{r-1} \left(\frac{\hat{X}_{i,j_s}}{\sqrt{N}}\right)^{a_s} - \sum_{\substack{j_1,\ldots,j_{\hat{r}}\in[N]\\\text{distinct}}} \prod_{s=1}^{\hat{r}} \frac{\hat{X}_{i,j_s}}{\sqrt{N}} \right| \le O_d(\delta').$$

Also as above, by the induction hypothesis on (26), we get

$$\left| \sum_{\substack{j_1,\ldots,j_{\hat{r}}\in[N]\\\text{distinct}}} \prod_{s=1}^{\hat{r}} \frac{\hat{X}_{i,j_s}}{\sqrt{N}} \right| \le O_d\left(\left(\frac{1}{\delta'}\right)^{r-1}\right).$$

Combining these, we have that

$$\left| \left( \sum_{\substack{j_1,\ldots,j_{r-1}\in[N]\\\text{distinct}}} \prod_{s=1}^{r-1} \left(\frac{\hat{X}_{i,j_s}}{\sqrt{N}}\right)^{a_s} \right) \left( \sum_{j_r=1}^{N} \left(\frac{\hat{X}_{i,j_r}}{\sqrt{N}}\right)^{a_r} \right) - \sum_{\substack{j_1,\ldots,j_{\hat{r}}\in[N]\\\text{distinct}}} \prod_{s=1}^{\hat{r}} \frac{\hat{X}_{i,j_s}}{\sqrt{N}} \right|$$

$$\le \left| \sum_{\substack{j_1,\ldots,j_{r-1}\in[N]\\\text{distinct}}} \prod_{s=1}^{r-1} \left(\frac{\hat{X}_{i,j_s}}{\sqrt{N}}\right)^{a_s} - \sum_{\substack{j_1,\ldots,j_{\hat{r}}\in[N]\\\text{distinct}}} \prod_{s=1}^{\hat{r}} \frac{\hat{X}_{i,j_s}}{\sqrt{N}} \right| \left| \sum_{j_r=1}^{N} \left(\frac{\hat{X}_{i,j_r}}{\sqrt{N}}\right)^{a_r} \right|$$

$$+ \left| \sum_{\substack{j_1,\ldots,j_{\hat{r}}\in[N]\\\text{distinct}}} \prod_{s=1}^{\hat{r}} \frac{\hat{X}_{i,j_s}}{\sqrt{N}} \right| \left| \sum_{j_r=1}^{N} \left(\frac{\hat{X}_{i,j_r}}{\sqrt{N}}\right)^{a_r} - 1 \right|$$

$$\le O_d(\delta')(1 + \delta'^{d+1}) + O_d\left(\left(\frac{1}{\delta'}\right)^{r-1}\right)\delta'^{d+1}.$$

Thus, we get that, using again the decomposition (27) and the bound (28) on the second term,

$$\left| \sum_{\substack{j_1,\ldots,j_r\in[N]\\ \text{distinct}}} \prod_{s=1}^{r} \left( \frac{\hat{X}_{i,j_s}}{\sqrt{N}} \right)^{a_s} - \sum_{\substack{j_1,\ldots,j_{\hat{r}}\in[N]\\ \text{distinct}}} \prod_{s=1}^{\hat{r}} \frac{\hat{X}_{i,j_s}}{\sqrt{N}} \right|$$

$$\leq O_d(\delta')(1+\delta'^{d+1}) + O_d\left(\left(\frac{1}{\delta'}\right)^{r-1}\right)\delta'^{d+1} + O_d(\delta')$$

$$= O_d(\delta') + O_d(\delta')\delta'^{d+1} + O_d\left(\left(\frac{1}{\delta'}\right)^{r-1}\right)\delta'^{d+1} + O_d(\delta')$$

$$\leq O_d(\delta').$$

If $a_{r-1} = 1$, then we get

$$\sum_{\substack{j_1,\ldots,j_{r-1}\in[N]\\ \text{distinct}}} \prod_{s=1}^{r-1} \left( \frac{\hat{X}_{i,j_s}}{\sqrt{N}} \right)^{a_s} = \sum_{\substack{j_1,\ldots,j_{\hat{r}}\in[N]\\ \text{distinct}}} \prod_{s=1}^{\hat{r}} \frac{\hat{X}_{i,j_s}}{\sqrt{N}}.$$

Again, by the induction hypothesis on (26), we get

$$\left| \sum_{\substack{j_1,\ldots,j_{\hat{r}}\in[N]\\ \text{distinct}}} \prod_{s=1}^{\hat{r}} \frac{\hat{X}_{i,j_s}}{\sqrt{N}} \right| \leq O_d\left(\left(\frac{1}{\delta'}\right)^{r-1}\right).$$

As above, we get that

$$\left| \left( \sum_{\substack{j_1,\ldots,j_{r-1}\in[N]\\ \text{distinct}}} \prod_{s=1}^{r-1} \left( \frac{\hat{X}_{i,j_s}}{\sqrt{N}} \right)^{a_s} \right) \left( \sum_{j_r=1}^{N} \left( \frac{\hat{X}_{i,j_r}}{\sqrt{N}} \right)^{a_r} \right) - \sum_{\substack{j_1,\ldots,j_{\hat{r}}\in[N]\\ \text{distinct}}} \prod_{s=1}^{\hat{r}} \frac{\hat{X}_{i,j_s}}{\sqrt{N}} \right|$$

$$\leq \left| \sum_{\substack{j_1,\ldots,j_{\hat{r}}\in[N]\\ \text{distinct}}} \prod_{s=1}^{\hat{r}} \frac{\hat{X}_{i,j_s}}{\sqrt{N}} \right| \left| \sum_{j_r=1}^{N} \left( \frac{\hat{X}_{i,j_r}}{\sqrt{N}} \right)^{a_r} - 1 \right|$$

$$\leq O_d\left(\left(\frac{1}{\delta'}\right)^{r-1}\right)\delta'^{d+1}.$$

Thus, we can conclude that, exactly as above by the decomposition (27) and the bound (28) on the second term,

$$\left| \sum_{\substack{j_1,\ldots,j_r\in[N]\\ \text{distinct}}} \prod_{s=1}^{r} \left( \frac{\hat{X}_{i,j_s}}{\sqrt{N}} \right)^{a_s} - \sum_{\substack{j_1,\ldots,j_{\hat{r}}\in[N]\\ \text{distinct}}} \prod_{s=1}^{\hat{r}} \frac{\hat{X}_{i,j_s}}{\sqrt{N}} \right|$$

$$\leq O_d\left(\left(\frac{1}{\delta'}\right)^{r-1}\right)\delta'^{d+1} + O_d(\delta')$$

$$\leq O_d(\delta').$$

Hence, for both cases, we get that (24) still holds for $r$.

**Proof of** (26)**.** We can also analogously show (26). First note that if $\hat{r} < r-1$, then (26) follows directly from the induction hypothesis since the term

$$\sum_{\substack{j_1,\ldots,j_{\hat{r}}\in[N]\\ \text{distinct}}} \prod_{s=1}^{\hat{r}} \frac{\hat{X}_{i,j_s}}{\sqrt{N}}$$

already appeared for $r-1$ by using

$$a'_s = \begin{cases} a_s & \text{if } s < r-1 \\ a_{r-1} + a_r & \text{if } s = r-1 \end{cases}$$

and we can directly apply the induction hypothesis. Thus, we only need to show (26) for $\hat{r} \geq r-1$ and in particular $\hat{r} \geq 1$. If $\hat{r} = 1$, then by (10), the term is at most $\frac{1}{\delta'} \leq O_d\left(\left(\frac{1}{\delta'}\right)^r\right)$. Otherwise, $\hat{r} \geq 2$ and we can do a similar expansion as above to get

$$\sum_{\substack{j_1,\ldots,j_{\hat{r}} \in [N] \\ \text{distinct}}} \prod_{s=1}^{\hat{r}} \frac{\hat{X}_{i,j_s}}{\sqrt{N}}$$

$$= \left(\sum_{\substack{j_2,\ldots,j_{\hat{r}} \in [N] \\ \text{distinct}}} \prod_{s=2}^{\hat{r}} \frac{\hat{X}_{i,j_s}}{\sqrt{N}}\right) \left(\sum_{j_1=1}^{N} \frac{\hat{X}_{i,j_1}}{\sqrt{N}} - \sum_{j_1 \in \{j_2,\ldots,j_{\hat{r}}\}} \frac{\hat{X}_{i,j_1}}{\sqrt{N}}\right)$$

$$= \left(\sum_{\substack{j_2,\ldots,j_{\hat{r}} \in [N] \\ \text{distinct}}} \prod_{s=2}^{\hat{r}} \frac{\hat{X}_{i,j_s}}{\sqrt{N}}\right) \left(\sum_{j_1=1}^{N} \frac{\hat{X}_{i,j_1}}{\sqrt{N}}\right) - \left(\sum_{\substack{j_2,\ldots,j_{\hat{r}} \in [N] \\ \text{distinct}}} \prod_{s=2}^{\hat{r}} \frac{\hat{X}_{i,j_s}}{\sqrt{N}}\right) \left(\sum_{j_1 \in \{j_2,\ldots,j_{\hat{r}}\}} \frac{\hat{X}_{i,j_1}}{\sqrt{N}}\right).$$

The first term is then, by the induction hypothesis and (10), at most $O_d\left(\left(\frac{1}{\delta'}\right)^{r-1}\right)\frac{1}{\delta'}$ in absolute value. The second term can be expanded as above and is thus equal to

$$\sum_{t=2}^{\hat{r}} \sum_{\substack{j_2,\ldots,j_{\hat{r}} \in [N] \\ \text{distinct}}} \prod_{s=2}^{\hat{r}} \left(\frac{\hat{X}_{i,j_s}}{\sqrt{N}}\right)^{1+\mathbb{1}_{[s=t]}}$$

By (24) for $r-1$, we get that this is within $O_d(\delta')$ of

$$\sum_{t=2}^{\hat{r}} \sum_{\substack{j_2,\ldots,j_{t-1},j_{t+1}\ldots,j_{\hat{r}} \in [N] \\ \text{distinct}}} \prod_{\substack{s=2 \\ s \neq t}}^{\hat{r}} \frac{\hat{X}_{i,j_s}}{\sqrt{N}}.$$

(again if $\hat{r} = 2$, then the above term should be interpreted as 1). This term now is, by (26) for $r-2$, $O_d\left(\left(\frac{1}{\delta'}\right)^{r-2}\right)$. Note that for $r=2$ (and thus $\hat{r}=2$) we cannot apply the induction hypothesis but the term is 1 and thus the bound still holds.

Combing these result, we get that

$$\left| \sum_{\substack{j_1,\ldots,j_{\hat{r}} \in [N] \\ \text{distinct}}} \prod_{s=1}^{\hat{r}} \frac{\hat{X}_{i,j_s}}{\sqrt{N}} \right| \leq O_d\left(\left(\frac{1}{\delta'}\right)^{r-1}\right)\frac{1}{\delta'} + O_d(\delta') + O_d\left(\left(\frac{1}{\delta'}\right)^{r-2}\right)$$

$$\leq O_d\left(\left(\frac{1}{\delta'}\right)^r\right).$$

This is exactly (26) for $r$.

By induction, we have now shown (24) and (25) for all $r$ and as argued above, this completes the proof of this lemma. $\square$

