# OpenReview forum: "Testably Learning Polynomial Threshold Functions"
_NeurIPS.cc/2024/Conference — NeurIPS 2024 poster_

### Official Review · Reviewer_9MWZ · 2024-06-29

**Soundness:** 3
**Presentation:** 3
**Contribution:** 2
**Rating:** 7
**Confidence:** 2

**Summary:**

This paper studies testably learning an n-dimensional Polynomial Threshold function using a reduction proved in a previous work called fooling. The authors give an analysis of a construction of fooling multilinear PTF and then further for fooling arbitrary PTF.

The paper completes itself with proof that push-forward cannot learn PTF and that fooling is the best they can have.

**Strengths:**

The topic is very interesting, as there is a line of work on general testable learning and testable learning in specific classes like halfspace. Testable learning PTF is the natural next step.

Though the authors use the technique of reducing testable learning to fooling from previous work, it seems the analysis of the construction of a fooling for multilinear PTF is also an important technique. As the authors mentioned, the previous fooling construction [GKK23] only works for degree 2 PTF, and the construction from [Kane19] needs some careful analysis to work for PTF with more than constant degree.

The paper is complete as they also show fooling is necessary as another more direct approach cannot testably learn PTF.

**Weaknesses:**

The last part of the paper shows that push-forward cannot learn PTF seems very interesting, but it seems hard for me to determine how significant the contribution is from the analysis of fooling multilinear PTF.

**Questions:**

I am wondering what are the original techniques the authors want to emphasize. Like using Taylor expansion to bound additional error terms or reducing PTF to multilinear PTF or anything else? I am wondering if the authors can emphasize a bit more.

---

> ### Author Rebuttal · Authors · 2024-08-06
>
> We want to thank the reviewer for the kind feedback. We appreciate that the reviewer thinks of testable learning of PTFs as a natural problem to consider.
>
> The primary motivation for including the impossibility result for proving testable learning guarantees via the push-forward is to show that a straightforward approach (which yields reasonable results for halfspaces [RV23]) does not yield anything for PTFs. This in turn motivates our use of the (more complicated) techniques of Kane [Kan11] (i.e., fooling). Thereby, it partially explains the difference in dependence on $d$ between our result and existing results for agnostically learning PTFs.
>
> Regarding the question raised by the reviewer, we think the following are the main technical contributions of our paper, and we plan to emphasize these more in the revision:
> - First, as already alluded to by the reviewer, we bound the size of additional error terms arising from a Taylor expansion (see lines 290-307 of our paper), which do not appear in [Kan11]. We could imagine that our analysis of these terms could also be applied to different testable learning problems in the future.
> - Second, we generalize the arguments given by Kane to move from multilinear to arbitrary PTFs (lines 335-346 of our paper). There, we showed that even under the weaker assumption of approximate moment-matching, we can show that his construction also works in our setting. In particular, we were able to circumvent the use of moments of high degree (i.e., depending on $n$), which would have given us only quasi-polynomial runtime, i.e., $n^{\log(n)}$. On the contrary, we showed that fooling arbitrary PTFs needs the same degree of moment-matching as fooling multilinear ones, which allowed us to conclude our main result.
>
> [Kan11]: k-Independent Gaussians Fool Polynomial Threshold Functions, Daniel M. Kane, 2011 IEEE 26th Annual Conference on Computational Complexity
>
> [RV23]: Testing Distributional Assumptions of Learning Algorithms, Ronitt Rubinfeld, Arsen Vasilyan, Proceedings of the 55th Annual ACM Symposium on Theory of Computing

---

> > ### Comment · Reviewer_9MWZ · 2024-08-10
> >
> > Thank you for your response. I think this paper will be worth reading for people interested in testable learning and more enjoyable if the authors improving their writing in the final version in the way they claimed in this rebuttal. I raised my score to 7.

---

### Official Review · Reviewer_zio4 · 2024-07-12

**Soundness:** 4
**Presentation:** 4
**Contribution:** 3
**Rating:** 7
**Confidence:** 4

**Summary:**

Background: Agnostic learning is a well-studied framework that models learning when no function is some hypothesis class F describes the data perfectly. Specifically, the agnostic learning framework requires the learning algorithm to give a hypothesis whose classification error is at most opt+$\epsilon$, where opt is the best prediction error among all hypotheses in the class F.

Almost all existing agnostic learning algorithms are distribution-specific, i.e. they assume the examples are drawn from some distribution, for example a Gaussian distribution.

A distribution-specific agnostic learning algorithm lacks in reliability, because it is allowed to output an extremely poor classifier if the examples do not come from e.g. Gaussian (or some other assumed distribution). Yet, fully eliminating such assumptions is shown to be impossible for many basic function classes (based on well-established cryptographic assumptions).

Testable learning is a framework that aims to mitigate the above mentioned limitation, by allowing the algorithm to abstain on a specific dataset if the examples do not come from the assumed distribution. Overall, this allows a user to be confident that the classifier indeed has error of at most opt+$\epsilon$, as required by the agnostic learning framework. Testable learning has been a focus of many works in recent years (see the paper for references).

The paper studies testable learning of polynomial threshold functions (PTFs) under the Gaussian distribution. I.e. the function class F considered in this work consists of functions of the form sign$(p(x))$, where p is a degree-$d$ polynomial.  The work in n dimensions, with accuracy parameter $\epsilon$, the paper gives an algorithm for testable learning of constant-degree PTFs with a run-time of $n^{poly(1/\epsilon)}$}.

The paper is based on the moment-matching framework of [Gollakota, Klivans, Kothari ‘23], and shows that the direct approach used in [Vasilyan, Rubinfeld ‘23] to handle linear threshold functions cannot be extended to PTFs.

In order to apply the moment-matching framework of  [Gollakota, Klivans, Kothari ‘23], the paper shows that PTFs are “fooled” by distributions whose low-degree moments are close to Gaussian moments. To do this, the paper expands on the approach of [Kane ‘11] that proves a less general statement that PTFs are “fooled” by distributions for which the marginal of every k coordinates equals to the k-dimensional Gaussian. The proof first first considers multilinear PTFs, and then reduces the case of general PTFs to that of multilinear PTFs.

**Strengths:**

- Polynomial threshold functions are an extremely well-studied class of hypotheses that has been the focus of many works in learning theory (including in works that appeared in NeurIPS).
- Previously no testable learning algorithms were known even for degree-2 polynomial threshold functions.
- The run-time of $n^{poly(1/\epsilon)}$ qualitatively matches the best run-time for the agnostic learning of polynomial threshold functions. For example, existing hardness results preclude run-times such as $poly(n/\epsilon)$ or $n^{polylog(1/\epsilon)}$.
- Studying polynomial threshold functions naturally extends previous works that study linear threshold functions.

**Weaknesses:**

-The run-time dependence of the algorithm on the degree d of the PTF can conceivably be sub-optimal. As explained on page 12, the run-time is $(n \epsilon)^{O_d(\epsilon^{-4d7^d})}$, whereas it is conceivable that this run-time could potentially be improved in the future to $(n \epsilon)^{poly(d/\epsilon)}$

**Questions:**

The paper mentions that the analysis does not require to extend the Carbery-Wright inequality to distributions whose low-degree moments match those of the Gaussian. Could you give some more high-level intuitive explanation for why this is the case?

**Limitations:**

I think that the limitations are discussed adequately

---

> ### Author Rebuttal · Authors · 2024-08-06
>
> We wish to thank the anonymous reviewer for their kind feedback. We are encouraged that the reviewer believes the problem we study is well-motivated and naturally extends previous work and that the reviewer appreciates that our work qualitatively matches existing lower bounds even for the agnostic setting.
>
> We agree that it is conceivable (but likely difficult) that the runtime dependence on $d$ could be improved (or potentially lower bounds could be established). We address this in more detail in the general rebuttal and would refer the reviewer there. We think of this as an interesting direction for future research.
>
> Regarding the question raised by the reviewer, we were also somewhat surprised that we do not need an analogue of Carbery-Wright for moment-matching distributions. On a high-level, the reason is as follows. The point in the analysis when we need such an anti-concentration result is once we have shown that $\mathbb{E}[\mathrm{sign}(p(X) \pm O_d(\varepsilon^d))] \approx \mathbb{E}[\mathrm{sign}(p(Y))] \pm O(\varepsilon)$ (where as in our paper $Y$ is Gaussian and $X$ is approximately moment-matching). Now, we would like to say that the left-hand side is roughly the same as $\mathbb{E}[\mathrm{sign}(p(X))]$, which would exactly be an analogue of Carbery-Wright for moment-matching distribution (i.e. showing that the probability that $p(X)$ is small is low). However, the trick here (which was already used in [Kan11]) is to apply the above to the polynomial $p \mp O_d(\varepsilon^d)$ and thus shift the additional factor to the side with the Gaussian $Y$, where we then can apply Carbery-Wright (for Gaussians).
> Thus, once we have a result relating $\mathrm{sign}(p(Y))$ and $\mathrm{sign}(p(X) \pm O_d(\varepsilon^d))$, by changing the polynomial slightly, we can shift the additional factors to the $Y$ side which allows us to use Carbery-Wright for the Gaussian instead of an extension to approximately moment-matching distributions.
>
> [Kan11]: k-Independent Gaussians Fool Polynomial Threshold Functions, Daniel M. Kane, 2011 IEEE 26th Annual Conference on Computational Complexity

---

> > ### Comment · Reviewer_zio4 · 2024-08-07
> >
> > Thank you for your response.

---

### Official Review · Reviewer_2r1G · 2024-07-13

**Soundness:** 3
**Presentation:** 3
**Contribution:** 3
**Rating:** 6
**Confidence:** 2

**Summary:**

This paper studied the problem of testably learning polynomial threshold functions (PTFs). The authors aimed to answer the question of whether PTFs are qualitatively harder to learn in the testably learning model, compared to the agnostic learning model. The authors answered the question in the negative, showing that the degree-d PTFs can be testably learned up to $\epsilon$ with respect to the standard gaussian in time and sample complexity $n^{poly(1/\epsilon)}$, which qualitatively matches the $n^{O(d^2/\epsilon^4)}$ sample complexity of agnostic learning degree-d PTFs. To prove the above result, the authors linked testable learning with distribution fooling, building upon previous results on polynomial approximations. The authors also showed that it is impossible to testably learn PTFs with the techniques from [RV23].

**Strengths:**

1. The paper provided the first sample and computational complexity on testably learning PTFs, showing that testably learning PTFs is qualitatively similar in hardness to agnostic learning PTFs. To reach this result, the authors overcome a handful of technical obstacles that arose in adapting the fooling techniques to testable learning. Critically, the authors constructed a new low-degree polynomial to approximate the PTFs based on [Kane11] with refined approximation error bounds. The techniques the authors applied here could be of independent interest.

**Weaknesses:**

1. Though the final sample complexity is $n^{poly(1/\epsilon)}$, the $poly(1/\epsilon)$ is of order $\epsilon^{-7^d}$, in other words, it is substantially worse than agnostic learning in terms of the order of $1/\epsilon$. It might be too harsh to say this is a serious weakness of this paper, as this is the first paper that provided these kinds of complex results; I think it would be an interesting future work to reduce the order of $1/\epsilon$ that is truly comparable to agnostic learning.

**Questions:**

see weakness above.

**Limitations:**

the authors have addressed the limitations properly.

---

> ### Author Rebuttal · Authors · 2024-08-06
>
> First, we would like to thank the reviewer for their kind feedback. We are encouraged by the fact they appreciate that we give the first result on testably learning PTFs.
>
> We agree with the reviewer that it would be interesting future work to try to improve the runtime, potentially to $n^{\mathrm{poly}(d/\varepsilon)}$. We address the dependence of the runtime on $d$ in the general rebuttal. Briefly, our worse runtime dependence on $d$ (w.r.t. the agnostic model) is inherited from the result of [Kan11], which we build on. An improvement of the runtime using our techniques would directly improve the result of [Kan11], which has stood for over 10 years. Furthermore, under widely believed hypotheses, the best runtime we could hope for is $n^{\mathrm{poly}(d/\varepsilon)}$.
>
> [Kan11]: k-Independent Gaussians Fool Polynomial Threshold Functions, Daniel M. Kane, 2011 IEEE 26th Annual Conference on Computational Complexity

---

### Official Review · Reviewer_2v3u · 2024-07-13

**Soundness:** 3
**Presentation:** 3
**Contribution:** 2
**Rating:** 5
**Confidence:** 4

**Summary:**

The authors study the problem of testing polynomial threshold functions in the agnostic setting in the testable learning paradigm. They present a testable learning algorithm that matches with the asymptotic bound (in terms of n) known for agnostic learning

**Strengths:**

Testing of polynomial threshold functions is a very natural problem and the testable learning paradigm is also very natural. The authors present the first such testable learning algorithm in the agnostic learning setting for PTFs.

**Weaknesses:**

The dependence on epsilon and the degree is very bad. This makes the results completely useless is practice. In fact the degree os the polynomial on the exponent is exponentially dependent on d.

**Questions:**

Can you please argue why the dependence on d is so bad and is there possibility of improving it. Can a better bounds be obtained for smaller d, like d=2,3.

---

> ### Author Rebuttal · Authors · 2024-08-06
>
> We wish to thank the anonymous reviewer for their feedback and questions. We are happy that the reviewer finds testable learning, particularly of PTFs, to be a very natural topic.
>
> The primary concern of the reviewer appears to be the dependence of our results on the error $\varepsilon$ and the degree $d$ of the PTF. In particular, that this dependence impacts the practical relevance of our results. While this is certainly a fair criticism, we would like to point out that:
> - By considering the testable model (rather than the standard agnostic model), we are actually taking a step *towards* practicality. Indeed, in the standard agnostic model, one makes assumptions on the distribution of the data which cannot be verified algorithmically (neither in theory nor in practice). On the other hand, in the testable model, one relies only on properties of the data which can be verified directly (in our case, via moment matching). The testable model is therefore harder (leading to worse dependences on the problem parameters), but also a better reflection of practical reality.
> - Even in the (easier) agnostic model, the runtime dependence of known algorithms on $\varepsilon$ and $d$ is quite bad, and moreover, this dependence cannot be improved much under typical hardness assumptions. In particular, under these assumptions, it is not possible to find an algorithm which is polynomial both in $n$ and in $1/\varepsilon$. For instance, [Kan11a] shows a runtime of $n^{O(d^2/\varepsilon^4)}$, which is beyond practical computation already when, say, $d=2$ and $\varepsilon = 0.25$.
> - The primary motivation of our paper is to gain further theoretical understanding of which learning problems might be 'hard' or 'easy' in some asymptotic sense. This follows a long line of papers in learning theory, some of which appeared in earlier editions of NeurIPS. Our main goal was to show that, for any fixed $d$ and $\varepsilon$, PTFs can be testably learned in polynomial time in $n$, thus qualitatively matching the agnostic setting. We believe that our result, while not immediately applicable in practice, is nonetheless of interest to the audience of NeurIPS.
>
> We give a more detailed explanation of why our dependence on $d$ is worse than in the agnostic model in our general rebuttal. In short, we inherit our dependence from [Kan11], and it seems nontrivial to improve them. We think determining the best-possible dependence is an interesting direction for future research.
>
> The reviewer asks specifically whether improvements are possible for small values of $d$. This is as an interesting suggestion. We rely on the 'fooling' result [Kan11] because it applies to PTFs of arbitrary degree. For PTFs of degree $d=2$ an earlier paper [DKN10] achieves a similar result to [Kan11], but with better (and more explicit) dependence on $\varepsilon$. It would be interesting to see if the result of [DKN10] can be translated to testable learning to achieve better dependence for $d=2$. However, we note that such a translation would likely require substantial additional technical effort. As our focus in this paper was to achieve a result for all choices of $d$ simultaneously, we did not pursue this direction.
>
> [Kan11]: k-Independent Gaussians Fool Polynomial Threshold Functions, Daniel M. Kane, 2011 IEEE 26th Annual Conference on Computational Complexity
>
> [Kan11a]: The Gaussian Surface Area and Noise Sensitivity of Degree-d Polynomial Threshold Functions, Daniel M. Kane, computational complexity vol. 20
>
> [DKN10] Bounded Independence Fools Degree-2 Threshold Functions, Ilias Diakonikolas, Daniel M. Kane, Jelani Nelson, 2010 IEEE 51st Annual Symposium on Foundations of Computer Science

---

> > ### Comment · Reviewer_2v3u · 2024-08-13
> >
> > Thanks for your detailed response.

---

### Official Review · Reviewer_pSUk · 2024-07-17

**Soundness:** 3
**Presentation:** 3
**Contribution:** 2
**Rating:** 6
**Confidence:** 3

**Summary:**

This paper investigates testable learning of polynomial threshold functions (PTFs) with respect to the standard Gaussian distribution. The authors extend previous work on testable learning of halfspaces to show that PTFs of arbitrary constant degree can be testably learned up to excess error \epsilon in time n^poly(1/\epsilon),  matching the best known guarantees in the agnostic model. The key technical contribution is showing that distributions which approximately match the moments of a Gaussian up to degree poly(1/\epsilon) fool constant-degree PTFs.

**Strengths:**

- The paper is well written and easy to follow.
- The paper makes significant progress on an open problem in learning theory by extending testable learning to PTFs.

**Weaknesses:**

- It is not clear how important the problem of testable learning for PTFs is and if the results and/or techniques have applicability to other learning theory problems.

**Questions:**

- It seems like the testable learning setting is similar to the samples coming from a distribution close to the distribution D in some sense? Is there any model of learning theory that explicitly studies this?
- Do you see a path to improving the sample complexity to polynomial in both n and 1/\epsilon, rather than n^poly(1/\epsilon)?
- The authors have looked at Gaussian distribution in this work. What other distributions could this result be extended to?
- Can you provide intuition for why the runtime dependence on d is so much worse than in the agnostic model? Do you believe this gap is inherent or an artifact of the analysis?
- Did the authors look into getting any sort of lower bounds for this problem?

**Limitations:**

Yes

---

> ### Author Rebuttal · Authors · 2024-08-06
>
> We thank the reviewer for their kind feedback and insightful questions. We appreciate that the reviewer found our paper easy to follow, and that they believe we make significant progress on an open problem in learning theory.
>
> The main concerns of the reviewer appear to be the relative importance of the problem we study (testable learning of PTFs), and the potential applicability of our results and techniques to other problems in learning theory.
>
> To motivate the importance of our main result, we wish to briefly discuss the importance of PTFs and the testable learning model:
> - As reviewer zio4 mentions, PTFs are very well-studied in (theoretical) computer science, and in particular in learning theory. They are a natural extension of linear threshold functions, introducing non-linearity while maintaining some amount of structure. For this reason, they are often used as a test-case to determine the boundaries of efficient learning algorithms, which is also their role in our paper.
> - The testable learning model was introduced relatively recently as an extension of the (standard) agnostic model. It has already attracted significant attention, evidenced by several publications in leading conferences (including NeurIPS) [DKK+23, GKK23, GKSV23, GKSV24, RV23]. A recurring theme in these works is an attempt to determine whether testable learning comes at an additional computational cost with respect to agnostic learning. Our work continues on this theme by proving that, qualitatively speaking, the class of PTFs can be testably learned at no additional cost (for any fixed $d$). No such results were previously available, even for $d=2$.
>
> Beyond the inherent importance of our results, we believe there is potential for future applications of our proof techniques (as mentioned by reviewer 2r1G). We refine an earlier error analysis of [Kan11], thereby translating a fooling result for $k$-independent Gaussians to a testable learning guarantee. Our methods could prove useful in future translations from existing results in approximation theory to testable learning, as well.
>
> Finally, we wish to reply to the questions by reviewer pSUk:
> 1. This is the right intuition. In the context of our paper, where we only consider approximate moment-matching to test the data, testable learning corresponds to agnostic learning w.r.t. the class of distributions whose low-degree moments are close to those of a Gaussian. However, the testable model is more general than this, as it makes no assumptions on the type of testing algorithm used. This means we do not have such a correspondence in general. Agnostic learning w.r.t. classes of distributions that contain (but are broader than) the Gaussian has been considered in the literature before. A key distinction is that previous approaches typically focus on a class with nice mathematical properties (e.g., log-concave distributions), whereas in the testable model one considers classes for which membership can be verified efficiently from a small sample.
> 2. For this question, we would refer the reviewer to our general rebuttal. In short, there is evidence that this is not possible, even for the easier agnostic learning model with respect to the Gaussian. This is not made sufficiently clear in the paper, and we will improve this in the revision.
> 3. This is an interesting direction for potential future work: The most natural generalization that we see, based also on other works on testable learning, e.g., [GKK23], [GKSV24], [GKSV23], is to strongly log-concave distributions. However, the result in our paper does not directly generalize to any other distribution than Gaussian.
> 4. For this question, we would again refer to our general rebuttal. In short, it is not clear whether the gap is inherent to the testable model or arises from our proof techniques. We inherit our dependences from [Kan11], and it seems nontrivial to improve them. Intuitively, one expects worse dependences in the testable model as one needs a stronger notion of polynomial approximation than in the (standard) agnostic model for the polynomial regression algorithm to (provably) work (compare Thm. 6 to Thm. 7). In the paper, we work with the notion of 'fooling', which also corresponds to a strong notion of polynomial approximation (namely 'sandwiching', cf. Lines 228-232).
> 5. We did not study lower bounds specifically for *testable* learning of PTFs beyond the impossibility result for the approach used by [RV23] (cf. Section 2.4). (As mentioned in our answer to question 2, there is a lower bound for agnostic learning that also applies to the testable setting). However, we think it is an interesting future research direction to either prove a computational gap between agnostic and testable learning (w.r.t. $d$), or prove that no gap exists.
>
> [DKK+23]: Efficient Testable Learning of Halfspaces with Adversarial Label Noise, Ilias Diakonikolas, Daniel M. Kane, Vasilis Kontonis, Sihan Liu, Nikos Zarifis, Advances in Neural Information Processing Systems 36 (NeurIPS 2023)
>
> [GKK23]: A Moment-Matching Approach to Testable Learning and a New Characterization of Rademacher Complexity, Aravind Gollakota, Adam R. Klivans, Pravesh K. Kothari, Proceedings of the 55th Annual ACM Symposium on Theory of Computing
>
> [GKSV23]: Tester-Learners for Halfspaces: Universal Algorithms, Aravind Gollakota, Adam R. Klivans, Konstantinos Stavropoulos, Arsen Vasilyan, Advances in Neural Information Processing Systems 36 (NeurIPS 2023)
>
> [GKSV24]: An Efficient Tester-Learner for Halfspaces, Aravind Gollakota, Adam R. Klivans, Konstantinos Stavropoulos, Arsen Vasilyan, The Twelfth International Conference on Learning Representations
>
> [Kan11]: k-Independent Gaussians Fool Polynomial Threshold Functions, Daniel M. Kane, 2011 IEEE 26th Annual Conference on Computational Complexity
>
> [RV23]: Testing Distributional Assumptions of Learning Algorithms, Ronitt Rubinfeld, Arsen Vasilyan, Proceedings of the 55th Annual ACM Symposium on Theory of Computing

---

### Author Rebuttal · Authors · 2024-08-06

First and foremost, we would like to thank the reviewers for their time and valuable feedback. We appreciate that many of the reviewers consider testably learning PTFs as a natural and well-motivated problem.

Several reviewers correctly pointed out that the dependence of our runtime on $d$ (the degree of the PTF) is worse than in the (standard) agnostic setting. We inherit this dependence from the result in [Kan11] since we modify their construction. We note that the dependence in our result is no worse than in [Kan11], even though it holds in a strictly more general setting. Any improvement to our dependence would immediately imply an improvement over the results in [Kan11]. While in [Kan11] it was speculated that such a better dependence is conceivable, this was not achieved since the first dissemination of the result over ten years ago. In light of this, achieving a better dependence on $d$ is a great, but likely difficult open question for future work. It is also related to the intriguing question whether a strictly larger runtime is necessary for testable learning over agnostic learning. We would also like to highlight that our impossibility result (Section 2.4) rules out a natural, "simpler" approach to prove guarantees for testably learning PTFs. In some sense, this shows that achieving better runtime dependences for testably learning PTFs is likely difficult without also improving the result of [Kan11].

Furthermore, we would like to stress that even for $d = 2$, our result is the first for testably learning PTFs. Moreover, as pointed out by reviewer zio4, for any fixed $d$, our dependence on $\varepsilon$ is qualitatively optimal in the sense that it matches known lower bounds (which hold even in the simpler agnostic setting). In particular, for a fixed $d$, our runtime scales as $n^{\mathrm{poly}(1/\varepsilon)}$ and known lower bounds (either in the SQ model [DKPZ21] or under standard cryptographic assumptions [Tie23]) imply that this is necessary. For example, these lower bounds rule out runtimes such as $\mathrm{poly}(n, 1/\varepsilon)$.

Kind regards, the Authors




[Kan11]: k-Independent Gaussians Fool Polynomial Threshold Functions, Daniel M. Kane, 2011 IEEE 26th Annual Conference on Computational Complexity

[DKPZ21]: The Optimality of Polynomial Regression for Agnostic Learning under Gaussian Marginals in the SQ Model, Ilias Diakonikolas, Daniel M. Kane, Thanasis Pittas, Nikos Zarifis, Proceedings of Thirty Fourth Conference on Learning Theory

[Tie23]: Hardness of Agnostically Learning Halfspaces from Worst-Case Lattice Problems, Stefan Tiegel, Proceedings of Thirty Sixth Conference on Learning Theory

---

### Decision · Program_Chairs · 2024-09-25

**Decision:**

Accept (poster)

**Comment:**

Testable learning is a recently proposed learning paradigm where algorithms are required to "check" the assumptions they need to make on data to succeed, instead of taking them on faith. This paper shows how to testably learn a polynomial threshold function that succeeds with high probability when the input data is Gaussian (or passes the check), extending previous results from the agnostic setting. The reviewers generally agreed that this is an interesting and nontrivial theoretical result which goes beyond the prior work in this area. I recommend acceptance.